



# Technical Note: Classical and statistical thermodynamic treatment of adsorption and desorption kinetics and rates

Daniel A. Knopf[1], Markus Ammann[2]

[1]School of Marine and Atmospheric Sciences, Stony Brook University, Stony Brook, New York, USA
5   [2]Laboratory of Environmental Chemistry, Paul Scherrer Institute, Villigen, Switzerland

*Correspondence to*: Daniel A. Knopf (daniel.knopf@stonybrook.edu) and Markus Ammann (markus.ammann@psi.ch)

**Abstract.**

Adsorption and desorption represent the initial processes of the interaction of gas species with the condensed phase. It has important implications for evaluating heterogeneous (gas-to-solid) and multiphase chemical kinetics involved in catalysis, 10   environmental interfaces, and, in particular, aerosol particles. When describing gas uptake, gas-to-particle partitioning, and the chemical transformation of aerosol particles the desorption lifetime is a crucial parameter to assess the underlying chemical kinetics such as surface reaction and surface-to-bulk transfer. The desorption lifetime, in turn, depends on the desorption free energy which is affected by the chosen adsorbate model and standard states. To assess the impact of those conditions on desorption energy and, thus, desorption lifetime, we provide a complete classical and statistical thermodynamic treatment of 15   the adsorption and desorption process considering transition state theory for two typically applied adsorbate models, the 2D ideal gas and the 2D ideal lattice gas, the latter being equivalent to Langmuir adsorption. Both models apply to solid and liquid substrate surfaces. We derive the thermodynamic and microscopic relationships for adsorption and desorption equilibrium constants, adsorption and desorption rates, first-order adsorption and desorption rate coefficients, and the corresponding pre-exponential factors. Although, some of these derivations can be found in the literature, this study aims to bring all derivations 20   into one place to facilitate the interpretation and analysis of desorption energies for their application in multiphase chemical kinetics. This exercise allows for a microscopic interpretation of the underlying processes including the surface accommodation coefficient and highlights the importance of the choice of adsorbate model and standard states when analyzing and interpreting adsorption and desorption processes. We demonstrate how the choice of adsorbate model choice affects equilibrium surface concentrations and coverages, desorption rates, and decay of the adsorbate species with time. In addition, 25   we show how those results differ when applying a concentration- or activity-based description. Our treatment demonstrates that the pre-exponential factor can differ by orders of magnitude depending on the choice of adsorbate model with similar effects on the desorption lifetime, yielding significant uncertainties in the desorption energy. Furthermore, uncertainties in surface coverage and assumptions in standard surface coverage can lead to significant changes in desorption energies derived from measured desorption rates. Providing a comprehensive thermodynamic and microscopic representation aims to guide 30   theoretical and experimental assessments of desorption energies and estimate potential uncertainties in applied desorption energies and corresponding desorption lifetimes important for improving our understanding of multiphase chemical kinetics.



**Short Summary (500 character in total)**

Adsorption on and desorption of gas molecules from solid or liquid surfaces or interfaces represent the initial interaction of gas-to-condensed phase processes that can define the physicochemical evolution of the condensed phase. We apply a thermodynamic and microscopic treatment of these multiphase processes to evaluate how adsorption and desorption rates and surface accommodation depend on the choice adsorption model and standard states with implications for desorption energy
and lifetime.

## 1 Introduction

Any interaction between gas-phase species and condensed matter, including liquid, semi-solid, and solid phases, commences by adsorption and desorption processes (McNaught and Wilkinson, 2014;Langmuir, 1918, 1916, 1915). Those are of importance in the research areas of catalysis and, in particular, multiphase chemical kinetics or phase transfer kinetics involving
environmental surfaces and interfaces (Cussler, 2009;Chorkendorff and Niemantsverdriet, 2007;Finlayson-Pitts and Pitts, 2000;Ravishankara, 1997;Solomon, 1999). Surfaces including water bodies, ice, and terrestrial and anthropogenic structures can provide interfaces at which phase transfer processes, multiphase and heterogeneous reactions can take place. In the atmospheric sciences, multiphase chemical reactions have been the foci of research since the realization that heterogeneous reactions on the surface of polar stratospheric clouds lead to the activation of inert chlorine reservoir species that subsequently
result in ozone depletion, manifested in the spring southern hemispheric ozone hole (Solomon, 1999;Rowland, 1991). By now it is well established that gas-particle interactions play crucial roles in particle growth by condensation, gas-particle partitioning, and the chemical evolution of particles during aerosol formation and aging (Pöschl et al., 2007;Kolb et al., 2010;Rudich et al., 2007;George and Abbatt, 2010;Pöschl and Shiraiwa, 2015;Moise et al., 2015;Ammann et al., 2013;Crowley et al., 2013;Kroll et al., 2011;Donahue et al., 2011;Jimenez et al., 2009). The role of reversible adsorption and desorption has
been addressed in many studies of gas uptake and multiphase chemical reactions, in particular for the decoupling of mass transport and chemical reaction (Kolb et al., 1995;Hanson and Ravishankara, 1991;Kolb et al., 2010;Ammann et al., 2013;Crowley et al., 2013;Pöschl and Shiraiwa, 2015).

According to the Frenkel equation, the desorption lifetime ($\tau_d$) of a surface-adsorbed chemical species (i.e., the adsorbate) follows an Arrhenius-type behavior (Arrhenius, 1889b;Arrhenius, 1889a;Laidler, 1949;Frenkel, 1924;Laidler et al., 1940):

$$\tau_d = \frac{1}{k_d} = \frac{1}{A} e^{\frac{E^0_{des}}{RT}}, \tag{1}$$

where $E^0_{des}$ is the desorption energy with the energy reference of the gas molecule at rest at $T = 0\ K$ (as outlined below), $k_d$ is a first-order desorption rate coefficient, $A$ is a pre-exponential factor, $R$ is the general gas constant, and $T$ is temperature.





Activated adsorption and desorption processes are treated by considering an additional activation barrier. When describing multiphase chemical kinetics, $k_d$ affects the overall rate of transfer of a gas molecule into the bulk by impacting the loss rate

by surface reaction and the surface-to-bulk transfer and, thus, the bulk accommodation coefficient (Ammann and Pöschl, 2007;Pöschl et al., 2007;Shiraiwa and Poschl, 2021). For example, a kinetic multilayer model analysis of measured uptake coefficients for OH radicals on levoglucosan substrates yielded a tight correlation between $\tau_d$ and the chemical reaction rate coefficient at the surface, because the experimental data only allowed to constrain the product/ratio of the two (Arangio et al., 2015). A similar issue, the competition between adsorption (and uptake) and desorption, pertains to gas-particle partitioning

kinetics when describing condensation of water vapor and volatile organic compounds (VOCs) and volatilization of organic reaction products (Shiraiwa et al., 2013;Shiraiwa et al., 2012;Shiraiwa and Seinfeld, 2012). Thus, accurate derivation of the chemical reaction kinetics requires accurate $\tau_d$ values. Atmospheric trace gases and water vapor can undergo reversible adsorption on aerosol, cloud, and ground surfaces over a wide range of temperatures from below 200 to above 300 K. Especially at low temperatures, large values of $\tau_d$ could counteract slow rates of chemical reaction and diffusion, enhancing

the overall gas uptake which may involve reversible, reactive, and catalytic processes on the surface or in the bulk of the particles (Ammann et al., 2013;Crowley et al., 2013;Kolb et al., 2010;Pöschl et al., 2007;Rudich et al., 2007;Li et al., 2020;Li and Knopf, 2021).

Equation (1) does not explicitly show that the desorption rate depends on the choice of adsorbate model and standard states. Once the pre-exponential factor $A$ is expressed in terms of the free energy of activation (Campbell et al.,

2016;Donaldson et al., 2012;Kolasinski, 2012), adsorbate model and standard states can lead to significant changes in the pre-exponential factor $A$ and thus $\tau_d$. Vice versa, when using experimentally observed desorption rates to derive $E_{des}^0$, assumptions about the adsorbate model can result, as we show in this study, in significant changes in the corresponding $E_{des}^0$ values. It is known that the choice of standard states and adsorbate model impacts the interpretation of the equilibrium constant and desorption process (Campbell et al., 2016;Donaldson et al., 2012;Kolasinski, 2012).

The difference in adsorbate models reflects the treatment of the potential well in which the adsorbate "sits" in (Hill, 1986;Campbell et al., 2016). The most commonly applied adsorbate model is the 2D ideal gas which lacks one translational degree of freedom compared to the 3D ideal gas (Hill, 1986). It is defined by the condition of negligible lateral potential wells; thus, it can freely move parallel across the surface. The other extreme is the 2D ideal lattice gas where the absorbate cannot overcome the potential well of the adsorption site. Thus, it exerts only vibrational movements parallel and vertical to the

surface. A model that can describe both extremes is, e.g., the ideal hindered translator model (Hill, 1986;Campbell et al., 2016;Sprowl et al., 2016). This new adsorbate model (Sprowl et al., 2016) is not discussed in this study. The choice of adsorbate model and corresponding standard states will result in different equilibrium constants, pre-exponential factors, and, thus, desorption rates. This, ultimately, will also render $E_{des}^0$ and $\tau_d$, important parameters when examining and interpretating the multiphase chemical kinetics at environmental interfaces.

The purpose of this study is to provide a holistic description of the thermodynamic functions derived from microscopic principles (i.e., corresponding partition functions) that allow for the calculation of the pre-exponential factor of the desorption





rate based on transition state (TS) theory for the case of the 2D ideal gas and 2D ideal lattice gas. We will apply statistical thermodynamics to describe the microscopic, i.e., on the molecular level, processes and classical thermodynamics that define the overall energy and equilibrium conditions. Although, some aspects of the presented derivations can be found in statistical

thermodynamic textbooks (Hill, 1986;Kolasinski, 2012) and articles (Campbell et al., 2016;Donaldson et al., 2012;Savara, 2013), a complete treatment of adsorption and desorption including the TS and respective standard states is not readily available in the literature, as far as the authors are aware of. An outcome of this exercise is an improved understanding of the defining parameters that govern typically measured and reported thermodynamic parameters and their dependency on chosen standard states. For example, the presented derivations demonstrate that the pre-exponential factor, commonly assumed to be around

$10^{13}$ s$^{-1}$ (Atkins and de Paula, 2006), can differ by orders of magnitude in response to the choice of standard state and adsorbate model (Campbell et al., 2016). This, in turn, will alter interpretation and analyses of multiphase chemical kinetics occurring at interfaces.

The outline of this study is guided by ways of the derivation of the thermodynamic functions. TS theory assumes thermodynamic equilibrium between the adsorbed state and the TS for desorption (Kolasinski, 2012;Eyring, 1935). The

description of this equilibrium in terms of the basic thermodynamic functions is based on adsorption thermodynamics. Since the desorption rate and the pre-exponential factor are expressed in terms of molecular properties (i.e., the microscopic picture), the linkage between statistical thermodynamics and the thermodynamic functions have to be considered and applied. However, the foundational derivations for the thermodynamics and statistical thermodynamics of adsorption are not well established and not treated in comprehensive ways in textbooks. We therefore retrace this theory first for the case of desorption as an overall

process. This will then serve as the basis for applying this theory to the TS theory for desorption and adsorption and to derive the pre-exponential factor for desorption. Subsequently, combination of the rate expression of desorption and adsorption establishes the links between the overall adsorption thermodynamics and the microscopic kinetic parameters including the interpretation of the surface accommodation coefficient. Lastly, we evaluate how our findings impact interpretation and analysis of measured or theoretically derived $E^0_{des}$ values.

Since the basis for describing desorption by TS theory requires consideration of thermodynamic equilibria, in section 2 to 5 and the Supplement, we introduce first the overall desorption thermodynamics in more detail to provide the necessary equations and terminology. Section 2 discusses the general thermodynamic functions for describing adsorption and desorption, their derivations from microscopic properties (partition functions), and definitions of the standard states. Section 3 provides the derivation of equilibrium thermodynamic functions that describe the desorption process for the two different adsorbate

models. The results so far will be applied in Section 4 to derive the desorption rates and associated pre-exponential factors for the different adsorbate models in terms of thermodynamic and microscopic quantities. Section 5 presents the derivation of the adsorption rate including thermodynamic and microscopic treatment and evaluation of the surface accommodation coefficient. In Section 6, by combination of the previous results we consider the equilibrium between adsorption and desorption to derive the corresponding equilibrium constants demonstrating that the derivations are internally consistent. Section 7 provides the

derivation of the kinetic parameters from equilibrium between adsorption and desorption. Section 8 discusses how the choices



made for standard states and the type of adsorbate model impact surface concentration, activity, and coverage, adsorption and desorption rates, and $E_{des}^0$ and $\tau_d$ values and thus our interpretation of multiphase chemical kinetics. This is followed by the conclusions section.

To fundamentally follow all derivations presented in this document, an excess number of equations would need to be shown, which would have rendered this document difficult to read. In the supplement we provided all necessary definitions, equations, and derivations from first principles to follow the thoughts in the main document. The reader is encouraged to study this document side-by-side with the Supplement that contains all information leading to the results shown here. We apply the definitions of parameters and standard states given in the Supplement. The Supplement includes all necessary detailed derivations of the thermodynamic equations for 3D ideal gas, 2D ideal gas, 2D ideal lattice gas, and TS. It includes the

following sections: (S1) Definition of desorption and adsorption equilibrium constants; (S2) Derivation of thermodynamic functions for desorption and adsorption; (S3) Standard molar enthalpies, entropies, and Gibbs free energies; (S4) Derivation of Equilibrium Constants; (S5) Standard molar Gibbs free energy change and equilibrium constant between the 3D ideal gas and the transition state for adsorption; (S6) Adsorption-Desorption Equilibrium.

## 2. Thermodynamic and Microscopic Considerations of the Adsorption/Desorption Process

In this section we define the nomenclature, signage, and units involved of partition functions, thermodynamic quantities, and standard states when describing the activated and non-activated adsorption and desorption processes.

### 2.1 Gibbs Free Energy, Enthalpy, and Entropy of the Adsorption and Desorption Process

The spontaneous occurrence of adsorption implies an exergonic process with the thermodynamic condition (Bolis, 2013):

$$\Delta G_{ads,m}^0 = \Delta H_{ads,m}^0 - T\Delta S_{ads,m}^0 < 0 \tag{2}$$

$$\Delta H_{ads,m}^0 = H_{ads,m}^0 - H_{g,m}^0 = -\Delta H_{des,m}^0 \tag{3}$$

$$\Delta S_{ads,m}^0 = S_{ads,m}^0 - S_{g,m}^0 = -\Delta S_{des,m}^0 \tag{4}$$

Since adsorption of a gas on a substrate results in an increase of molecular ordering and $\Delta S_{ads,m}^0 < 0$, the change in enthalpy $\Delta H_{ads,m}^0$ has to be negative. In this study, $\Delta G_m^0$ and $\Delta H_m^0$ are expressed in units J mol$^{-1}$ and $\Delta S_m^0$ in units J mol$^{-1}$ K$^{-1}$.

For the remainder of the text, the subscripts denote the process direction in the order of (from left to right) process (adsorption

or desorption), educt (e.g., adsorbate), and product (e.g., gas species). Subscript $m$ denotes molar quantities.

### 2.2 Adsorption and Desorption Energy and Activation Barrier

We define the energy reference as the inner energy of the gas molecule at rest at $T = 0\ K$. The adsorbed or desorbing molecule is at the bottom of a potential well, at $-q_{des}^0$ with $q_{des}^0$ being a positive number in units of Joule indicating the



necessary heat for the molecule to desorb as depicted in Fig. 1. Different processes can contribute to $q_{des}^0$ such as molecular rotations and vibrations or other molecular interactions. In molecular quantities and at constant volume, accounting for the number of adsorbed molecules in the system, $N_{ads}$, yields

$$U_{ads}(0) = -N_{ads}q_{des}^0 = -\varepsilon_{des}^0, \tag{5}$$

where $\varepsilon_{des}^0$ represents the molecular desorption energy and in molar quantities we obtain

$$E_{des}^0 = N_A q_{des}^0 \tag{6}$$

and, thus,

$$U_{ads,m}(0) = -E_{des}^0 . \tag{7}$$

We treat the general case of activated adsorption/desorption here, meaning that the TS's inner energy is elevated by the barrier height above the reference level. For an activated adsorption and desorption process the molecule in the TS does not have any interactions with the surface but sits on top of the energy barrier, $q_b^0$ (Fig. 1). In other words, $q_b^0$ must not dependent

on the interaction forces that make up $E_{des}^0$. In molecular quantities, accounting for the number of molecules in the TS in the system, $N_{TS}$, and at constant volume yields

$$U_{TS}(0) = N_{TS}q_b^0 = \varepsilon_b^0 \tag{8}$$

and in molar quantities

$$U_{TS,m}(0) = E_b^0 . \tag{9}$$

In the literature, the desorption energy often includes the energy barrier (Kolasinski, 2012), so that the activation of desorption is, expressed in our notation here, as

$$E_{des,act}^0 = E_{des}^0 + E_b^0 . \tag{10}$$

For the remainder of the document, we treat the desorption energy and energy barrier separately. For non-activated adsorption or desorption processes, $E_b^0 = 0$, and all equations simplify accordingly.

**2.3 Relationship Between Partition Functions and Thermodynamic Quantities**

We use statistical thermodynamics to relate the microscopic properties to the matter's bulk properties. Via the partition function $Q$ we can express the thermodynamic functions $U$, $S$ (entropy), $H$ (enthalpy) and $G$ (Gibbs free energy) in the following way (Atkins and de Paula, 2006)

$$U - U(0) = -\left(\frac{\partial \ln Q}{\partial \beta}\right)_V \quad \text{with } \beta = \frac{1}{k_B T} \tag{11}$$

$$S = \frac{U-U(0)}{T} + k_B \ln Q \tag{12}$$

$$H - H(0) = -\left(\frac{\partial \ln Q}{\partial \beta}\right)_V + k_B T \mathcal{V} \left(\frac{\partial \ln Q}{\partial \mathcal{V}}\right)_T \tag{13}$$

$$G - G(0) = -kT \ln Q + k_B T \mathcal{V} \left(\frac{\partial \ln Q}{\partial \mathcal{V}}\right)_T . \tag{14}$$





$T$ and $\mathcal{V}$ are the system's temperature and volume, respectively, and $k_B$ is the Boltzmann constant. We first calculate the molecular quantities $U$, $H$, $G$, $S$, and then express them as molar quantities,

$U_m=U/n$, $H_m=H/n$, $G_m=G/n$, $S_m=S/n$, via $= \frac{N}{N_A}$, $R = N_A k$, and $q_m = \frac{q}{n}$,  (15)

where $q$ is the molecular partition function (Atkins and de Paula, 2006), $N$ is the number of molecules in the system, $n$ is the number of moles in the system, $N_A$ is the Avogadro number, and $R$ is the general gas constant.

As introduced below for the cases of 3D ideal gas, 2D ideal gas, 2D ideal lattice gas, and TS for desorption, we will apply the appropriate partition functions (see also Supplement S3). For the 3D and 2D ideal gases we will use the canonical partition

function, expressed for indistinguishable and independent molecules as $Q = q^N/N!$ (Atkins and de Paula, 2006). For the 2D ideal lattice gas, we will have to modify the canonical partition function to introduce adsorption sites (Hill, 1986).

**2.4 Concentration, Standard States of Gas Species and Adsorbates, and Activities**

The concentration of the 3D ideal gas in the gas phase is given by

$\mathcal{N}_g = \frac{N_g}{\mathcal{V}}$,  (16)

where $N_g$ is the number of gas molecules in the system and $\mathcal{V}$ is the volume of the system. Its standard concentration is

$\left(\frac{N_g}{\mathcal{V}}\right)^0 = \frac{n_g^0 \cdot N_A}{\mathcal{V}^0} = \frac{N_A}{\mathcal{V}_m^0}$,  (17)

where $n_g^0$ is the standard number of moles of the gas species (typically set equal to 1), $N_A$ is Avogadro's number, and $\mathcal{V}_m^0$ indicates the standard molar volume reflecting $n_g^0$. For $n_g^0 = 1$ mol, $\mathcal{V}_m^0 = 24.8$ L mol⁻¹ at 298 K and 1000 hPa. We define the gas phase activity, $a_g$, as the concentration in the gas phase, $\mathcal{N}_g$, divided by the standard concentration, $\left(N_g/\mathcal{V}\right)^0$:

$a_g = \frac{(N_g/\mathcal{V})}{(N_g/\mathcal{V})^0} = \frac{\mathcal{N}_g}{(N_g/\mathcal{V})^0} = \frac{\mathcal{N}_g}{(N_A/\mathcal{V}_m^0)}$.  (18)

The concentration for the adsorbate representing a 2D ideal gas, we define as

$\mathcal{N}_{ads} = \frac{N_{ads}}{\mathcal{A}}$,  (19)

where $N_{ads}$ is the number of gas molecules on the surface and $\mathcal{A}$ is the surface of the system. Its standard concentration is

$\left(\frac{N_{ads}}{\mathcal{A}}\right)^0 = \frac{n_{ads}^0 \cdot N_A}{\mathcal{A}^0} = \frac{N_A}{\mathcal{A}_m^0}$,  (20)

where $n_{ads}^0$ is the standard number of moles of adsorbate and $\mathcal{A}_m^0$ indicates the corresponding standard molar surface area. Several suggestions have been made for the surface concentrations (Donaldson et al., 2012;Ammann et al., 2013;Campbell et al., 2016;Kemball and Rideal, 1946;de Boer, 1968). Campbell et al. (2016) argue that choosing $(N_{ads}/\mathcal{A})^0 = e^{1/3}(N_A/\mathcal{V}_m^0)^{2/3}$, the adsorbate considered as a 2D ideal gas has an entropy of 2/3 of that of the gas species, i.e., $S_{ads}^0 = \frac{2}{3}S_g^0$ when just considering only the translational degrees of freedom (see below). Since a 2D ideal gas is a simple and

straightforward assumption especially for physisorption, this standard state has advantages. This standard surface





concentration corresponds to $(N_{ads}/\mathcal{A})^0 = 1.17 \times 10^{13}$ cm$^{-2}$ at 298 K at 1000 hPa. In comparison, the IUPAC Task Group on Atmospheric Chemical Kinetic Data Evaluation is using $(N_{ads}/\mathcal{A})^0 = 1.61 \times 10^{12}$ cm$^{-2}$ (Ammann et al., 2013;Crowley et al., 2013). We define the surface activity for the 2D ideal gas, $a_{ads,2D}$, as the concentration at the surface, $\mathcal{N}_{ads}$, divided by the standard surface concentration, $(N_{ads}/\mathcal{A})^0$:

$$a_{ads,2D} = \frac{(N_{ads}/\mathcal{A})}{(N_{ads}/\mathcal{A})^0} = \frac{(\mathcal{N}_{ads})}{(N_A/\mathcal{A}_m^0)} = \frac{(\mathcal{N}_{ads})}{(N_A/\mathcal{A}_m^0)} \cdot \tag{21}$$

The concentration for the molecule in the TS for desorption, we define as

$$\mathcal{N}_{TS} = \frac{N_{TS}}{\mathcal{A}}, \tag{22}$$

where $N_{TS}$ is the number of molecules in the TS and $\mathcal{A}$ is the area of the system. Its standard concentration is

$$\left(\frac{N_{TS}}{\mathcal{A}}\right)^0 = \frac{n_{TS}^0 \cdot N_A}{\mathcal{A}^0} = \frac{N_A}{\mathcal{A}_m^0}, \tag{23}$$

where $n_{TS}^0$ is the standard moles of TS molecules and $\mathcal{A}_m^0$ indicates the standard molar surface area. Since the TS does not interact with the surface, it is treated as a 2D ideal gas. Hence, we define the surface activity for the TS, $a_{TS}$, as the concentration of the TS, $\mathcal{N}_{TS}$, divided by the standard concentration of the TS, $(N_{TS}/\mathcal{A})^0$:

$$a_{TS} = \frac{(N_{TS}/\mathcal{A})}{(N_{TS}/\mathcal{A})^0} = \frac{(\mathcal{N}_{TS})}{(N_A/\mathcal{A}_m^0)} = \frac{(\mathcal{N}_{TS})}{(N_A/\mathcal{A}_m^0)} \cdot \tag{24}$$

For many applications, it has been common to normalize the surface concentration, $\mathcal{N}_{ads}$, to a maximum concentration

$$\mathcal{N}_{ads,max} = \frac{N_{ads,max}}{\mathcal{A}} \cdot \tag{25}$$

Then, the surface concentration can also be expressed as a coverage

$$\theta = \frac{\frac{N_{ads}}{\mathcal{A}}}{\frac{N_{ads,max}}{\mathcal{A}}} = \frac{\mathcal{N}_{ads}}{\mathcal{N}_{ads,max}} \tag{26}$$

with a corresponding standard surface coverage

$$\theta^0 = \left(\frac{N_{ads}}{\mathcal{A}}\right)^0 / \frac{N_{ads,max}}{\mathcal{A}} \cdot \tag{27}$$

Similar to the 3D ideal gas, also for the 2D ideal gas case, in principle, there is no limit to the surface concentration. To remain within physically reasonable bounds, all equations in conjunction with the 2D ideal gas model relate to conditions of small surface coverages only.

For the 2D ideal lattice gas case, the maximum number of equivalent but distinguishable sites is $N_{ads,max} = M$, which will be important for the statistical thermodynamic derivation (Supplement S2.3). A physically reasonable choice for $M$ is such that $\frac{M}{\mathcal{A}} = 10^{15}$ cm$^{-2}$. Then, the standard surface coverage is $\theta^0 = 0.0117$ at 298 K. We define the surface activity for the 2D ideal lattice gas, $a_{ads,latt}$, as the surface coverage, $\theta/(1-\theta)$, divided by the standard surface coverage, $\theta^0/(1-\theta^0)$:

$$a_{ads,latt} = \frac{(\theta/(1-\theta))}{(\theta^0/(1-\theta^0))}, \tag{28}$$

where $a_{ads,latt}$ does not depend linearly on surface coverage, $\theta$, and standard surface coverage, $\theta^0$. The reason for this, ultimately, lies in the canonical partition function describing equivalent but distinguishable adsorption sites (Supplement Eqs.





(40 and 41)). For example, from the derivation of the chemical potential of the adsorbed 2D ideal lattice gas (Supplement Eq. (56)), it can be clearly seen that Eq. (28) provides a self-consistent definition of the activity for this adsorbate model.

## 3. Thermodynamic Functions of the Desorption Equilibrium

We derive the desorption equilibrium constants for the 2D ideal gas and 2D ideal lattice gas in equilibrium with the gas phase considering the respective standard states and partition functions. See also general definitions for equilibrium constants

outlined in Supplement section S1. For both adsorbate models we also derive the change in enthalpy and entropy between the adsorbed and the gas molecule. The derivations in this section will demonstrate the importance of standard states when calculating the equilibrium constants for the desorption processes.

### 3.1 Desorption equilibrium for adsorbed 2D ideal gas

The adsorbed 2D ideal gas is characterized by molecules moving freely parallel to the surface with a constant binding energy

to the surface. In other words, the adsorbate vibrates in all directions but has only free translational motion in the horizontal plane. The thermodynamic desorption equilibrium constant is defined by the ratio of the activity in the gas phase ($a_g$) to that on the surface ($a_{ads}$),

$$K^0_{des,2D,g} = \frac{a_g}{a_{ads,2D}} = \frac{\frac{(N_g/\mathcal{V})}{(N_g/\mathcal{V})^0}}{\frac{(N_{ads}/\mathcal{A})}{(N_{ads}/\mathcal{A})^0}} = \frac{\frac{\mathcal{N}_g}{(N_g/\mathcal{V})^0}}{\frac{(\mathcal{N}_{ads})}{(N_A/\mathcal{A}^0_m)}} = \frac{\frac{\mathcal{N}_g}{(N_A/\mathcal{V}^0_m)}}{\frac{(\mathcal{N}_{ads})}{(N_A/\mathcal{A}^0_m)}} \,. \tag{29}$$

As indicated by the definition of the adsorbate surface activity, $a_{ads,2D}$, used in the definition of the equilibrium constant,

for the 2D ideal gas, the surface activity and thus, also the surface concentration, are linearly correlated with the gas phase activity and concentration (i.e., number density). This is often expressed with a constant ($K_{lin}$) directly relating gas phase number density with surface concentration (Crowley et al., 2010):

$$\mathcal{N}_{ads} = K_{lin}\mathcal{N}_g \,. \tag{30}$$

As mentioned above, no limitations by surface area or number of sites are convoluted in this equation. The relationship between

$K_{lin}$ and the equilibrium constant is:

$$K_{lin} = \frac{\mathcal{N}_{ads}}{\mathcal{N}_g} = \frac{(N_{ads}/\mathcal{A})^0}{K^0_{des,2D,g}(N_g/\mathcal{V})^0} \,. \tag{31}$$

The equilibrium constant, $K^0_{des,2D,g}$, is also related to the free energy change, $\Delta G^0_{des,2D,g,m}$. Since $\Delta G^0_{des,2D,g,m} = G^0_{g,m} - G^0_{ads,2D,m}$, we can associate the two free energies with the two partition functions for the two states, and thus express the equilibrium constant as (see Supplement Eqs. (123))

$$K^0_{des,2D,g} = e^{-\Delta G^0_{des,2D,g,m}/RT} = \frac{q^0_{g,m}}{q^0_{ads,2D,m}} e^{-\frac{E^0_{des}}{RT}} \,. \tag{32}$$





The two partition functions, $q_{g,m}^0$ and $q_{ads,2D,m}^0$ are evaluated using the standard molar volume and area, respectively. Typical values for standard partition functions are given in Table S1. The desorption or activation energy at the molecule's zero-point energy reflects the energy to elevate the adsorbed molecule from the lowest vibrational state to the lowest vibrational state of the activated complex, i.e., the molecular state from which the adsorbate can directly desorb into the gas phase. In other words, $E_{des}$ corresponds to the depth of the potential well (per mole). It has a positive value as defined above (Eq. (5)). When applying the standard adsorption enthalpy and entropy in Eq. (32) (via $\Delta G_{des,2D,g,m}^0$), those have to be based on the same standard concentrations as given in Eqs. (17) and (20), to result in the same $K_{des,2D,g}^0$. Applying the expressions for the partition functions (see Supplement Eqs. (92) and (99)):

$$K_{des,2D,g}^0 = \frac{\mathcal{V}_m^0(2\pi mk_BT/h^2)^{3/2}}{\mathcal{A}_m^0(2\pi mk_BT/h^2)}e^{-\frac{E_{des}^0}{RT}} = \frac{\mathcal{V}_m^0}{\mathcal{A}_m^0}(2\pi mk_BT/h^2)^{1/2}e^{-\frac{E_{des}^0}{RT}} = \frac{(N_{ads}/\mathcal{A})^0}{(N_g/\mathcal{V})^0}(2\pi mk_BT/h^2)^{1/2}e^{-\frac{E_{des}^0}{RT}} \tag{33}$$

and thus, it follows

$$K_{lin} = (2\pi mk_BT/h^2)^{-1/2}e^{\frac{E_{des}^0}{RT}} . \tag{34}$$

Hence, $K_{lin}$ can be readily calculated if vibrations are not considered. For a molecule at 298 K with molecular weight of 60 g mol$^{-1}$ and $E_{des}^0$ = 45 kJ mol$^{-1}$, $K_{lin}$ is about 0.1 cm, a typical value also for many species (Crowley et al., 2010).

The standard free energy change (and the equilibrium constant) is also related to the adsorption entropy and enthalpy via (Supplement Eq. (119ff.))

$$-RT\ln\left(K_{des,2D,g}^0\right) = \Delta G_{des,2D,g,m}^0 = \Delta H_{des,2D,g,m}^0 - T\Delta S_{des,2D,g,m}^0 = E_{des}^0 - RT\ln\left[\frac{\left(\frac{q_{g,m}^0}{N_A}\right)}{\left(\frac{q_{ads,2D,m}^0}{N_A}\right)}\right] . \tag{35}$$

As shown in the Supplement (Eqs. (12), (28), and (120))

$$\Delta H_{des,2D,g,m}^0 = H_{g,m} - H_{ads,2D,m} = \frac{5}{2}RT - \frac{4}{2}RT + E_{des}^0 = \frac{1}{2}RT + E_{des}^0 . \tag{36}$$

The enthalpy difference is due to the change in translational degrees of freedom between the 3D and 2D ideal gases, and the binding energy of the 2D ideal gas on the surface.

As derived in the Supplement (Eq. (16)) from statistical thermodynamics, the entropy in the gas phase is given by the Sackur-Tetrode equation (Campbell et al., 2016;Atkins and de Paula, 2006;Hill, 1986) as

$$S_{g,m}^0 = R\ln\left(\frac{e^{5/2}q_{g,m}^0}{N_A}\right) = R\ln\left(\mathcal{V}_m^0(2\pi mk_BT/h^2)^{3/2}e^{5/2}\right) = R\ln\left(\frac{(2\pi mk_BT/h^2)^{3/2}e^{5/2}}{(N_g/\mathcal{V})^0}\right), \tag{37}$$

while the entropy on the surface is (Supplement Eq. (33)):

$$S_{ads,2D,m}^0 = R\ln\left(\frac{e^2 q_{ads,2D,m}^0}{N_A}\right) = R\ln\left(e^2\mathcal{A}_m^0(2\pi mk_BT/h^2)^{2/2}\right) = R\ln\left(\frac{e^2(2\pi mk_BT/h^2)^{2/2}}{(N_{ads}/\mathcal{A})^0}\right) . \tag{38}$$

As already mentioned above, following Campbell et al. (2016), because the standard state is chosen as $(N_{ads}/\mathcal{A})^0 = e^{1/3}(N_A/\mathcal{V}_m^0)^{2/3}$, the entropy on the surface is 2/3 of that in the gas phase (Eq. (37)), because



$$S_{ads,2D,m}^0 = R \ln\left(\frac{(2\pi m k_B T/h^2)e^2}{(N_{ads}/\mathcal{A})^0}\right) = R \ln\left(\frac{(2\pi m k_B T/h^2)e^{5/3}}{(N_A/\mathcal{V}_m^0)^{2/3}}\right) = \frac{2}{3}R \ln\left(\frac{(2\pi m k_B T/h^2)^{3/2}e^{5/2}}{(N_g/\mathcal{V})^0}\right) = \frac{2}{3}S_{g,m}^0 \ . \tag{39}$$

From this follows (Supplement Eq. (121))

$$\Delta S_{des,2D,g,m}^0 = S_{g,m}^0 - S_{ads,2D,m}^0 = R \ln\left(\frac{e^{5/2}q_{g,m}^0}{N_A}\right) - R \ln\left(\frac{e^2 q_{ads,2D,m}^0}{N_A}\right) = R \ln\left(\frac{e^{5/2}q_{g,m}^0 N_A}{N_A e^2 q_{ads,2D,m}^0}\right) = R \ln\left(\frac{e^{1/2}q_{g,m}^0}{q_{ads,2D,m}^0}\right) = \frac{1}{2}R +$$

$$R \ln\left(\frac{q_{g,m}^0}{q_{ads,2D,m}^0}\right), \tag{40}$$

Using $\Delta H_{des,2D,g,m}^0$ (Eq. (36)) and $\Delta S_{des,2D,g,m}^0$ (Eq. (40)) together in the second part of Eq. (35) results in the last expression of Eq. (35). Thus, the expressions for the thermodynamic functions are all consistent with each other.

Substituting the definition of $\theta$ (Eq. (27)) into the equation for the adsorption entropy (Eq. (39)) leads to:

$$S_{ads,2D,m}^0 = R \ln\left(\frac{(2\pi m k_B T/h^2)e}{(N_{ads,max}/\mathcal{A})}\right) + R \ln(e/\theta^0) = S_{trans,2D} + S_{cov} \ . \tag{41}$$

Thus, the adsorption entropy can be considered the sum of a translational term, $S_{trans,2D}$, and a coverage-dependent term, $S_{cov}$. For $\theta^0 = 0.012$, $S_{cov} = 5.42R$. At room temperature, $S_{trans,2D}$ varies around $23R$.

### 3.2 Desorption equilibrium for adsorbed 2D ideal lattice gas

In contrast to the adsorbate being equivalent to a 2D ideal gas, where molecules freely diffuse parallel across the surface, the adsorbed molecule could also randomly populate a fixed number of adsorption sites, where the adsorbates have only vibrational degrees of freedom in three directions. This adsorption model is generally referred to as Langmuir adsorption (Langmuir, 1915, 1916, 1932). It is worthwhile noting that this concept holds for solid and liquid surfaces as long as the number of adsorption sites is given by $M$. In other words, it is not necessary to know how the $M$ adsorption sites are distributed over the surface and time. The corresponding picture would be to treat the adsorbate as a 2D ideal lattice gas (Campbell et al., 2016). The activity is then given by $\frac{(\theta/(1-\theta))}{(\theta^0/(1-\theta^0))}$ (Supplement S2.3). In analogy to Eq. (29), the equilibrium constant is formulated as the ratio of activities

$$K_{des,latt,g}^0 = \frac{a_g}{a_{ads,latt}} = \frac{\frac{(N_g/\mathcal{V})}{(N_g/\mathcal{V})^0}}{\frac{(\theta/(1-\theta))}{(\theta^0/(1-\theta^0))}} = \frac{\frac{\mathcal{N}_g}{(N_g/\mathcal{V})^0}}{\frac{(\theta/(1-\theta))}{(\theta^0/(1-\theta^0))}} = \frac{\frac{\mathcal{N}_g}{(N_A/\mathcal{V}_m^0)}}{\frac{(\theta/(1-\theta))}{(\theta^0/(1-\theta^0))}} \ . \tag{42}$$

In the traditional formulation of Langmuir adsorption, the coverage is related to the gas phase concentration via

$$\theta = \frac{K_{Lang}\mathcal{N}_g}{(1+K_{Lang}\mathcal{N}_g)}, \tag{43}$$

where $K_{Lang}$ is the Langmuir adsorption constant. From this, we can derive

$$(\theta/(1-\theta)) = K_{Lang}\mathcal{N}_g \ . \tag{44}$$

This equation clearly demonstrates the usefulness of the definition of the adsorbate surface activity. Thus, for the relationship between the $K_{Lang}$ and $K_{ads,latt}^0$, we obtain





$$K_{Lang} = \frac{(\theta^0/(1-\theta^0))}{K^0_{des,latt,g}(N_g/\mathcal{V})^0} \ . \tag{45}$$

This relationship demonstrates that the functional form of the dependence of the surface coverage with pressure or concentration in the gas phase is the same for both definitions of the equilibrium constants (apart from the inverse formulation of the equilibrium constant as the ratio of gas-to-surface concentrations (Eq. (42)) versus surface-to-gas concentrations). However, only $K^0_{des,latt,g}$ can be related to the free energy change directly. Also in this case, the standard free energy change, $\Delta G^0_{des,latt,g,m}$, embodied in $K^0_{des,latt,g}$, can be related to the partition functions describing the molecules in the gas phase and 330 adsorbed phases as (see Supplement Eq. (137))

$$K^0_{des,latt,g} = \frac{\left(\frac{q^0_{g,m}}{N_A}\right)}{q_{ads,latt}\frac{(1-\theta^0)}{\theta^0}} e^{-\frac{E^0_{des}}{RT}} \ . \tag{46}$$

When inserting the expressions for the standard molar partition functions for the translational motions (see Supplement Eq. (92)):

$$K^0_{des,latt,g} = \frac{\left(\frac{\mathcal{V}^0_m(2\pi mk_BT/h^2)^{3/2}}{N_A}\right)}{q_{ads,latt}\frac{(1-\theta^0)}{\theta^0}} e^{-\frac{E^0_{des}}{RT}} = \frac{(2\pi mk_BT/h^2)^{3/2}}{(N_g/\mathcal{V})^0 q_{ads,latt}\frac{(1-\theta^0)}{\theta^0}} e^{-\frac{E^0_{des}}{RT}} \ . \tag{47}$$

$K_{Lang}$ can now be expressed as

$$K_{Lang} = \frac{q_{ads,latt}e^{\frac{E^0_{des}}{RT}}}{(2\pi mk_BT/h^2)^{3/2}} \ . \tag{48}$$

Hence, $K_{Lang}$ can be readily calculated. For a molecule at 273 K with molecular weight of 48 g mol[-1], and vibration frequency of about 10[13] s[-1], and $E^0_{des} = 70$ kJ mol[-1], $K_{Lang}$ is about 10[-13] cm[3], representing a typical value (Ammann et al., 2013).

Since $K^0_{des,latt,g}$ is also related to the enthalpy and entropy of adsorption, we can write

$$-RT\ln\left(K^0_{des,latt,g}\right) = \Delta G^0_{des,latt,g,m} = \Delta H^0_{des,latt,g,m} - T\Delta S^0_{des,latt,g,m} = E^0_{des} - RT\ln\frac{\left(\frac{q^0_{g,m}}{N_A}\right)}{q_{ads,latt}\frac{(1-\theta^0)}{\theta^0}} \ . \tag{49}$$

In variation to the case of the 2D ideal gas, and neglecting vibrations, $U_{ads,latt,m} = -E^0_{des}$ (Supplement Eq. (45)), due to the absence of translational motion (while in the gas phase, $U_{g,m} = \frac{3}{2}RT$, or for the 2D ideal gas, $U_{ads,2D,m} = RT - E^0_{des}$). Also, as shown in the Supplement Eq. (49) (neglecting contribution of vibrations in gas and adsorbed phase), we obtain

$$H^0_{ads,latt,m} = -E^0_{des} - RT\frac{\ln(1-\theta^0)}{\theta^0} . \tag{50}$$

Overall, for the change in enthalpy between gas and adsorbed states (see also Supplement Eq. (133)), we obtain

$$\Delta H^0_{des,latt,g,m} = H^0_{g,m} - H^0_{ads,latt,m} = \frac{5}{2}RT + E^0_{des} + RT\frac{\ln(1-\theta^0)}{\theta^0} \ . \tag{51}$$

We can now obtain the relationship between the desorption energy and the adsorption enthalpy as

$$E^0_{des} = \Delta H^0_{des,latt,g,m}(T) - \frac{5}{2}RT - RT\frac{\ln(1-\theta^0)}{\theta^0} \ . \tag{52}$$





Thus, in the case of the 2D ideal lattice gas, the relationship between the desorption energy and the enthalpy contains the
standard surface coverage explicitly.

For the entropy of the adsorbed 2D ideal lattice gas (Supplement Eqs. (54) and (103)), we can write

$$S^0_{ads,latt,m} = R\left(\ln q_{ads,latt} - \beta\left(\frac{\partial \ln q_{ads,latt}}{\partial \beta}\right)\right) + R\left(\ln\left(\frac{(1-\theta^0)}{\theta^0}\right) - \frac{\ln(1-\theta^0)}{\theta^0}\right) = S^0_{ads,latt,vib} + S^0_{ads,latt,config} \, . \tag{53}$$

The adsorption entropy has a contribution for the vibrations in three dimensions at the site, $S_{ads,latt,vib}$, (related to $q_{ads,latt}$,
Supplement Eq. (38)) and a configurational contribution, $S_{ads,latt,config}$. Using the above standard state of $\theta^0 = 0.012$ leads to

$$S^0_{ads,latt,m} = S^0_{ads,latt,vib} + 5.42R \, . \tag{54}$$

Typical values for $S_{vib}$ for 3 dimensions at room temperature, assuming a vibration frequency of $10^{14}$ s$^{-1}$, are around $4.90R$
(Campbell et al., 2016;McQuarrie, 2000;Atkins and de Paula, 2006). Note that also another choice of $\theta^0$ has been used, i.e.,
$\theta^0 = 0.5$, because then, the $\theta^0/(1-\theta^0)$ is unity. Consequently, this leads to a different numerical value for the standard
adsorption entropy ($S_{ads,latt,config} = 1.39R$). The choice of the standard state adopted here and suggested by Campbell et al.

(2016) has the advantage that the standard adsorbate coverage is low and the coverage dependent contributions $S_{ads,latt,config}$
for the 2D ideal lattice gas and $S_{cov}$ for the 2D ideal gas have nearly the same value (5.417 and 5.423, respectively).

For the change in entropy upon desorption, we can derive (Supplement Eq. (135))

$$\Delta S^0_{des,latt,g,m} = S^0_{g,m} - S^0_{ads,latt,m} = R\ln\left(\frac{e^{5/2} q^0_{g,m}}{N_A}\right) - R\left(\ln q_{ads,latt} - \beta\left(\frac{\partial \ln q_{ads,latt}}{\partial \beta}\right)\right) - R\left(\ln\left(\frac{(1-\theta^0)}{\theta^0}\right) - \frac{\ln(1-\theta^0)}{\theta^0}\right) \, .$$

$$\tag{55}$$

## 365    3.3 Adsorbate model comparison of surface concentration, activity, and coverage

We can now use the results in section 3.1 and 3.2 to evaluate the equilibrium conditions between gas phase and surface
concentrations and activities and respective coverages for the 2D ideal gas and 2D ideal lattice gas, presented in Figs. 2-4. The
thermodynamic quantities to reproduce these figures are given in Table S1. Figure 2 illustrates the behavior of the adsorption
equilibria for the 2D ideal gas and the 2D ideal lattice gas cases in terms of surface concentration versus gas phase
concentration. As intuitively clear from the defining equations, for the 2D ideal gas case, the surface concentration increases
linearly with gas phase concentration without a limitation, thus, increasing beyond a monolayer coverage, here assumed as
$10^{19}$ m$^{-2}$. In turn, for the 2D ideal lattice gas case, the initially linear increase is followed by the well-established adsorption
saturation due to the limitation by the number of available sites on the surface, known as Langmuir adsorption. Note that we
purposely chose a larger desorption energy for this case, leading to the higher initial slope. Assuming the same desorption
energy for both cases, the initial slopes would be the same for both adsorption models. As shown in Fig. 3, when normalizing
the surface concentration to the maximum number of adsorption sites to obtain the coverage, the picture remains the same.

In contrast to Figs. 2 and 3, when considered in terms of activities, both adsorbate models exhibit a linear relationship
between the surface activity and the gas phase activity as shown in Fig. 4. While trivial for the 2D ideal gas case, for the 2D





ideal lattice gas case, this is related to the definition of the activity as being proportional to $\theta/(1-\theta)$. Note that the gas phase

activity range in Fig. 4 covers the same gas phase concentration range as in Figs. 2 and 3. Also note that the numerical values

for the activities are completely different for the two cases. For example, for the 2D ideal gas case, at values of $\theta$ of 0.5 and

0.8, $a_{ads,2D}$ is 42.8 and 68.4, respectively, while for the 2D ideal lattice gas at the same coverages, $a_{ads,latt}$ is 85.9 and 336.8,

respectively. On the one hand, the different slopes of surface activity as a function of gas phase activity are related to the

normalization to the two different standard states. On the other hand, when considered as a function of $\theta$, the relationship

between the two surface activities is highly non-linear due to the diverging nature of the $\theta/(1-\theta)$ term for high $\theta$.

## 4. Derivation of the Desorption Rate and Pre-Exponential Factor $A$

Above we have outlined the determination of the equilibrium constant $K_{des}^0$ and the importance to consider the standard

concentrations. In this section we will derive the desorption rate and its pre-exponential factor $A$ from TS theory, which starts

from the free energy change between the adsorbate and the TS. This exercise will demonstrate the necessity of knowing the

standard state of the entropic contribution or the standard concentrations of the TS and adsorbate for the correct derivation of

$A$. As we will show below, the pre-exponential factor $A$ in the desorption rate coefficient, $k_{des}$, includes the entropic change

between the adsorbed and TS of the desorbing molecule. If we like to calculate $A$, the standard desorption entropy has to be

based on the same standard concentrations as for the definition of the activity. Again, the same activity definitions have to be

applied to calculate actual desorption rates. We will see that without knowledge of the chosen standard state of the entropy or

standard concentrations of TS and adsorbate species, $A$ cannot be accurately derived. Furthermore, we examine two cases of

adsorbate where we first treat the adsorbate as a 2D ideal gas and secondly as a 2D ideal lattice gas. The TS is treated as a 2D

ideal gas in both cases. This section follows the derivations outlined in Campbell et al. (2016). Detailed derivations are given

in the Supplement.

In general, the desorption rate can be expressed as

$$\frac{R_{des}}{\mathcal{A}} = -\frac{d\mathcal{N}_{ads}}{dt} = -k_{des}\mathcal{N}_{ads} \, , \tag{56}$$

where $k_{des}$ represents the first-order rate coefficient for desorption (in units s$^{-1}$), describing the rate of change of surface

concentration. As is evident from the definitions of activity above, the surface concentration is not necessarily proportional to

the surface activity. We therefore introduce a separate rate expression and corresponding desorption rate coefficient acting on

surface activities, $R_{des}^a$ and $k_{des}^a$, respectively, as

$$R_{des}^a = -\frac{da_{ads}}{dt} = -k_{des}^a a_{ads} \, . \tag{57}$$



## 4.1 Desorption of a 2D ideal gas

According to conventional transition state theory (CTST) (Kolasinski, 2012), $E_{des}$ is the activation energy necessary to elevate an adsorbed species from the lowest vibrational state to the lowest vibrational state of the activated complex, i.e., the molecular state from which the adsorbate can directly desorb into the gas phase. Note that desorption is always considered an activated process, thus, also including the case of desorption of a physisorbed molecule, and irrespective of whether an energy barrier is considered or not. In CTST, rates are derived from assuming equilibrium between the adsorbed state and the TS, which is the reason for discussing the overall adsorption/desorption equilibrium in detail above. The TS for desorption is assumed to exist at some fixed distance from the surface without any interactions with it. In principle, the TS resembles a 2D ideal gas, but as discussed further below and in the Supplement S3.4, CTST assumes molecules in the TS exhibit translational motion along the reaction coordinate, which for the case of desorption is orthogonally away from the surface. The associated equilibrium constant is related to the free energy change between the adsorbed state and the TS, each expressed with the corresponding standard molar partition function, $q_{ads,2D,m}^0$, and $q_{TS,m}^0$ (Supplement sections S1, S2, and S4, Eqs. (3), (22), (60), (117) and (152))

$$K_{des,2D,TS}^0 = \frac{\left(\frac{q_{TS,m}^0}{N_A}\right)}{\left(\frac{q_{ads,2D,m}^0}{N_A}\right)} e^{-\frac{\left(E_{des}^0+E_b^0\right)}{RT}} = \frac{q_{TS,m}^0}{q_{ads,2D,m}^0} e^{-\frac{\left(E_{des}^0+E_b^0\right)}{RT}} \ . \tag{58}$$

The equilibrium constant is also related to the ratio of activities

$$K_{des,2D,TS}^0 = e^{-\Delta G_{des,2D,TS,m}^0/RT} = \frac{\frac{(N_{TS}/\mathcal{A})}{(N_{TS}/\mathcal{A})^0}}{\frac{(N_{ads}/\mathcal{A})}{(N_{ads}/\mathcal{A})^0}} = \frac{\frac{\mathcal{N}_{TS}}{(N_{TS}/\mathcal{A})^0}}{\frac{\mathcal{N}_{ads}}{(N_{ads}/\mathcal{A})^0}} = \frac{a_{TS}}{a_{ads}} = \frac{q_{TS,m}^0}{q_{ads,2D,m}^0} e^{-\frac{\left(E_{des}^0+E_b^0\right)}{RT}} \ . \tag{59}$$

As discussed above, the entropy values depend strongly on the configuration (i.e., degrees of freedom) of the species in the adsorbed state and the TS.

Within this CTST approach, the desorption rate can be obtained by assuming that the TS has a finite width $d$ across which the molecule moves with its mean thermal velocity in the direction orthogonal to the surface

$$\frac{R_{des,2D}}{\mathcal{A}} = \kappa \left(\frac{N_{TS}}{\mathcal{A}}\right) \frac{(k_B T/2\pi m)^{1/2}}{d} \ , \tag{60}$$

where $\kappa$ is a transmission coefficient defining the probability with which an activated complex proceeds to desorption (Kolasinski, 2012). The partition function for the translational motion of the transition state in the direction of desorption is

$$q_{TS,des} = (2\pi m k_B T/h^2)^{1/2} d \ . \tag{61}$$

Solving this for $d$ and inserting into Eq. (60) allows to express the desorption rate as a function of this partition function:

$$\frac{R_{des,2D}}{\mathcal{A}} = \kappa \left(\frac{k_B T}{h}\right) \left(\frac{N_{TS}}{\mathcal{A}}\right) \frac{1}{q_{TS,des}} \ . \tag{62}$$

The surface concentration of the TS can be derived from the equilibrium (Eq. (59))

$$\mathcal{N}_{TS} = \frac{N_{TS}}{\mathcal{A}} = \frac{q_{TS,m}^0}{q_{ads,2D,m}^0} \frac{(N_{TS}/\mathcal{A})^0}{(N_{ads}/\mathcal{A})^0} e^{-\frac{\left(E_{des}^0+E_b^0\right)}{RT}} \mathcal{N}_{ads} \ . \tag{63}$$





Inserting Eq. (63) into Eq. (62) leads to

$$\frac{R_{des,2D}}{\mathcal{A}} = \kappa \left(\frac{k_BT}{h}\right)\left(\frac{1}{q_{TS,des}}\frac{q_{TS,m}^0}{q_{ads,2D,m}^0}\right)\frac{(N_{TS}/\mathcal{A})^0}{(N_{ads}/\mathcal{A})^0}e^{-\frac{\left(E_{des}^0+E_b^0\right)}{RT}}\mathcal{N}_{ads} . \tag{64}$$

When considering surface activities, by dividing by the standard surface concentration we obtain

$$R_{des,2D}^a = \frac{\frac{R_{des,2D}}{\mathcal{A}}}{(N_{ads}/\mathcal{A})^0} = \kappa \left(\frac{k_BT}{h}\right)\left(\frac{1}{q_{TS,des}}\frac{q_{TS,m}^0}{q_{ads,2D,m}^0}\right)\frac{(N_{TS}/\mathcal{A})^0}{(N_{ads}/\mathcal{A})^0}e^{-\frac{\left(E_{des}^0+E_b^0\right)}{RT}}a_{ads,2D} . \tag{65}$$

As further discussed in Supplemental Section 3.4, the activation process can be conceptually envisioned by bringing the molecules in the 2D ideal gas from the zero-point energy to the actual energy level that allows for the formation of the TS.

Thus, activation does not include the energy of the motion along the desorption coordinate, and as such is less than the energy associated with the TS. When defining $\Delta G_{des,2D,act,m}^0$ of desorption as $\Delta G_{des,2D,TS,m}^0$ (see Supplement Eq. (145) and (146)) minus the TS's free energy associated with the motion along the desorption coordinate, expressed by its molecular partition function, $q_{TS,des}$, we obtain

$$e^{-\Delta G_{des,2D,act,m}^0/RT} = \frac{e^{-\Delta G_{des,2D,TS,m}^0/RT}}{q_{TS,des}} = \frac{1}{q_{TS,des}}\frac{q_{TS,m}^0}{q_{ads,2D,m}^0}e^{-\frac{\left(E_{des}^0+E_b^0\right)}{RT}} . \tag{66}$$

With this definition of $\Delta G_{des,2D,act,m}^0$, we can express the desorption rate as

$$\frac{R_{des,2D}}{\mathcal{A}} = \kappa \left(\frac{k_BT}{h}\right)\left(\frac{1}{q_{TS,des}}\frac{q_{TS,m}^0}{q_{ads,2D,m}^0}\right)\frac{(N_{TS}/\mathcal{A})^0}{(N_{ads}/\mathcal{A})^0}e^{-\frac{\left(E_{des}^0+E_b^0\right)}{RT}}\mathcal{N}_{ads} = \kappa \left(\frac{k_BT}{h}\right)e^{-\Delta G_{des,2D,act,m}^0/RT}\frac{(N_{TS}/\mathcal{A})^0}{(N_{ads}/\mathcal{A})^0}\mathcal{N}_{ads} \tag{67}$$

and obtain for the activity-based desorption rate

$$R_{des,2D}^a = \kappa \left(\frac{k_BT}{h}\right)\left(\frac{1}{q_{TS,des}}\frac{q_{TS,m}^0}{q_{ads,2D,m}^0}\right)\frac{(N_{TS}/\mathcal{A})^0}{(N_{ads}/\mathcal{A})^0}e^{-\frac{\left(E_{des}^0+E_b^0\right)}{RT}}a_{ads,2D} = \kappa \left(\frac{k_BT}{h}\right)e^{-\Delta G_{des,2D,act,m}^0/RT}\frac{(N_{TS}/\mathcal{A})^0}{(N_{ads}/\mathcal{A})^0}a_{ads,2D} . \tag{68}$$

Thus, we can derive the desorption rate coefficient as

$$k_{des,2D} = k_{des,2D}^a = \kappa \left(\frac{k_BT}{h}\right)\frac{(N_{TS}/\mathcal{A})^0}{(N_{ads}/\mathcal{A})^0}e^{-\Delta G_{des,2D,act,m}^0/RT} = \kappa \left(\frac{k_BT}{h}\right)e^{-\Delta G_{des,2D,act,m}^0/RT} , \tag{69}$$

where we assume $\frac{(N_{TS}/\mathcal{A})^0}{(N_{ads}/\mathcal{A})^0} = 1$. Equation (69) is consistent with Eq. (4.4.24) in Kolasinski (2012), since the standard concentrations are the same for the TS and the adsorbed state in this case.

Following Campbell et al. (2016) defining $q_{TS}^{0'}/\mathcal{A}_m^0$ as the partition function for the TS after omitting motion in the direction of the reaction coordinate, this leaves the partition function for a 2D ideal gas (Supplement Eq. (117)):

$$\left(\frac{q_{TS,m}^0}{q_{TS,des}}\right) = q_{TS,m}^{0'} = q_{TS,2D,m}^0 = \mathcal{A}_m^0(2\pi mk_BT/h^2)^{2/2} . \tag{70}$$

Using Eq. (70) in Eq. (71), we obtain

$$\frac{R_{des}}{\mathcal{A}} = \kappa \left(\frac{k_BT}{h}\right)\left(\frac{q_{TS,m}^{0'}}{q_{ads,2D,m}^0}\right)\frac{(N_{TS}/\mathcal{A})^0}{(N_{ads}/\mathcal{A})^0}e^{-\frac{\left(E_{des}^0+E_b^0\right)}{RT}}\mathcal{N}_{ads} \tag{71}$$

and





$$R^a_{des,2D} = \kappa \left(\frac{k_B T}{h}\right)\left(\frac{q^{0\prime}_{TS,m}}{q^0_{ads,2D,m}}\right)\frac{(N_{TS}/\mathcal{A})^0}{(N_{ads}/\mathcal{A})^0} e^{-\frac{\left(E^0_{des}+E^0_b\right)}{RT}} a_{ads,2D} \,.$$ (72)

Identifying Eq. (71) with Eq. (56) yields

$$k_{des,2D} = k^a_{des,2D} = \kappa \left(\frac{k_B T}{h}\right)\left(\frac{q^{0\prime}_{TS}}{q^0_{ads,2D,m}}\right)\frac{(N_{TS}/\mathcal{A})^0}{(N_{ads}/\mathcal{A})^0} e^{-\frac{\left(E^0_{des}+E^0_b\right)}{RT}} \,.$$ (73)

We can convert the standard molar partition functions back to the molecular ones. For that, we consider that $\left(\frac{N_{ads}}{\mathcal{A}}\right)^0 = \frac{n_{ads} \cdot N_A}{\mathcal{A}^0} = \frac{N_A}{\mathcal{A}^0_m}$ and analogously for the TS, then we obtain

$$\frac{1}{q_{TS,des}}\frac{q^0_{TS,m}}{q^0_{ads,2D,m}}\frac{(N_{TS}/\mathcal{A})^0}{(N_{ads}/\mathcal{A})^0} = \frac{q^{0\prime}_{TS,m}}{q^0_{ads,2D,m}}\frac{(N_{TS}/\mathcal{A})^0}{(N_{ads}/\mathcal{A})^0} = \frac{\frac{q'_{TS}}{\mathcal{A}}}{\frac{q_{ads,2D}}{\mathcal{A}}} = \frac{q'_{TS}}{q_{ads,2D}} \,.$$ (74)

This yields

$$k_{des,2D} = k^a_{des,2D} = \kappa \left(\frac{k_B T}{h}\right)\left(\frac{q'_{TS}}{q_{ads,2D}}\right) e^{-\frac{\left(E^0_{des}+E^0_b\right)}{RT}} \,.$$ (75)

Hence, we have an expression for $k_{des,2D}$ based on thermodynamic quantities (Eq. (69)) and on molecular properties (Eq. (75)). The latter is consistent with Eq. (4.4.20) given by (Kolasinski, 2012):

$$k_{des,2D} = k^a_{des,2D} = \kappa \frac{k_B T}{h}\frac{q_{\ddagger}}{q_{ads}} e^{-\frac{\left(E^0_{des}+E^0_b\right)}{RT}} \,,$$ (76)

where $q_{\ddagger}$ represents the partition function of the TS, for which, in the explanation of Kolasinski, 'the loose vibration in the direction of desorption has been factored out' and can be identified with $q'_{TS}$. Note that factoring out a 'loose' vibration has the same effect on $q_{TS}$ as assigning the TS a translation over the length $d$, as discussed above and in other text books (Hill, 1986;Pilling and Seakins, 1996). As outlined above, in the literature, the desorption energy often includes the energy barrier (Kolasinski, 2012), i.e., $E^0_{des,act} = E^0_{des}+E^0_b$. For non-activated adsorption or desorption, $E^0_b$ can be set to zero.

475        The above derivations include the definition of the free energy of desorption (i.e., the free energy change between the adsorbed and the TS) and, thus, allow us to evaluate the pre-exponential factor $A$. We first formulate $k_{des}$ using the definition of $\Delta G^0_{des,2D,act,m}$ (Eq. 66), equate with the expression in Eq. (75) and apply the relationship $\Delta H^0_{des,2D,act,m} = E^0_b + E^0_{des}$ (Supplement Eq. (148))

$$k_{des,2D} = k^a_{des,2D} = \kappa \left(\frac{k_B T}{h}\right)\frac{(N_{TS}/\mathcal{A})^0}{(N_{ads}/\mathcal{A})^0} e^{-\Delta G^0_{des,2D,act,m}/RT} = \kappa \left(\frac{k_B T}{h}\right)\left(\frac{q'_{TS}}{q_{ads,2D}}\right) e^{-\frac{\left(E^0_{des}+E^0_b\right)}{RT}} \equiv$$

$$\kappa \left(\frac{k_B T}{h}\right)\frac{(N_{TS}/\mathcal{A})^0}{(N_{ads}/\mathcal{A})^0} e^{\Delta S^0_{des,2D,act,m}/R} e^{-\Delta H^0_{des,2D,act,m}/RT} = \kappa \left(\frac{k_B T}{h}\right)\left(\frac{q'_{TS}}{q_{ads,2D}}\right) e^{-\frac{\left(E^0_{des}+E^0_b\right)}{RT}} \equiv$$

$$\kappa \left(\frac{k_B T}{h}\right)\frac{(N_{TS}/\mathcal{A})^0}{(N_{ads}/\mathcal{A})^0} e^{\Delta S^0_{des,2D,act,m}/R} e^{-\frac{\left(E^0_{des}+E^0_b\right)}{RT}} = \kappa \left(\frac{k_B T}{h}\right)\left(\frac{q'_{TS}}{q_{ads,2D}}\right) e^{-\frac{\left(E^0_{des}+E^0_b\right)}{RT}} \,.$$ (77)

With this, we can define the pre-exponential factor $A$ as




$$A_{des,2D} = \kappa \left(\frac{k_B T}{h}\right)\left(\frac{q'_{TS}}{q_{ads,2D}}\right) = \kappa \left(\frac{k_B T}{h}\right)\frac{(N_{TS}/\mathcal{A})^0}{(N_{ads}/\mathcal{A})^0} e^{\Delta S^0_{des,2D,act,m}/R} = \kappa \left(\frac{k_B T}{h}\right)\frac{(N_{TS}/\mathcal{A})^0}{(N_{ads}/\mathcal{A})^0} e^{\frac{S^0_{act,m}-S^0_{ads,2D,m}}{R}} . \tag{78}$$

Equation (78) demonstrates the relevance of knowing the standard state. The first expression on the right-hand side, the

formulation in terms of the molecular partition functions ($q'_{TS}$, $q_{ads,2D}$), indicates that the value of $A_{des,2D}$ is directly linked to the assumptions of the adsorbate model as a basis for the calculation of the partition functions. In contrast, when $A_{des,2D}$ is obtained from the entropy of activation ($\Delta S^0_{des,2D,act,m}$), the Arrhenius term needs to be corrected by the ratio of the standard states, $\frac{(N_{TS}/\mathcal{A})^0}{(N_{ads}/\mathcal{A})^0}$.

Above derivations (Eq. (77)) now allow for the interpretation of $A_{des,2D}$. Let us assume $\kappa \approx 1$. Also recall that if both

adsorbed and TS are 2D ideal gases and if we neglect vibrations, $\left(\frac{q'_{TS}}{q_{ads,2D}}\right) = \frac{(2\pi m k_B T/h^2)}{(2\pi m k_B T/h^2)} = 1$, which is equivalent to having no significant change in entropy, i.e., $\Delta S^0_{des,2D,act,m} = 0$. This leads to the commonly applied value of $A_{des,2D} \approx \frac{k_B T}{h} \approx 6 \times 10^{12} \approx 10^{13}$ s⁻¹ at room temperature (298 K). It is clear, that if the ratio of the partition functions deviates significantly from one and, thus, there are significant changes in $\Delta S^0_{des,2D,act,m}$ when going from the adsorbed state to the activated state, substantial deviations from the 'benchmark' value of $10^{13}$ s⁻¹ are expected. For example, $A_{des,2D} > 10^{13}$ s⁻¹ with

$\Delta S^0_{des,2D,act,m} > 0$ and $\frac{q^{0'}_{TS,m}}{q^0_{ads,2D,m}} > 1$, which represents the case where a greater number of accessible configurations of the TS (more degrees of freedom) are available that are more easily excited by thermal energy than the adsorbed state. In contrast, $A_{des,2D} < 10^{13}$ s⁻¹, $\Delta S^0_{des,2D,act,m} < 0$ and $\frac{q^{0'}_{TS,m}}{q^0_{ads,2D,m}} < 1$ indicates that the TS is constrained where, e.g., the molecule has to obtain a specific configuration in the activated complex. This case is usually associated with activated adsorption (Kolasinski, 2012).


## 4.2 Desorption of a 2D ideal lattice gas

For the case of the adsorbate being a 2D ideal lattice gas, but the TS a 2D ideal gas, the associated equilibrium constant is related to the free energy change between the TS and the absorbed state, each expressed with the corresponding standard molar partition function, $q^0_{TS,m}$, and $q_{ads,latt}$ (Supplement Eqs. (4), (38), and (175)):

$$K^0_{des,latt,TS} = \frac{\left(\frac{q^0_{TS,m}}{N_A}\right)}{q_{ads,latt}\frac{(1-\theta^0)}{\theta^0}} e^{-\frac{\left(E^0_{des}+E^0_b\right)}{RT}} . \tag{79}$$

The equilibrium constant is also related to the ratio of activities:

$$K^0_{des,latt,TS} = e^{-\Delta G^0_{des,latt,TS,m}/RT} = \frac{\frac{(N_{TS}/\mathcal{A})}{(N_{TS}/\mathcal{A})^0}}{\frac{(\theta/(1-\theta))}{(\theta^0/(1-\theta^0))}} = \frac{\frac{N_{TS}}{(N_{TS}/\mathcal{A})^0}}{\frac{(\theta/(1-\theta))}{(\theta^0/(1-\theta^0))}} = \frac{\left(\frac{q^0_{TS,m}}{N_A}\right)}{\left(q_{ads,latt}\frac{(1-\theta^0)}{\theta^0}\right)} e^{-\frac{\left(E^0_{des}+E^0_b\right)}{RT}} . \tag{80}$$





Note that $q_{ads,latt}$ represents only vibrations and rotations. In addition, for the 2D ideal lattice gas, the surface activity is based on the coverage, and correspondingly, for the normalization to the standard state, $\theta^0/(1-\theta^0)$ is replacing $(N_{ads}/\mathcal{A})^0$. Using

the same procedure as for the 2D ideal gas case, i.e., rearranging Eq. (80), leads to (Campbell et al., 2016):

$$\mathcal{N}_{TS} = \frac{N_{TS}}{\mathcal{A}} = \frac{\left(\frac{q^0_{TS,m}}{N_A}\right)\frac{(\theta/(1-\theta))}{(\theta^0/(1-\theta^0))}}{\left(q_{ads,latt}\frac{(1-\theta^0)}{\theta^0}\right)}(N_{TS}/\mathcal{A})^0 e^{-\frac{\left(E^0_{des}+E^0_b\right)}{RT}} = \frac{\left(\frac{q^0_{TS,m}}{N_A}\right)}{(q_{ads,latt})}(N_{TS}/\mathcal{A})^0 e^{-\frac{\left(E^0_{des}+E^0_b\right)}{RT}}(\theta/(1-\theta)) . \tag{81}$$

Setting this into Eq. (62) yields

$$\frac{R_{des,latt}}{\mathcal{A}} = \kappa\left(\frac{k_BT}{h}\right)\frac{\left(\frac{q^0_{TS,m}}{N_A}\right)}{q_{TS,des}q_{ads,latt}}(N_{TS}/\mathcal{A})^0 e^{-\frac{\left(E^0_{des}+E^0_b\right)}{RT}}(\theta/(1-\theta)). \tag{82}$$

We note that Eq. (82) differs from Eq. (71) for the ideal 2D gas, such that $q_{ads,latt}$ has only vibrational degrees of freedom

(instead of two translational motions) (Campbell et al., 2016).

Equation (82) highlights that the desorption rate is not proportional to the surface concentration but is depending non-linearly on the surface coverage $\theta$ for high $\theta$. Figure 5 highlights this behavior. The desorption rate first changes linearly with coverage for both adsorbate models, but then strongly non-linearly for the 2D ideal lattice gas when approaching high ($\theta = 1$) surface coverages. This fact makes conversion of the rate expression to the surface activity challenging. The rate of change in

surface activity is related to the rate of change in $\theta$ as (Supplement Eq. (2))

$$\frac{R_{des,latt}}{\mathcal{A}\mathcal{N}_{ads,max}} = -\frac{d\theta}{dt} . \tag{83}$$

Assuming that the steady state surface concentration of the TS remains much smaller than the number of adsorbed molecules (in a time interval necessary to populate the TS), and correspondingly the desorption rate remains relatively small in comparison to the actual coverage, we can write

$$\frac{R_{des,latt}}{\mathcal{A}\mathcal{N}_{ads,max}} = -\frac{d\theta}{dt} \approx -\frac{d(\theta/(1-\theta))}{dt} , \tag{84}$$

since

$$\lim_{\theta \to 0}\left(\frac{\theta}{1-\theta}\right) \approx \theta .$$

In other words, for small rates of change of $\theta$, the desorption rate in terms of rate of change of activity can be assumed to depend linearly on $\theta$. Since this concerns the rate of change of $\theta$, Eq. (84) holds for any coverage. This allows us to express

the desorption rate in terms of activity as

$$\frac{R_{des,latt}}{\mathcal{A}\mathcal{N}_{ads,max}(\theta^0/(1-\theta^0))} \approx -\frac{d\left(\frac{(\theta/(1-\theta))}{(\theta^0/(1-\theta^0))}\right)}{dt} = -\frac{da_{ads,latt}}{dt} . \tag{85}$$

Therefore, dividing Eq. (82) by $\mathcal{N}_{ads,max}(\theta^0/(1-\theta^0))$ leads to the corresponding activity-based desorption rate expression

$$R^a_{des,latt} = \kappa\left(\frac{k_BT}{h}\right)\frac{\left(\frac{q^0_{TS,m}}{N_A}\right)}{q_{TS,des}(q_{ads,latt})}e^{-\frac{\left(E^0_{des}+E^0_b\right)}{RT}}\frac{(N_{TS}/\mathcal{A})^0}{\mathcal{N}_{ads,max}}a_{ads,latt} . \tag{86}$$





We now follow a similar derivation as for the 2D ideal gas. We define $\Delta G^0_{des,latt,act,m}$ of desorption as $\Delta G^0_{des,latt,TS,m}$
minus the TS's free energy associated with the motion along the desorption coordinate and obtain

$$e^{-\Delta G^0_{des,latt,act,m}/RT} = \frac{e^{-\Delta G^0_{des,latt,TS,m}/RT}}{q_{TS,des}} = \frac{1}{q_{TS,des}} \frac{\left(\frac{q^0_{TS,m}}{N_A}\right)}{\left(q_{ads,latt} \frac{(1-\theta^0)}{\theta^0}\right)} e^{-\frac{\left(E^0_{des}+E^0_b\right)}{RT}} . \tag{87}$$

Thus, we can express the desorption rate for an adsorbate treated as a 2D ideal lattice gas as

$$\frac{R_{des,latt}}{\mathcal{A}} = \kappa \left(\frac{k_B T}{h}\right) \frac{\left(\frac{q^0_{TS,m}}{N_A}\right)}{q_{TS,des}\left(q_{ads,latt} \frac{(1-\theta^0)}{\theta^0}\right)} e^{-\frac{\left(E^0_{des}+E^0_b\right)}{RT}} \frac{(1-\theta^0)}{\theta^0} (N_{TS}/\mathcal{A})^0 (\theta/(1-\theta)) = $$

$$\kappa \left(\frac{k_B T}{h}\right) e^{-\Delta G^0_{des,latt,act,m}/RT} \frac{(1-\theta^0)}{\theta^0} (N_{TS}/\mathcal{A})^0 (\theta/(1-\theta)) = \kappa \left(\frac{k_B T}{h}\right) e^{-\Delta G^0_{des,latt,act,m}/RT} (N_{TS}/\mathcal{A})^0 \frac{(\theta/(1-\theta))}{(\theta^0/(1-\theta^0))} . \tag{88}$$

The activity-based desorption rate expression becomes

$$R^a_{des,latt} = \kappa \left(\frac{k_B T}{h}\right) \frac{\left(\frac{q^0_{TS,m}}{N_A}\right)}{q_{TS,des}\left(q_{ads,latt} \frac{(1-\theta^0)}{\theta^0}\right)} e^{-\frac{\left(E^0_{des}+E^0_b\right)}{RT}} \frac{(1-\theta^0)}{\theta^0} \frac{(N_{TS}/\mathcal{A})^0}{\mathcal{N}_{ads,max}} a_{ads,latt} = $$

$$\kappa \left(\frac{k_B T}{h}\right) e^{-\Delta G^0_{des,latt,act,m}/RT} \frac{(1-\theta^0)}{\theta^0} \frac{(N_{TS}/\mathcal{A})^0}{\mathcal{N}_{ads,max}} a_{ads,latt} . \tag{89}$$

Therefore, the desorption rate coefficient (in units s⁻¹) related to the surface activity is given by

$$k^a_{des,latt} = \kappa \left(\frac{k_B T}{h}\right) \frac{\left(\frac{q^0_{TS,m}}{N_A}\right)}{q_{TS,des}(q_{ads,latt})} \frac{(N_{TS}/\mathcal{A})^0}{\mathcal{N}_{ads,max}} e^{-\frac{\left(E^0_{des}+E^0_b\right)}{RT}} = \kappa \left(\frac{k_B T}{h}\right) \frac{(1-\theta^0)}{\theta^0} \frac{(N_{TS}/\mathcal{A})^0}{\mathcal{N}_{ads,max}} e^{-\Delta G^0_{des,latt,act,m}/RT} . \tag{90}$$

While the activity-based desorption rate expression (Eq. (86)) clearly displays the first order decay behavior of the activity,
driven by $k^a_{des,latt}$, Eqs. (82) and (88) demonstrate that when expressed in terms of molecules desorbing per unit area and time,
it is not first order in the surface concentration but shows a strong dependence on the surface coverage, $(\theta/(1-\theta))$, otherwise
included in the activity. Therefore, for high surface coverage, an apparent $k_{des,latt}$ cannot easily be derived. For low coverage
(of the adsorbate, not of the transition state), $(\theta/(1-\theta)) \approx \theta = \frac{\mathcal{N}_{ads}}{\mathcal{N}_{ads,max}}$, the rate equations (82) simplifies to

$$\frac{R_{des,latt}}{\mathcal{A}} \approx \kappa \left(\frac{k_B T}{h}\right) \frac{\left(\frac{q^0_{TS,m}}{N_A}\right)}{q_{TS,des}(q_{ads,latt})} \frac{(N_{TS}/\mathcal{A})^0}{\mathcal{N}_{ads,max}} e^{-\frac{\left(E^0_{des}+E^0_b\right)}{RT}} \mathcal{N}_{ads} . \tag{91}$$

From this it follows, $k_{des,latt}(\theta \ll 1) = k^a_{des,latt}$. This demonstrates that the decay of surface concentration at high coverage
cannot be used to derive $E^0_{des}$, as also pointed out by Campbell et al. (2016). In other words, the decay of the surface coverage
is not a first-order process at high coverages. Using Eq. (70) in Eq. (82), yields

$$\frac{R_{des,latt}}{\mathcal{A}} = \kappa \left(\frac{k_B T}{h}\right) \frac{\left(\frac{q^{0'}_{TS,m}}{N_A}\right)}{(q_{ads,latt})} (N_{TS}/\mathcal{A})^0 e^{-\frac{\left(E^0_{des}+E^0_b\right)}{RT}} (\theta/(1-\theta)). \tag{92}$$

Note that the last equation is consistent with the desorption rate derived by Campbell et al. (2016) for the special case of $\theta^0 =$
0.5.





We can now express the desorption rate coefficient as

$$k_{des,latt}^a = \kappa \left(\frac{k_B T}{h}\right) \frac{\left(\frac{q_{TS,m}^{0'}}{N_A}\right)}{(q_{ads,latt})} \frac{(N_{TS}/\mathcal{A})^0}{\mathcal{N}_{ads,max}} e^{-\frac{\left(E_{des}^0+E_b^0\right)}{RT}} = \kappa \left(\frac{k_B T}{h}\right) \frac{(q_{TS}'/\mathcal{A})}{q_{ads,latt}\mathcal{N}_{ads,max}} e^{-\frac{\left(E_{des}^0+E_b^0\right)}{RT}} = \frac{(2\pi m k_B T/h^2)}{q_{ads,latt}\mathcal{N}_{ads,max}} e^{-\frac{\left(E_{des}^0+E_b^0\right)}{RT}} .$$

(93)

For the second and third expression in Eq. (93), we have converted the standard molar partition function back to the molecular

ones, using $\left(\frac{N_{TS}}{\mathcal{A}}\right)^0 = \frac{n_{TS}\cdot N_A}{\mathcal{A}^0} = \frac{N_A}{\mathcal{A}_m^0}$.

We can establish the link between the entropy and the pre-exponential factor by taking the expression for $k_{des}^a$ and

inserting the definition of $\Delta G_{des,latt,act,m}^0$ accounting for the relationship between $E_{des}^0$ and $\Delta H_{des,latt,act,m}^0$ (Supplement Eq. 171):

$\Delta H_{des,latt,act,m}^0 = H_{act,m}^0 - H_{ads,latt,m}^0 = 2RT - \frac{N_A \cdot h\nu}{e^{\beta h\nu}-1} + RT \frac{\ln(1-\theta^0)}{\theta^0} + E_{des}^0 + E_b^0 .$ (94)

Neglecting vibrations, we obtain

$\Delta H_{des,latt,act,m}^0 \approx 2RT + E_{des}^0 + E_b^0 + RT \frac{\ln(1-\theta^0)}{\theta^0} .$ (95)

Then it follows

$$k_{des,latt}^a = \kappa \left(\frac{k_B T}{h}\right) \frac{(N_{TS}/\mathcal{A})^0}{(\theta^0/(1-\theta^0))\mathcal{N}_{ads,max}} e^{-\Delta G_{des,latt,act,m}^0/RT} = \kappa \left(\frac{k_B T}{h}\right) \frac{(q_{TS}'/\mathcal{A})}{q_{ads,latt}\mathcal{N}_{ads,max}} e^{-\frac{\left(E_{des}^0+E_b^0\right)}{RT}} \equiv$$

$\kappa \left(\frac{k_B T}{h}\right) \frac{(N_{TS}/\mathcal{A})^0}{(\theta^0/(1-\theta^0))\mathcal{N}_{ads,max}} e^{\Delta S_{des,latt,act,m}^0/R} e^{-\Delta H_{des,latt,act,m}^0/RT} = \kappa \left(\frac{k_B T}{h}\right) \frac{(q_{TS}'/\mathcal{A})}{q_{ads,latt}\mathcal{N}_{ads,max}} e^{-\frac{\left(E_{des}^0+E_b^0\right)}{RT}} \equiv$

$\kappa \left(\frac{k_B T}{h}\right) \frac{(N_{TS}/\mathcal{A})^0}{(\theta^0/(1-\theta^0))\mathcal{N}_{ads,max}} e^{-2} e^{\Delta S_{des,latt,act,m}^0/R} e^{-\frac{\left(E_{des}^0+E_b^0\right)}{RT}} (1-\theta^0)^{-\frac{1}{\theta^0}} = \kappa \left(\frac{k_B T}{h}\right) \frac{(q_{TS}'/\mathcal{A})}{q_{ads,latt}\mathcal{N}_{ads,max}} e^{-\frac{\left(E_{des}^0+E_b^0\right)}{RT}} .$ (96)

With this, we can derive the pre-exponential factor as

$$A_{des,latt}^a = \kappa \left(\frac{k_B T}{h}\right) \frac{(q_{TS}'/\mathcal{A})}{q_{ads,latt}\mathcal{N}_{ads,max}} = \kappa \left(\frac{k_B T}{h}\right) \frac{(N_{TS}/\mathcal{A})^0 (1-\theta^0)^{-\frac{1}{\theta^0}}}{(\theta^0/(1-\theta^0))\mathcal{N}_{ads,max}} e^{-2} e^{\left(\frac{\Delta S_{des,latt,act,m}^0}{R}\right)} =$$

$\kappa \left(\frac{k_B T}{h}\right) \frac{(N_{TS}/\mathcal{A})^0}{(\theta^0/(1-\theta^0))\mathcal{N}_{ads,max}} e^{-2} e^{\left(\frac{S_{act,m}^0 - S_{ads,latt,m}^0}{R}\right)} .$ (97)

We can, thus, identify

$\frac{(N_{TS}/\mathcal{A})^0 (1-\theta^0)^{-\frac{1}{\theta^0}}}{(\theta^0/(1-\theta^0))} e^{\left(\frac{\Delta S_{des,latt,act,m}^0}{R}\right)} = e^2 \frac{(q_{TS}'/\mathcal{A})}{q_{ads,latt}} .$ (98)

Again, as for the previous case, Eqs. (96) and (97) clearly show that when using thermodynamic data to assess the TS, the correct standard state needs being applied to calculate $A_{des,latt}^a$ from the entropy of activation.





### 4.3 Adsorbate model comparison of desorption rate and pre-exponential factor

Since, strictly speaking, the desorption rate law is representing a first-order process acting on the surface activity, it is also straightforward to understand that the desorption rate, when expressed as rate of change of activity per unit time is proportional to the surface activity, as shown in Fig. 6, independent of the adsorbate model. Thus, even when the surface coverage gets high, the activity based first-order desorption rate coefficient remains constant. The consequence of this becomes then directly apparent in Fig. 7, showing the desorption rate expressed as rate of change of surface concentration per unit area and unit time,

as a function of the surface coverage. For the 2D ideal gas case, the linear relationship is maintained, i.e., the surface concentration-based desorption rate coefficient is constant, and thus independent of the surface coverage. In contrast, for the 2D ideal lattice gas case, the desorption rate is rapidly increasing towards high surface coverages, clearly demonstrating the non-first order behavior of desorption when expressed in terms of surface concentration. This behavior is a consequence of the high configurational entropy at high coverages and naturally results from a consistent description of the surface activity.

Therefore, the dependency of the desorption rate on coverage is not due to surface sites with different desorption energies, but a consequence of the applied lattice gas adsorption model that entails a limited number of equivalent sites. In other words, the lifetime of an individual adsorbate molecule depends on the overall surface coverage, exerting shorter adsorbate lifetimes for greater surface coverages. Therefore, as also pointed out by Campbell et al. (2016), experimental desorption rate measurements need to be analyzed with care when deriving the desorption energy from measured desorption rates.

The features of the rate law for desorption acting as a first order process on the surface activity become then also manifest in the time dependent decay of the surface coverage for the two adsorbate models. As expected for the 2D ideal gas case, where surface activity and surface coverage are proportional to each other, the first order and thus single exponential decay of the surface activity leads to a corresponding single exponential decay of the surface coverage, as shown in Fig. 8. In contrast, as demonstrated in Fig. 9, the single exponential decay of the surface activity of the 2D ideal lattice gas case leads to a non-

exponential decay of the surface coverage. This further emphasizes the need to carefully analyze experimental data of desorption rate measurements, especially if short time scales are considered.

As discussed above the pre-exponential factor is often assumed to be $A_{des} \approx 10^{13}$ s$^{-1}$. Figure 10 shows $A_{des}$ for both adsorbate models as a function of temperature. For the 2D ideal gas, $A_{des}$ displays a weak temperature dependency and, when approaching room temperature, $A_{des,2D}$ is close to typically applied $10^{13}$ s$^{-1}$. For the 2D ideal lattice gas, $A_{des,latt}^{a}$ is about 3

orders of magnitude larger and exhibits a stronger temperature dependency compared to the 2D ideal gas. The greater values for $A_{des,latt}^{a}$ can be understood the following way. When going from a localized bound species (i.e., 2D ideal lattice gas) to a 2D ideal gas (TS), it is very likely that the ratio of partition functions is larger than one and $\Delta S_{des,latt,act,m}^{0} > 0$. Hence, it can be expected that in these cases $A_{des,latt}^{a} > 10^{13}$ s$^{-1}$, as demonstrated in Fig. 10. Even when ignoring internal rotations, the change in translational degrees of freedom between the 2D ideal lattice gas adsorbate and the 2D ideal gas of the TS, the configurational

contribution to the 2D ideal lattice gas adsorbate leads to an increase in $A_{des,latt}^{a} > 10^{15}$ s$^{-1}$ (if $\kappa$ remains 1).





## 5. Rate of Adsorption

We derive the adsorption rates of gas molecules transferring into the 2D ideal and 2D ideal lattice gas absorbates. The adsorption proceeds via the TS, which as in the case of desorption, is treated as a 2D ideal gas.

When considered from the gas phase side, the equilibrium constant between the gas phase and the adsorbed state is given
by the inverse ratio of activities compared to the case of desorption, as now the adsorbed state is the product:

$$K^0_{ads,g,2D} = \frac{a_{ads,2D}}{a_g} = \frac{\frac{(N_{ads}/\mathcal{A})}{(N_{ads}/\mathcal{A})^0}}{\frac{(N_g/\mathcal{V})}{(N_g/\mathcal{V})^0}} = \frac{\frac{(N_{ads})}{(N_A/\mathcal{A}_m^0)}}{\frac{(N_g/\mathcal{V})}{(N_A/\mathcal{V}_m^0)}} \equiv \frac{1}{K^0_{des,2D,g}} \tag{99}$$

and

$$K^0_{ads,g,latt} = \frac{a_{ads,latt}}{a_g} = \frac{\frac{(\theta/(1-\theta))}{(\theta^0/(1-\theta^0))}}{\frac{(N_g/\mathcal{V})}{(N_g/\mathcal{V})^0}} = \frac{\frac{(\theta/(1-\theta))}{(\theta^0/(1-\theta^0))}}{\frac{(N_g/\mathcal{V})}{(N_A/\mathcal{V}_m^0)}} \equiv \frac{1}{K^0_{des,latt,g}} . \tag{100}$$

The relationship to the equilibrium constant of desorption holds irrespective of whether the adsorbed state is a 2D ideal gas or
2D ideal lattice gas due to the reversible nature of the adsorption equilibrium.

In general, the adsorption rate can be expressed as

$$\frac{R_{ads,2D}}{\mathcal{A}} = \frac{dN_{ads}}{dt} = -\frac{dN_g}{dt}\frac{\mathcal{V}}{\mathcal{A}} = k_{ads}N_g\frac{\mathcal{V}}{\mathcal{A}} \tag{101}$$

and

$$\frac{R_{ads,latt}}{\mathcal{A}} = \frac{dN_{ads}}{dt} = \frac{d\theta}{dt}N_{ads,max} = -\frac{dN_g}{dt}\frac{\mathcal{V}}{\mathcal{A}} = k_{ads}N_g\frac{\mathcal{V}}{\mathcal{A}} \tag{102}$$

with $\frac{d\theta}{dt} = \frac{k_{ads}}{N_{ads,max}}N_g\frac{\mathcal{V}}{\mathcal{A}}$ , $\tag{103}$

where $k_{ads}$ represents the first-order rate coefficient for adsorption (in units s$^{-1}$), describing the rate of change of gas phase concentration or activity. Considering the rate expression in terms of gain of adsorbed molecules per unit area and time, the rate of loss from the gas phase needs to be multiplied by $\frac{\mathcal{V}}{\mathcal{A}}$. Since adsorption proceeds via the TS that is assumed to be similar to a 2D ideal gas, $k_{ads}$ is the same first-order rate coefficient for the adsorption into the 2D ideal gas and 2D ideal lattice gas
adsorbate model.

The rate of change of surface activity for the 2D ideal gas is given by

$$R^a_{ads,2D} = \frac{R_{ads,2D}}{\mathcal{A}(N_{ads,2D}/\mathcal{A})^0} = k_{ads}N_g\frac{\mathcal{V}}{\mathcal{A}(N_{ads,2D}/\mathcal{A})^0} = k_{ads}N_g\frac{\mathcal{V}(N_g/\mathcal{V})^0}{\mathcal{A}(N_{ads,2D}/\mathcal{A})^0(N_g/\mathcal{V})^0} = k_{ads}\frac{N_g}{(N_g/\mathcal{V})^0}\frac{\mathcal{V}(N_g/\mathcal{V})^0}{\mathcal{A}(N_{ads,2D}/\mathcal{A})^0} =$$

$$k_{ads}a_g\frac{\mathcal{V}\mathcal{A}_m^0}{\mathcal{A}\mathcal{V}_m^0} = k^a_{ads}a_g\frac{\mathcal{V}\mathcal{A}_m^0}{\mathcal{A}\mathcal{V}_m^0} , \tag{104}$$

and for the 2D ideal lattice gas as





$$R^a_{ads,latt} = \frac{da_{ads,latt}}{dt} = \frac{d}{dt}\frac{\theta}{(1-\theta)}\frac{(1-\theta^0)}{\theta^0} \approx \frac{d\theta}{dt}\frac{(1-\theta^0)}{\theta^0} = \frac{k_{ads}}{\mathcal{N}_{ads,max}}\mathcal{N}_g\frac{\mathcal{V}}{\mathcal{A}}\frac{(1-\theta^0)}{\theta^0} = \frac{k_{ads}}{\mathcal{N}_{ads,max}}\frac{N_g}{(N_g/\mathcal{V})^0}\frac{\mathcal{V}(N_g/\mathcal{V})^0}{\mathcal{A}}\frac{(1-\theta^0)}{\theta^0} =$$

$$\frac{k_{ads}}{\mathcal{N}_{ads,max}}a_g\frac{\mathcal{V}}{\mathcal{A}}\frac{(N_g/\mathcal{V})^0(1-\theta^0)}{\theta^0} = \frac{k^a_{ads}}{\mathcal{N}_{ads,max}}a_g\frac{\mathcal{V}}{\mathcal{A}}\frac{(N_g/\mathcal{V})^0(1-\theta^0)}{\theta^0}. \tag{105}$$

We note that although $k_{ads}$ is the same for both gas adsorbate models, $R^a_{ads,2D}$ and $R^a_{ads,latt}$ will differ, as evident from Eqs. (104) and (105), respectively and shown further below (Eqs. (126) and (127)). As outlined above and in the Supplement (S3.3), the activity of the 2D ideal lattice gas scales with $\frac{(\theta/(1-\theta))}{(\theta^0/(1-\theta^0))}$. To provide an analytical solution using our definitions, in Eq.

(105), we make the assumption $\frac{d}{dt}\frac{\theta}{(1-\theta)}\frac{(1-\theta^0)}{\theta^0} \approx \frac{d\theta}{dt}\frac{(1-\theta^0)}{\theta^0}$, meaning that we consider small enough rates of change so that this condition is justified. Since $k^a_{ads}$ describes the decay of the gas phase activity, which is proportional to its concentration, we follow that $k^a_{ads} = k_{ads}$. In turn, the factor $\frac{\mathcal{V}}{\mathcal{A}}$ needs to be normalized to the corresponding ratio of standard molar volume to surface area to convert from gas phase activity change to surface activity change, in the last expression of Eq. (105).

TS theory for adsorption is encompassing the same steps as that for desorption but starting from the gas phase side.

Considered from the gas phase, the equilibrium constant between the gas phase and the TS for adsorption is related to the free energy change between the gas and the TS, each expressed with the corresponding standard molar partition functions, as defined by (Supplement Eq. (199)):

$$K^0_{ads,g,TS} = e^{-\Delta G^0_{ads,g,TS,m}/RT} = \frac{q^0_{TS,m}}{q^0_{g,m}}e^{-\frac{E^0_b}{RT}}. \tag{106}$$

Note that we treat the general case of activated adsorption here, meaning that the TS's inner energy is elevated by the barrier

height above the reference level ($E^0_b$), leading to the corresponding Arrhenius term in Eq. (106). The equilibrium constant is also related to the ratio of activities (Supplement Eq. (5)):

$$K^0_{ads,g,TS} = e^{-\Delta G^0_{ads,g,TS,m}/RT} = \frac{a_{TS}}{a_g} = \frac{\frac{(N_{TS}/\mathcal{A})}{(N_{TS}/\mathcal{A})^0}}{\frac{(N_g/\mathcal{V})}{(N_g/\mathcal{V})^0}} = \frac{\frac{(N_{TS})}{(N_A/\mathcal{A}^0_m)}}{\frac{(N_g/\mathcal{V})}{(N_A/v^0_m)}}. \tag{107}$$

As in the case of desorption, the adsorption rate can be obtained by assuming that the TS has the same finite width $d$ across which the molecule moves with its mean thermal velocity in the direction orthogonal to the surface

$$\frac{R_{ads,2D}}{\mathcal{A}} = \kappa\left(\frac{N_{TS}}{\mathcal{A}}\right)\frac{(k_BT/2\pi m)^{1/2}}{d}, \tag{108}$$

where $\kappa$ is the same transmission coefficient defining the probability with which the activated complex proceeds to adsorption as that for desorption due to microscopic reversibility (Kolasinski, 2012). The partition function for the translational motion of the TS in the direction of adsorption is

$$q_{TS,ads} = (2\pi mk_BT/h^2)^{1/2}d. \tag{109}$$

Solving this for $d$ and inserting into Eq. (108) allows to express the adsorption rate as a function of this partition function:

$$\frac{R_{ads,2D}}{\mathcal{A}} = \kappa\left(\frac{k_BT}{h}\right)\left(\frac{N_{TS}}{\mathcal{A}}\right)\frac{1}{q_{TS,ads}}. \tag{110}$$





The surface concentration of the TS can be derived from the equilibrium (Eq. (107))

$$\mathcal{N}_{TS} = \frac{N_{TS}}{\mathcal{A}} = \frac{q^0_{TS,m}}{q^0_{g,m}} e^{-\frac{E^0_b}{RT}} \frac{(N_{TS}/\mathcal{A})^0}{(N_g/\mathcal{V})^0} \mathcal{N}_g \ . \tag{111}$$

Inserting Eq. (111) into Eq. (110) leads to

$$\frac{R_{ads,2D}}{\mathcal{A}} = \kappa \left(\frac{k_B T}{h}\right) \frac{1}{q_{TS,ads}} \frac{q^0_{TS,m}}{q^0_{g,m}} e^{-\frac{E^0_b}{RT}} \frac{(N_{TS}/\mathcal{A})^0}{(N_g/\mathcal{V})^0} \mathcal{N}_g \ . \tag{112}$$

When defining $\Delta G^0_{ads,g,act,m}$ of adsorption as $\Delta G^0_{ads,g,TS,m}$ (Supplement Eqs. (193) and (207)) minus the TS's free energy associated with the motion along the adsorption coordinate, expressed by its molecular partition function, $q_{TS,ads}$, we obtain

$$e^{-\Delta G^0_{ads,g,act,m}/RT} = \frac{e^{-\Delta G^0_{ads,g,TS,m}/RT}}{q_{TS,ads}} = \frac{1}{q_{TS,ads}} \frac{q^0_{TS,m}}{q^0_{g,m}} e^{-\frac{E^0_b}{RT}} \ . \tag{113}$$

With this definition of $\Delta G^0_{ads,g,act,m}$, we can express the adsorption rate as

$$\frac{R_{ads,2D}}{\mathcal{A}} = \kappa \left(\frac{k_B T}{h}\right) e^{-\Delta G^0_{ads,g,act,m}/RT} \frac{(N_{TS}/\mathcal{A})^0}{(N_g/\mathcal{V})^0} \mathcal{N}_g \ . \tag{114}$$

When using the definition of the adsorption rate coefficient linking the loss rate from the gas phase with the gain of adsorbed species on the surface, i.e., $\frac{R_{ads,2D}}{\mathcal{A}} = k_{ads} \mathcal{N}_g \frac{\mathcal{V}}{\mathcal{A}}$, the adsorption rate coefficient becomes

$$k_{ads} = \left(\frac{k_B T}{h}\right) e^{-\Delta G^0_{ads,g,act,m}/RT} \frac{(N_{TS}/\mathcal{A})^0}{(N_g/\mathcal{V})^0} \frac{\mathcal{A}}{\mathcal{V}} = \kappa \left(\frac{k_B T}{h}\right) \frac{1}{q_{TS,ads}} \frac{q^0_{TS,m}}{q^0_{g,m}} e^{-\frac{E^0_b}{RT}} \frac{(N_{TS}/\mathcal{A})^0}{(N_g/\mathcal{V})^0} \frac{\mathcal{A}}{\mathcal{V}} \ . \tag{115}$$

Defining $q^{0\prime}_{TS}/\mathcal{A}^0_m$ as the partition function for the TS after omitting motion in the direction of the reaction coordinate (Campbell et al., 2016), this leaves the partition function for a 2D ideal gas (Supplement Eqs. (60) and (118)):

$$\left(\frac{q^0_{TS,m}}{q_{TS,ads}}\right) = q^{0\prime}_{TS,m} = q^0_{TS,2D,m} = \mathcal{A}^0_m (2\pi m k_B T/h^2)^{2/2} \ . \tag{116}$$

Using Eq. (116) in Eq. (112), we obtain

$$\frac{R_{ads,2D}}{\mathcal{A}} = \kappa \left(\frac{k_B T}{h}\right) \frac{q^{0\prime}_{TS,m}}{q^0_{g,m}} e^{-\frac{E^0_b}{RT}} \frac{(N_{TS}/\mathcal{A})^0}{(N_g/\mathcal{V})^0} \mathcal{N}_g \tag{117}$$

and identifying Eq. (117) with Eq. (104) yields

$$k_{ads} = \kappa \left(\frac{k_B T}{h}\right) \frac{q^{0\prime}_{TS,m}}{q^0_{g,m}} e^{-\frac{E^0_b}{RT}} \frac{(N_{TS}/\mathcal{A})^0}{(N_g/\mathcal{V})^0} \frac{\mathcal{A}}{\mathcal{V}} \ . \tag{118}$$

This is the same result as in Eq. (115) when using thermodynamic quantities.

We can convert the standard molar partition functions back to the molecular ones. For that, we consider that $\left(\frac{N_{TS}}{\mathcal{A}}\right)^0 = \frac{n_{TS} \cdot N_A}{\mathcal{A}^0} = \frac{N_A}{\mathcal{A}^0_m}$ and $\left(\frac{N_g}{\mathcal{V}}\right)^0 = \frac{n_g \cdot N_A}{\mathcal{V}^0} = \frac{N_A}{\mathcal{V}^0_m}$, we obtain

$$\frac{1}{q_{TS,ads}} \frac{q^0_{TS,m}}{q^0_{g,m}} \frac{(N_{TS}/\mathcal{A})^0}{(N_g/\mathcal{V})^0} \frac{\mathcal{A}}{\mathcal{V}} = \frac{q^{0\prime}_{TS,m}}{q^0_{g,m}} \frac{(N_{TS}/\mathcal{A})^0}{(N_g/\mathcal{V})^0} \frac{\mathcal{A}}{\mathcal{V}} = \frac{\mathcal{A}^0_m (2\pi m k_B T/h^2)^{2/2} \frac{N_A}{\mathcal{A}^0_m}}{\mathcal{V}^0_m (2\pi m k_B T/h^2)^{3/2} \frac{N_A}{\mathcal{V}^0_m}} \frac{\mathcal{A}}{\mathcal{V}} = \frac{q^{\prime}_{TS}/\mathcal{A}}{q_g/\mathcal{V}} \frac{\mathcal{A}}{\mathcal{V}} = \frac{1}{(2\pi m k_B T/h^2)^{1/2}} \frac{\mathcal{A}}{\mathcal{V}} \ . \tag{119}$$

This yields



$$k_{ads} = \kappa \left(\frac{k_B T}{h}\right) \frac{1}{(2\pi m k_B T/h^2)^{1/2}} e^{-\frac{E_b^0}{RT}} \frac{\mathcal{A}}{\mathcal{V}} . \tag{120}$$

As in the case of desorption, we can compare the thermodynamic derivation of $k_{ads}$ (left hand side below) with the one based on the partition functions (right hand side below):

$$k_{ads} = \kappa \left(\frac{k_B T}{h}\right) e^{-\Delta G_{ads,g,act,m}^0/RT} \frac{(N_{TS}/\mathcal{A})^0}{(N_g/\mathcal{V})^0} \frac{\mathcal{A}}{\mathcal{V}} = \kappa \left(\frac{k_B T}{h}\right) \left(\frac{q_{TS}'/\mathcal{A}}{q_g/\mathcal{V}}\right) e^{-\frac{E_b^0}{RT}} \frac{\mathcal{A}}{\mathcal{V}} \equiv$$


$$\kappa \left(\frac{k_B T}{h}\right) \frac{(N_{TS}/\mathcal{A})^0}{(N_g/\mathcal{V})^0} e^{\Delta S_{ads,g,act,m}^0/R} e^{-\Delta H_{ads,g,act,m}^0/RT} \frac{\mathcal{A}}{\mathcal{V}} = \kappa \left(\frac{k_B T}{h}\right) \left(\frac{q_{TS}'/\mathcal{A}}{q_g/\mathcal{V}}\right) e^{-\frac{E_b^0}{RT}} \frac{\mathcal{A}}{\mathcal{V}} , \tag{121}$$

with $\Delta H_{ads,g,act,m}^0 = -\frac{1}{2}RT + E_b^0$ (Supplement Eq. (195)), we obtain

$$\kappa \left(\frac{k_B T}{h}\right) \frac{(N_{TS}/\mathcal{A})^0}{(N_g/\mathcal{V})^0} e^{\Delta S_{ads,g,act,m}^0/R} e^{1/2} e^{-\frac{E_b^0}{RT}} = \kappa \left(\frac{k_B T}{h}\right) \left(\frac{q_{TS}'/\mathcal{A}}{q_g/\mathcal{V}}\right) e^{-\frac{E_b^0}{RT}} .$$

$$\kappa \left(\frac{k_B T}{h}\right) \frac{(N_{TS}/\mathcal{A})^0}{(N_g/\mathcal{V})^0} e^{\Delta S_{ads,g,act,m}^0/R} e^{1/2} = \kappa \left(\frac{k_B T}{h}\right) \left(\frac{q_{TS}'/\mathcal{A}}{q_g/\mathcal{V}}\right) . \tag{122}$$

In the case of adsorption, the Arrhenius term is only driven by the barrier height. Therefore, the pre-exponential factor for

adsorption is (since $k_{ads}^a = k_{ads}$)

$$A_{ads,2D} = A_{ads,2D}^a = \kappa \left(\frac{k_B T}{h}\right) \left(\frac{q_{TS}'/\mathcal{A}}{q_g/\mathcal{V}}\right) \frac{\mathcal{A}}{\mathcal{V}} = \kappa \left(\frac{k_B T}{h}\right) \frac{(N_{TS}/\mathcal{A})^0}{(N_g/\mathcal{V})^0} e^{1/2} e^{\Delta S_{ads,g,act,m}^0/R} \frac{\mathcal{A}}{\mathcal{V}} = \kappa \left(\frac{k_B T}{h}\right) \frac{(N_{TS}/\mathcal{A})^0}{(N_g/\mathcal{V})^0} \frac{\mathcal{A}}{\mathcal{V}} e^{1/2} e^{\left(\frac{S_{act,m}^0 - S_{g,m}^0}{R}\right)} .$$

$$\tag{123}$$

Thus, we can identify

$$\frac{(N_{TS}/\mathcal{A})^0}{(N_g/\mathcal{V})^0} e^{\Delta S_{ads,g,act,m}^0/R} = \left(\frac{q_{TS}'/\mathcal{A}}{q_g/\mathcal{V}}\right) e^{-1/2} . \tag{124}$$

This emphasizes the relationship between the entropy of activation and the ratio of the corresponding partition functions. Note

that when neglecting vibrations, $\left(\frac{q_{TS}'/\mathcal{A}}{q_g/\mathcal{V}}\right) = \frac{(2\pi m k_B T/h^2)^{2/2}}{(2\pi m k_B T/h^2)^{3/2}} = \frac{1}{(2\pi m k_B T/h^2)^{1/2}}$, which allows estimating the entropy of activation

for adsorption. For the examples discussed here (see Table S1), $\Delta S_{ads,g,act,m}^0$ = -53.98 J K$^{-1}$ mol$^{-1}$.

Thus, essentially, the gas loses one translational degree of freedom, and the rate of adsorption (vibrations neglected) can

be written as

$$\frac{R_{ads,2D}}{\mathcal{A}} = k_{ads} \mathcal{N}_g \frac{\mathcal{V}}{\mathcal{A}} = \kappa \left(\frac{k_B T}{h}\right) \left(\frac{q_{TS}'/\mathcal{A}}{q_g/\mathcal{V}}\right) e^{-\frac{E_b^0}{RT}} \mathcal{N}_g \frac{\mathcal{A}}{\mathcal{V}} \frac{\mathcal{V}}{\mathcal{A}} = \kappa \left(\frac{k_B T}{h}\right) \frac{h}{\sqrt{2\pi m k_B T}} e^{-\frac{E_b^0}{RT}} \mathcal{N}_g = \frac{p}{\sqrt{2\pi m k_B T}} \kappa e^{-\frac{E_b^0}{RT}} . \tag{125}$$

For the case considering activities, we obtain

$$R_{ads,2D}^a = k_{ads} a_g \frac{\mathcal{V} \mathcal{A}_m^0}{\mathcal{A} \mathcal{V}_m^0} = \kappa \left(\frac{k_B T}{h}\right) \left(\frac{q_{TS}'/\mathcal{A}}{q_g/\mathcal{V}}\right) e^{-\frac{E_b^0}{RT}} \frac{\mathcal{A}}{\mathcal{V}} a_g \frac{\mathcal{V} \mathcal{A}_m^0}{\mathcal{A} \mathcal{V}_m^0} = \kappa \left(\frac{k_B T}{h}\right) \frac{h}{\sqrt{2\pi m k_B T}} e^{-\frac{E_b^0}{RT}} \frac{\mathcal{N}_g}{(N_g/\mathcal{V})^0} \frac{\mathcal{A}_m^0}{\mathcal{V}_m^0} = \kappa \frac{p}{\sqrt{2\pi m k_B T}} e^{-\frac{E_b^0}{RT}} \frac{\mathcal{A}_m^0}{N_A}$$

$$\tag{126}$$

For the case of the 2D ideal lattice gas we can write, using the same definition for $k_{ads}$





$$R^a_{ads,latt} = \frac{k_{ads}}{\mathcal{N}_{ads,max}} a_g \frac{\mathcal{V}}{\mathcal{A}} \frac{(N_g/\mathcal{V})^0(1-\theta^0)}{\theta^0} = \kappa \left(\frac{k_BT}{h}\right)\left(\frac{q'_{TS}/\mathcal{A}}{q_g/\mathcal{V}}\right) e^{-\frac{E_b^0}{RT}} \frac{\mathcal{A}}{\mathcal{V}} \frac{1}{\mathcal{N}_{ads,max}} a_g \frac{\mathcal{V}}{\mathcal{A}} \frac{(N_g/\mathcal{V})^0(1-\theta^0)}{\theta^0} =$$

$$\kappa \left(\frac{k_BT}{h}\right)\left(\frac{q'_{TS}/\mathcal{A}}{q_g/\mathcal{V}}\right) e^{-\frac{E_b^0}{RT}} \frac{1}{\mathcal{N}_{ads,max}} a_g \frac{(N_g/\mathcal{V})^0(1-\theta^0)}{\theta^0} = \kappa \left(\frac{k_BT}{h}\right) \frac{h}{\sqrt{2\pi m k_B T}} e^{-\frac{E_b^0}{RT}} \frac{N_g}{(N_g/\mathcal{V})^0} \frac{1}{\mathcal{N}_{ads,max}} \frac{(N_g/\mathcal{V})^0(1-\theta^0)}{\theta^0} =$$

$$\kappa \frac{p}{\sqrt{2\pi m k_B T}} e^{-\frac{E_b^0}{RT}} \frac{1}{(N_g/\mathcal{V})^0} \frac{1}{\mathcal{N}_{ads,max}} \frac{(N_g/\mathcal{V})^0(1-\theta^0)}{\theta^0} = \kappa \frac{p}{\sqrt{2\pi m k_B T}} e^{-\frac{E_b^0}{RT}} \frac{1}{\mathcal{N}_{ads,max}} \frac{(1-\theta^0)}{\theta^0} . \qquad (127)$$

$\frac{p}{\sqrt{2\pi m k_B T}}$ represents the Hertz-Knudsen expression of the flux of molecules attempting to stick on surface atoms. Thus, CTST

is consistent with the collision rate multiplied with $\kappa$ for the case that the activated complex associated with the TS is considered a 2D ideal gas, the barrier is negligible, and no internal vibrations are considered.

As discussed in the previous section, the TS for adsorption is the same as that for desorption and is considered a 2D ideal gas. This means that the adsorptive flux, i.e., the adsorption rate in terms of gain of molecules per surface area and time, is simply proportional to the gas phase concentration, independent of the adsorption model used to describe the final state of

adsorption, as shown in Fig. 11. For the same reason, also the rate of change of surface activity is linearly related to the gas phase activity, as shown in Fig. 12. However, the meaning of the rate of change of surface activity is entirely different for the two adsorbate models, as discussed for the case of desorption. While for the 2D ideal gas model, the rate of change of surface activity is linearly related to the rate of change of surface coverage, for the 2D ideal lattice gas case, the same rate of change of surface activity is governed by a strongly non-linear relationship to the rate of change of surface coverage, thus, depending

on the actual coverage. This explains the visible slight deviations between $R^a_{ads,2D}$ and $R^a_{ads,latt}$ in Fig. 12 at high gas phase activity values, reflecting in fact different rates of change of surface coverages.

We can now look at the surface accommodation coefficient, $\alpha_s$, which is operationally defined as the ratio between the adsorption rate and the gas-kinetic collision rate (Kolb et al., 2010;Ammann et al., 2013;Crowley et al., 2013) in the description of the adsorption rate following

$$\frac{R_{ads,2D}}{\mathcal{A}} = \kappa \frac{p}{\sqrt{2\pi m k_B T}} = \mathcal{N}_g \kappa \frac{\sqrt{k_B T}}{\sqrt{2\pi m}} = \mathcal{N}_g \kappa \frac{\sqrt{8 k_B T}}{4\sqrt{\pi m}} = \alpha_s \mathcal{N}_g \frac{\omega}{4} , \qquad (128)$$

where $\omega$ represents the thermal velocity of the gas species. Keeping with this definition, but putting in the more general expression for the adsorption rate based on TS theory, the interpretation of $\alpha_s$ becomes different, as it is related to

$$\alpha_s = \frac{\kappa \left(\frac{k_BT}{h}\right)\frac{q'_{TS}/\mathcal{A}}{q_g/\mathcal{V}} e^{-\frac{E_b^0}{RT}} \mathcal{N}_g}{\mathcal{N}_g \frac{\omega}{4}} = \frac{\kappa \left(\frac{k_BT}{h}\right)\frac{q'_{TS}/\mathcal{A}}{q_g/\mathcal{V}} e^{-\frac{E_b^0}{RT}}}{\left(\frac{k_BT}{h}\right)(2\pi m k_B T/h^2)^{-1/2}} = \kappa \frac{q'_{TS}/\mathcal{A}}{q_g/\mathcal{V}} e^{-\frac{E_b^0}{RT}}(2\pi m k_B T/h^2)^{1/2} . \qquad (129)$$

Therefore, $\alpha_s = \kappa$ (and $= 1$, if $\kappa = 1$), if $\frac{q'_{TS}/\mathcal{A}}{q_g/\mathcal{V}} = (2\pi m k_B T/h^2)^{-1/2}$ and $E_b^0 = 0$, but is different in the presence of a barrier

or if other contributions are relevant in the partition functions of the activated complex associated with the TS or the gas phase species (such as internal vibrations or rotations). As mentioned above, the ratio of the partition functions is also related to the corresponding entropy of activation (i.e., non-zero if $\alpha_s$ deviates from $\kappa$).





Figure 13 shows how $\alpha_s$ depends on $E_b^0$ under the assumption of $\kappa = 1$ and $\frac{q'_{TS/\mathcal{A}}}{q_g/\mathcal{V}} = (2\pi m k_B T/h^2)^{-1/2}$ (see Table S1).

Hence, $\alpha_s$ depends exponentially on the activation energy of adsorption. A transmission coefficient $\kappa < 1$, will yield lower $\alpha_s$

values. If the TS is more constrained than the assumed 2D ideal gas, expressed by $\Delta S_{ads,g,act,m}^0 = R \ln\left(\frac{q_{TS,2D,m}^0}{e^{1/2} q_{g,m}^0}\right)$ (Eqs. (124)

and Supplement Eq. (197)), this will further lower $\alpha_s$.

## 6. Adsorption-Desorption Equilibrium

We consider equilibrium between adsorption and desorption and demonstrate that this results in the proper equilibrium

constants for gas adsorption into a 2D ideal gas and a 2D ideal lattice gas, proving that the CTST formulation of the rates leads

back to the equilibrium definition, from which we started off. We also show that this works both when using partition functions

and thermodynamic expressions. Hence, the derivations of all thermodynamic functions are internally consistent.

Considering the equilibrium, for the case that the adsorbed state is a 2D ideal gas, at low coverage:

$$\frac{R_{ads,2D}}{\mathcal{A}} = \frac{R_{des,2D}}{\mathcal{A}}$$

$$\kappa \left(\frac{k_B T}{h}\right) \frac{q'_{TS/\mathcal{A}}}{q_g/\mathcal{V}} e^{-\frac{E_b^0}{RT}} \mathcal{N}_g = \kappa \left(\frac{k_B T}{h}\right) \left(\frac{q'_{TS}}{q_{ads,2D}}\right) e^{-\frac{\left(E_{des}^0 + E_b^0\right)}{RT}} \mathcal{N}_{ads}$$

$$\frac{1/\mathcal{A}}{q_g/\mathcal{V}} \mathcal{N}_g = \left(\frac{1}{q_{ads,2D}}\right) e^{-\frac{E_{des}^0}{RT}} \mathcal{N}_{ads}$$

$$\frac{\mathcal{V}}{\mathcal{A}} \mathcal{N}_g = \left(\frac{q_g}{q_{ads,2D}}\right) e^{-\frac{E_{des}^0}{RT}} \mathcal{N}_{ads}$$

$$\frac{\mathcal{N}_{ads}}{\mathcal{N}_g} = \frac{q_{ads,2D}}{q_g} \frac{\mathcal{V}}{\mathcal{A}} e^{\frac{E_{des}^0}{RT}} = (2\pi m k_B T/h^2)^{-1/2} e^{\frac{E_{des}^0}{RT}} = K_{lin} . \tag{130}$$

This is the same result as given in Eq. (31) and consistent with the relation between $K_{lin}$ and the equilibrium constant.

Performing the same derivation starting with the thermodynamic expressions is given in the Supplement (Eqs. (216ff)).

For the case of the activity-based adsorption and desorption rates, we obtain

$$R_{ads,2D}^a = R_{des,2D}^a$$

$$\kappa \left(\frac{k_B T}{h}\right) \left(\frac{q'_{TS/\mathcal{A}}}{q_g/\mathcal{V}}\right) e^{-\frac{E_b^0}{RT}} \frac{\mathcal{A}}{\mathcal{V}} a_g \frac{\mathcal{V} \mathcal{A}_m^0}{\mathcal{A} \mathcal{V}_m^0} = \kappa \left(\frac{k_B T}{h}\right) \frac{q'_{TS}}{q_{ads,2D}} e^{-\frac{\left(E_{des}^0 + E_b^0\right)}{RT}} a_{ads,2D}$$

$$\frac{1/\mathcal{A}}{q_g/\mathcal{V}} a_g \frac{\mathcal{A}_m^0}{\mathcal{V}_m^0} = \left(\frac{1}{q_{ads,2D}}\right) e^{-\frac{E_{des}^0}{RT}} a_{ads,2D}$$

$$\frac{\mathcal{V}}{\mathcal{A}} a_g \frac{\mathcal{A}_m^0}{\mathcal{V}_m^0} = \left(\frac{q_g}{q_{ads,2D}}\right) e^{-\frac{E_{des}^0}{RT}} a_{ads,2D}$$

$$\frac{a_{ads,2D}}{a_g} = \frac{q_{ads,2D}}{q_g} \frac{\mathcal{V}}{\mathcal{A}} e^{\frac{E_{des}^0}{RT}} \frac{\mathcal{A}_m^0}{\mathcal{V}_m^0} = (2\pi m k_B T/h^2)^{-1/2} e^{\frac{E_{des}^0}{RT}} \frac{\mathcal{A}_m^0}{\mathcal{V}_m^0} = K_{lin} \frac{\mathcal{A}_m^0}{\mathcal{V}_m^0} . \tag{131}$$





This is the same result as in Eqs. (130). The derivation using the thermodynamic expressions is outlined in the Supplement Eqs. (221ff).

For the case when the adsorbed state on the surface is treated as a 2D ideal lattice gas using Eqs. (92), (102), and (121):

$$\frac{R_{ads,latt}}{\mathcal{A}} = \frac{R_{des,latt}}{\mathcal{A}}$$

$$\kappa\left(\frac{k_B T}{h}\right)\frac{q'_{TS}/\mathcal{A}}{q_g/\mathcal{V}}e^{-\frac{E_b^0}{RT}}\mathcal{N}_g = \kappa\left(\frac{k_B T}{h}\right)\frac{\left(\frac{q_{TS,m}^{0'}}{N_A}\right)}{(q_{ads,latt})}(N_{TS}/\mathcal{A})^0 e^{-\frac{\left(E_{des}^0+E_b^0\right)}{RT}}(\theta/(1-\theta)) ,$$

with $\left(\frac{q_{TS,m}^{0'}}{N_A}\right)(N_{TS}/\mathcal{A})^0 = \left(\frac{q_{TS,m}^{0'}}{N_A}\right)\frac{N_A}{\mathcal{A}_m^0} = \frac{q_{TS,m}^{0'}}{\mathcal{A}_m^0} = \frac{q'_{TS}}{\mathcal{A}}$ , we obtain

$$\kappa\left(\frac{k_B T}{h}\right)\frac{q'_{TS}/\mathcal{A}}{q_g/\mathcal{V}}e^{-\frac{E_b^0}{RT}}\mathcal{N}_g = \kappa\left(\frac{k_B T}{h}\right)\frac{(q'_{TS}/\mathcal{A})}{q_{ads,latt}}e^{-\frac{\left(E_{des}^0+E_b^0\right)}{RT}}(\theta/(1-\theta))$$

$$\frac{1}{q_g/\mathcal{V}}\mathcal{N}_g = \frac{1}{q_{ads,latt}}e^{-\frac{E_{des}^0}{RT}}(\theta/(1-\theta))$$

$$\frac{(\theta/(1-\theta))}{\mathcal{N}_g} = \frac{q_{ads,latt}}{q_g/\mathcal{V}}e^{\frac{E_{des}^0}{RT}} = (2\pi m k_B T/h^2)^{-3/2}e^{\frac{E_{des}^0}{RT}} = K_{Lang} \qquad (132)$$

This is the expected result outlined in Eq. (44). The derivation starting with the thermodynamic expressions is given in the Supplement Eqs. (226ff).

For the case of the activity-based adsorption (Eq. (127)) and desorption rates (Eq. (86), we obtain

$$R^a_{ads,latt} = R^a_{des,latt}$$

$$\kappa\left(\frac{k_B T}{h}\right)\left(\frac{q'_{TS}/\mathcal{A}}{q_g/\mathcal{V}}\right)e^{-\frac{E_b^0}{RT}}\frac{1}{\mathcal{N}_{ads,max}}a_g\frac{(N_g/\mathcal{V})^0(1-\theta^0)}{\theta^0} = \kappa\left(\frac{k_B T}{h}\right)\frac{(q'_{TS}/\mathcal{A})}{q_{ads,latt}\mathcal{N}_{ads,max}}e^{-\frac{\left(E_{des}^0+E_b^0\right)}{RT}}a_{ads,latt}$$

$$\frac{1}{q_g/\mathcal{V}}a_g\frac{(N_g/\mathcal{V})^0(1-\theta^0)}{\theta^0} = \frac{1}{q_{ads,latt}}e^{-\frac{E_{des}^0}{RT}}a_{ads,latt}$$

$$\frac{a_{ads,latt}}{a_g} = \frac{q_{ads,latt}}{q_g/\mathcal{V}}e^{\frac{E_{des}^0}{RT}}\frac{(N_g/\mathcal{V})^0(1-\theta^0)}{\theta^0}\frac{a_{ads,latt}}{a_g}$$

$$\frac{a_{ads,latt}}{a_g} = (2\pi m k_B T/h^2)^{-3/2}e^{\frac{E_{des}^0}{RT}}\frac{(N_g/\mathcal{V})^0(1-\theta^0)}{\theta^0} = K_{Lang}\frac{(N_g/\mathcal{V})^0(1-\theta^0)}{\theta^0}$$

$$\frac{a_{ads,latt}}{a_g}\frac{\theta^0}{(N_g/\mathcal{V})^0(1-\theta^0)} = \frac{(\theta/(1-\theta))}{\mathcal{N}_g} = K_{Lang} . \qquad (133)$$

This results in the same relationship as in Eq. (132). The derivation starting with the thermodynamic expressions is given in
the Supplement Eqs. (231ff). Thus, equating the adsorption and desorption rates, both derived based on TS theory, correctly reproduces the corresponding equilibrium constant.




## 7. Derivation of Kinetic Parameters from the Equilibrium Constants

In previous studies (Bartels-Rausch et al., 2005;Tabazadeh and Turco, 1993) equilibrium thermodynamic data or equilibrium coverage data have been used to constrain kinetic parameters of either adsorption or desorption. If $K^0_{ads,g,latt}$ or $K^0_{ads,g,2D}$ are

known as a function of temperature from measurements or extracted from fundamental thermodynamic data, the Arrhenius plot of its temperature dependence delivers $\Delta H^0_{ads,g,2D}$ or $\Delta H^0_{des,2D,g}$ as a slope and $\Delta S^0_{ads,g,2D}$ or $\Delta S^0_{des,2D,d}$ as an offset.

For the case of an adsorbed 2D ideal gas, we can derive the pre-exponential factor from equilibrium, $R^a_{ads,2D} = R^a_{des,2D}$, starting off with the molecular descriptions of respective rates (Eq. (131)). In addition, we make use of $\alpha_s$ and its relationship to microscopic properties (Eq. (129)) and the definition of $A_{des,2D}$ obtained from the derivation of the desorption rate (Eq.

(78)). By applying the thermodynamic equilibrium constant, we can then relate the microscopic picture to thermodynamic functions, obtaining $A_{des,2D}$ under equilibrium conditions:

$$\kappa\left(\frac{k_BT}{h}\right)\left(\frac{q'_{TS}/\mathcal{A}}{q_g/\mathcal{V}}\right)e^{-\frac{E^0_b}{RT}}\frac{\mathcal{A}}{\mathcal{V}}a_g\frac{\mathcal{V}\mathcal{A}^0_m}{\mathcal{A}\mathcal{V}^0_m} = \kappa\left(\frac{k_BT}{h}\right)\frac{q'_{TS}}{q_{ads,2D}}e^{-\frac{\left(E^0_{des}+E^0_b\right)}{RT}}a_{ads,2D}$$

$$\left(\frac{k_BT}{h}\right)\alpha_s a_g\frac{\mathcal{A}^0_m}{\mathcal{V}^0_m}(2\pi mk_BT/h^2)^{-1/2} = A_{des,2D}e^{-\frac{\left(E^0_{des}+E^0_b\right)}{RT}}a_{ads,2D}$$

$$A_{des,2D} = \left(\frac{k_BT}{h}\right)\alpha_s\frac{\mathcal{A}^0_m}{\mathcal{V}^0_m}(2\pi mk_BT/h^2)^{-1/2}e^{\frac{\left(E^0_{des}+E^0_b\right)}{RT}}\frac{a_g}{a_{ads,2D}}$$

$$A_{des,2D} = \left(\frac{k_BT}{h}\right)\alpha_s\frac{\mathcal{A}^0_m}{\mathcal{V}^0_m}(2\pi mk_BT/h^2)^{-1/2}e^{\frac{\left(E^0_{des}+E^0_b\right)}{RT}}K^0_{des,2D,g}$$

$$A_{des,2D} = \left(\frac{k_BT}{h}\right)\alpha_s\frac{\mathcal{A}^0_m}{\mathcal{V}^0_m}(2\pi mk_BT/h^2)^{-1/2}e^{\frac{\left(E^0_{des}+E^0_b\right)}{RT}}e^{-\Delta G^0_{des,2D,g,m}/RT}$$

$$A_{des,2D} = \left(\frac{k_BT}{h}\right)\alpha_s\frac{\mathcal{A}^0_m}{\mathcal{V}^0_m}(2\pi mk_BT/h^2)^{-1/2}e^{\frac{\left(E^0_{des}+E^0_b\right)}{RT}}e^{-(\Delta H^0_{des,2D,g,m}-T\Delta S^0_{des,2D,g,m})/RT}$$

$$A_{des,2D} = \left(\frac{k_BT}{h}\right)\alpha_s\frac{\mathcal{A}^0_m}{\mathcal{V}^0_m}(2\pi mk_BT/h^2)^{-1/2}e^{\frac{\left(E^0_{des}+E^0_b\right)}{RT}}e^{-(\frac{1}{2}RT+E^0_{des})/RT}e^{\Delta S^0_{des,2D,g,m}/R}$$

$$A_{des,2D} = \left(\frac{k_BT}{h}\right)\alpha_s\frac{\mathcal{A}^0_m}{\mathcal{V}^0_m}(2\pi mk_BT/h^2)^{-1/2}e^{-1/2}e^{\frac{E^0_b}{RT}}e^{\Delta S^0_{des,2D,g,m}/R} \ . \tag{134}$$

If the activation barrier $E^0_b$ is negligible, this simplifies to

$$A_{des,2D} = \alpha_s e^{\Delta S^0_{des,2D,g,m}/R}e^{-1/2}\left(\frac{\mathcal{A}^0_m}{\mathcal{V}^0_m}\right)\left(\frac{k_BT}{h}\right)(2\pi mk_BT/h^2)^{-1/2} \ . \tag{135}$$

$A_{des,2D}$ derived from equilibrium is the same result as for $A_{des,2D}$ derived from desorption using TS theory (Eq. (78)). Thus, the pre-exponential factor of desorption can be calculated from the desorption entropy ($\Delta S^0_{des,2D,g,m}$) and from the known value of $\alpha_s$ but only if the standard state, which has been used to obtain the entropy, is known.



For the case of an adsorbed 2D ideal lattice gas, we can derive the pre-exponential factor from equilibrium, $R^a_{ads,latt} = R^a_{des,latt}$ (Eq. (133)), in a similar ways as for the 2D ideal gas discussed above, using $A_{des,latt}$ from the derivation of the desorption rate (Eq. (97)) and $\alpha_s$ (Eq. (129)) as

$$\kappa\left(\frac{k_BT}{h}\right)\left(\frac{q'_{TS}/\mathcal{A}}{q_g/\mathcal{V}}\right)e^{-\frac{E^0_b}{RT}}\frac{1}{\mathcal{N}_{ads,max}}a_g\frac{(N_g/\mathcal{V})^0(1-\theta^0)}{\theta^0} = \kappa\left(\frac{k_BT}{h}\right)\frac{(q'_{TS}/\mathcal{A})}{q_{ads,latt}\mathcal{N}_{ads,max}}e^{-\frac{(E^0_{des}+E^0_b)}{RT}}a_{ads,latt}$$

$$\kappa\left(\frac{k_BT}{h}\right)\left(\frac{q'_{TS}/\mathcal{A}}{q_g/\mathcal{V}}\right)e^{-\frac{E^0_b}{RT}}\frac{1}{\mathcal{N}_{ads,max}}a_g\frac{(N_g/\mathcal{V})^0(1-\theta^0)}{\theta^0} = A^a_{des,latt}e^{-\frac{(E^0_{des}+E^0_b)}{RT}}a_{ads,latt}$$

$$\alpha_s\left(\frac{k_BT}{h}\right)(2\pi mk_BT/h^2)^{-1/2}\frac{1}{\mathcal{N}_{ads,max}}a_g\frac{(N_g/\mathcal{V})^0(1-\theta^0)}{\theta^0} = A^a_{des,latt}e^{-\frac{(E^0_{des}+E^0_b)}{RT}}a_{ads,latt}$$

$$A^a_{des,latt} = \alpha_s\left(\frac{k_BT}{h}\right)(2\pi mk_BT/h^2)^{-1/2}\frac{1}{\mathcal{N}_{ads,max}}\frac{(N_g/\mathcal{V})^0(1-\theta^0)}{\theta^0}e^{\frac{(E^0_{des}+E^0_b)}{RT}}\frac{a_g}{a_{ads,latt}}$$

$$A^a_{des,latt} = \alpha_s\left(\frac{k_BT}{h}\right)(2\pi mk_BT/h^2)^{-1/2}\frac{1}{\mathcal{N}_{ads,max}}\frac{(N_g/\mathcal{V})^0(1-\theta^0)}{\theta^0}e^{\frac{(E^0_{des}+E^0_b)}{RT}}K^0_{des,latt,g}$$

$$A^a_{des,latt} = \alpha_s\left(\frac{k_BT}{h}\right)(2\pi mk_BT/h^2)^{-1/2}\frac{1}{\mathcal{N}_{ads,max}}\frac{(N_g/\mathcal{V})^0(1-\theta^0)}{\theta^0}e^{\frac{(E^0_{des}+E^0_b)}{RT}}e^{-\frac{\Delta G^0_{des,latt,g,m}}{RT}}$$

$$A^a_{des,latt} = \alpha_s\left(\frac{k_BT}{h}\right)(2\pi mk_BT/h^2)^{-1/2}\frac{1}{\mathcal{N}_{ads,max}}\frac{(N_g/\mathcal{V})^0(1-\theta^0)}{\theta^0}e^{\frac{(E^0_{des}+E^0_b)}{RT}}e^{-\frac{\Delta H^0_{des,latt,g,m}-T\Delta S^0_{des,latt,g,m}}{RT}}$$

$$A^a_{des,latt} = \alpha_s\left(\frac{k_BT}{h}\right)(2\pi mk_BT/h^2)^{-1/2}\frac{1}{\mathcal{N}_{ads,max}}\frac{(N_g/\mathcal{V})^0(1-\theta^0)}{\theta^0}e^{\frac{(E^0_{des}+E^0_b)}{RT}}e^{-\frac{\left(\frac{5}{2}RT+E^0_{des}+RT\frac{\ln(1-\theta^0)}{\theta^0}\right)}{RT}}e^{\frac{\Delta S^0_{des,latt,g,m}}{R}}$$

$$A^a_{des,latt} = \alpha_s\left(\frac{k_BT}{h}\right)(2\pi mk_BT/h^2)^{-1/2}\frac{1}{\mathcal{N}_{ads,max}}\frac{(N_g/\mathcal{V})^0(1-\theta^0)}{\theta^0}e^{-\frac{5}{2}}(1-\theta^0)^{-\frac{1}{\theta^0}}e^{\frac{E^0_b}{RT}}e^{\frac{\Delta S^0_{des,latt,g,m}}{R}}. \tag{136}$$

If the activation barrier $E^0_b$ is negligible, this simplifies to

$$A^a_{des,latt} = \alpha_s e^{\Delta S^0_{des,latt,g,m}/R}e^{-5/2}\frac{(1-\theta^0)}{\theta^0}\left(\frac{1}{\mathcal{N}_{ads,max}}\right)\left(\frac{N_A}{\mathcal{V}^0_m}\right)(1-\theta^0)^{-1/\theta^0}\left(\frac{k_BT}{h}\right)(2\pi mk_BT/h^2)^{-1/2}. \tag{137}$$

$A^a_{des,latt}$ derived from equilibrium is the same result as for $A^a_{des,latt}$ derived from desorption using TS theory (Eq. (97)).

As can be seen, the activity-based $A^a_{des,latt}$ does not depend on the surface coverage. However, the standard surface coverage $\theta^0$, for which $\Delta S^0_{des,latt,g,m}$ has been derived, must be known (similar to the case described in Eq. 97). Hence, the pre-exponential factor $A_{des,latt}$ has a strong non-linear dependence on the standard surface coverage. When the underlying standard surface coverages are not known, additional uncertainties are introduced. When deriving the desorption rate (Eq. 82), the dependence on surface coverage is accounted for.





**8. Implications for the Assessment of Desorption Energy and Rate and Pre-exponential Factor**

The thermodynamic derivations above indicate that the underlying adsorption model, i.e., 2D ideal gas or 2D ideal lattice gas, will have a significant impact on desorption rates and the pre-exponential factor and, thus, on the evaluation of $E_{des}^0$ and $\tau_d$. This is particularly important for the case of the 2D ideal lattice gas model for which the desorption rate varies non-linearly with surface coverage, i.e., proportional to $(\theta/(1-\theta))$ (Eq. 82) since the surface activity is defined by $\frac{(\theta/(1-\theta))}{(\theta^0/(1-\theta^0))}$ (Eq. 42).

This implies that for same $E_{des}^0$, $\frac{R_{des}}{\mathcal{A}}$ can vary significantly depending on adsorbate coverage. Vice versa, if the coverage is not well-known, derivation of $E_{des}^0$ from measured $\frac{R_{des}}{\mathcal{A}}$ is associated with large uncertainties.

Figure 5 displays the variation of $\frac{R_{des}}{\mathcal{A}}$ for different $\theta$, covering a pristine surface to a fully occupied surface. As discussion of Fig. 7 alluded to (above), Fig. 5 demonstrates that the assumption of the underlying substrate model significantly impacts $\frac{R_{des}}{\mathcal{A}}$. The differences in $\frac{R_{des}}{\mathcal{A}}$ when applying a 2D ideal gas or a 2D ideal lattice gas are about 3 to 6 orders of magnitude over

a typical $\theta$ range. Furthermore, variation of $\frac{R_{des}}{\mathcal{A}}$ for the 2D ideal lattice gas is greater with $\theta$ due to its non-linear dependence on $\theta$. Figure 5 implies that the different sensitivities of the two adsorbate models on surface coverages can result in large differences in experimentally derived desorption rates besides uncertainties in the pre-exponential factor and $E_{des}^0$.

As outlined above, Fig. 10 highlights how the underlying adsorbate model impacts the pre-exponential factor. If the actual adsorbate system more closely behaves as a 2D ideal lattice gas but is analyzed assuming a 2D ideal gas, significant

uncertainties in $A_{des}$ can arise which, in turn, increase the uncertainty in the derivation of $E_{des}^0$ and estimation of the desorption lifetime.

Figure 14 presents estimates of $\tau_d$ for given $E_{des}^0$ as a function of temperature when applying a 2D ideal gas and 2D ideal lattice gas adsorbate model. For both adsorbate models, the temperature sensitivity of $\tau_d$ increases with increasing $E_{des}^0$. For given $\tau_d$ the difference in $E_{des}^0$, when applying the different adsorbate models can range from 10 to 15 kJ mol[-1], where larger

differences occur at higher temperatures. Hence, when deriving $E_{des}^0$ from $\tau_d$ values, in absence of knowledge of the underlying adsorbate model, $E_{des}^0$ is likely uncertain by 10 to 15 kJ mol[-1]. Vice versa, the corresponding uncertainty in $\tau_d$ is up to about 3 orders of magnitude. As outlined in the introduction, for experimental studies where $\tau_d$ is coupled to the surface reaction rate, the first-order surface reaction rate could also be uncertain by up to 3 orders of magnitude.

Figure 15 displays $E_{des}^0$ values derived from a variation of desorption rates applying a 2D ideal gas or 2D ideal lattice gas

adsorbate model as a function of surface coverage $\theta$. For example, for $\frac{R_{des}}{\mathcal{A}} = 1$ m[-2] s[-1], reflected by the uppermost red and blue curves, it is evident that the chosen adsorbate model results in significantly different $E_{des}^0$ values differing by at least 20 kJ mol[-1]. These results further support the importance of accurate knowledge of $\theta$. The $E_{des}^0$ values can vary by tens of kJ mol[-1], if $\theta$ is incorrectly determined or assumed. For example, if the substrate surface is assumed to be pristine but in fact $\theta = 0.2$, $E_{des}^0$ can be overestimated by $10 - 20$ kJ mol[-1].





Figures 5, 10, 14, and 15 highlight the potential uncertainties that arise by choice of the absorbate models for derivation

of $E_{des}^0$. In addition to those uncertainties, standard states applied in adsorption and desorption studies are often not known or

well documented. This can lead to additional uncertainties as also outlined above. $A_{des}$ values shown in Fig. 10 will be the

same for different choices of standard states as long as they have been consistently applied to the entropic contributions

$\Delta S_{des,2D,act,m}^0$, $\Delta S_{des,2D,g,m}^0$, $\Delta S_{des,latt,act,m}^0$, and $\Delta S_{des,latt,g,m}^0$ (Supplement Eqs. (150), (121), (173), and (134), respectively)

and within the equations for $A_{des}$ (Eqs. (87), (134), (97), (136)). The standard molar volume, $\mathcal{V}_m^0 = 24.8$ L mol$^{-1}$ at 298 K and

1000 hPa, is the typically applied parameter but one has to make sure to adjust this value to observational conditions, i.e.,

temperature and pressure, for both the entropic contribution and the derivation of the partition functions. The latter depends

linearly on the molar volume (Supplement Eqs. (7) and (92)). The actual surface coverage and applied standard surface

coverages are often less clear and furthermore, different standard states may have been chosen for the entropic contributions

and experimental conditions. To further complicate matters, standard surface coverages can be defined applying $\theta^0 = 0.012$ or

0.5, which both have their advantages as outlined above. If the standard surface coverage for the entropic contribution is based

on $\theta^0 = 0.012$ but the remainder of thermodynamic functions on $\theta^0 = 0.5$, $A_{des}$ will be erroneous and thus $E_{des}^0$ and $\tau_d$.

## 9. Conclusions

Reversible adsorption is a key process for any gas-condensed phase interaction, and particularly important when environmental

interfaces are involved including aerosol particles. This study provides a comprehensive treatment of the classic and statistical

thermodynamics of the adsorption and desorption process considering transition state theory for two typically applied

adsorbate models, the 2D ideal gas and the 2D ideal lattice gas which apply to solid or liquid substrate surfaces. We established

thermodynamic and microscopic relationships for adsorption and desorption equilibrium constants, adsorption and desorption

rates, first-order adsorption and desorption rate coefficients, and corresponding pre-exponential factors. These derivations

allow the interpretation of thermodynamic functions such as equilibrium constants in terms of their molecular properties, as

well as the calculation of explicit numeric expressions for the latter. This exercise demonstrates the importance of applied

assumptions of adsorbate model and standard states when analyzing and interpreting adsorption and desorption processes, the

latter being often ill-defined in experimental studies (Donaldson et al., 2012). The derivations allow for a microscopic

interpretation of the surface accommodation coefficient including the entropic contribution. Our treatment demonstrates that

the pre-exponential factor, when deriving the desorption lifetime from the desorption energy, can differ by orders of magnitude

depending on the choice of adsorbate model. Clearly, such a difference yields similar effects on the desorption lifetime, and

when used to estimate desorption energies (e.g., from interfacial residence times estimated from molecular dynamics

simulations or from measured desorption rates) significant uncertainties in the desorption energy are incurred. Furthermore,

uncertainties in surface coverage and assumptions in standard surface coverage can lead to significant changes in desorption

rates and thus in evaluated desorption energies for the rather common case of a 2D ideal lattice gas. The objective of providing





this comprehensive thermodynamic and microscopic treatment of the adsorption and desorption processes is to guide the theoretical and experimental assessments of adsorption and desorption rates, desorption energies and choice of standard states with implications for the corresponding desorption lifetimes. This in turn will improve, specifically, the analyses and interpretation of surface layer reaction rates and surface-to-bulk transport, and thus, bulk mass accommodation. More

generally, this provides a better basis for the prediction of gas-particle partitioning, multiphase chemical reactions, and the chemical evolution of atmospheric aerosol.


**Data availability.** All data needed to draw the conclusions in the present study are shown in the paper and/or the Supplement.

**Supplement.** The supplement related to this article is available online at:

**Author contributions.** DAK and MA envisioned this study and wrote this manuscript.

**Competing interests.** The authors are members of the editorial board of Atmospheric Chemistry and Physics. The peer review process was guided by an independent editor. The authors have no other competing interests to declare.

**Acknowledgements.** DAK acknowledges support from the National Science Foundation. MA appreciates support by the Swiss National Science Foundation.

**Financial support**. This research has been supported by the National Science Foundation, Division of Atmospheric and Geospace Sciences (grant no. AGS-1446286) and by the Swiss National Science Foundation (grant no 188662)




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



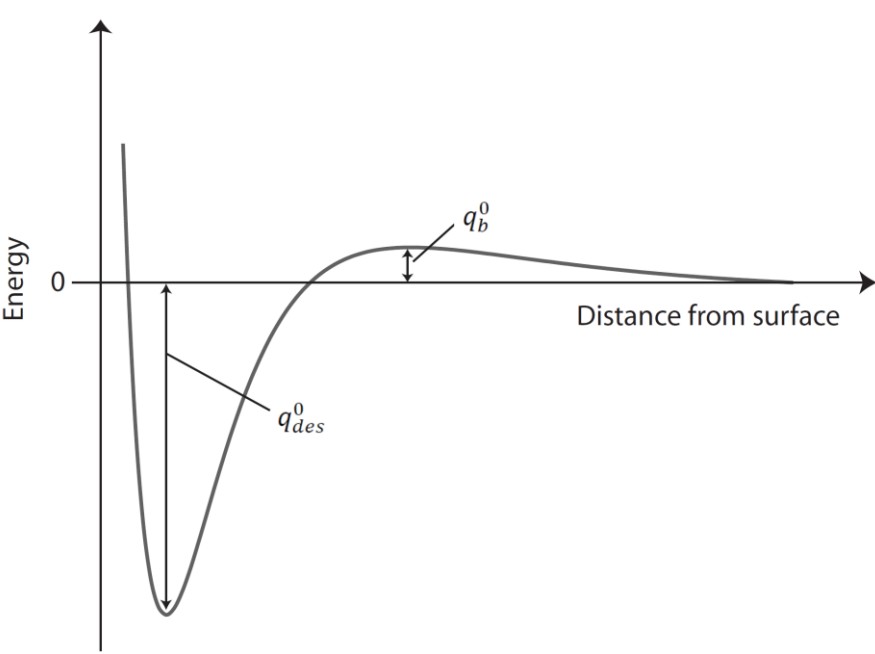


**Figure 1. Potential energy curve for adsorption and desorption processes expressed by the heat of desorption, $q_{des}^0$. For activated adsorption and desorption processes an additional energy barrier, expressed by $q_b^0$, must be overcome.**






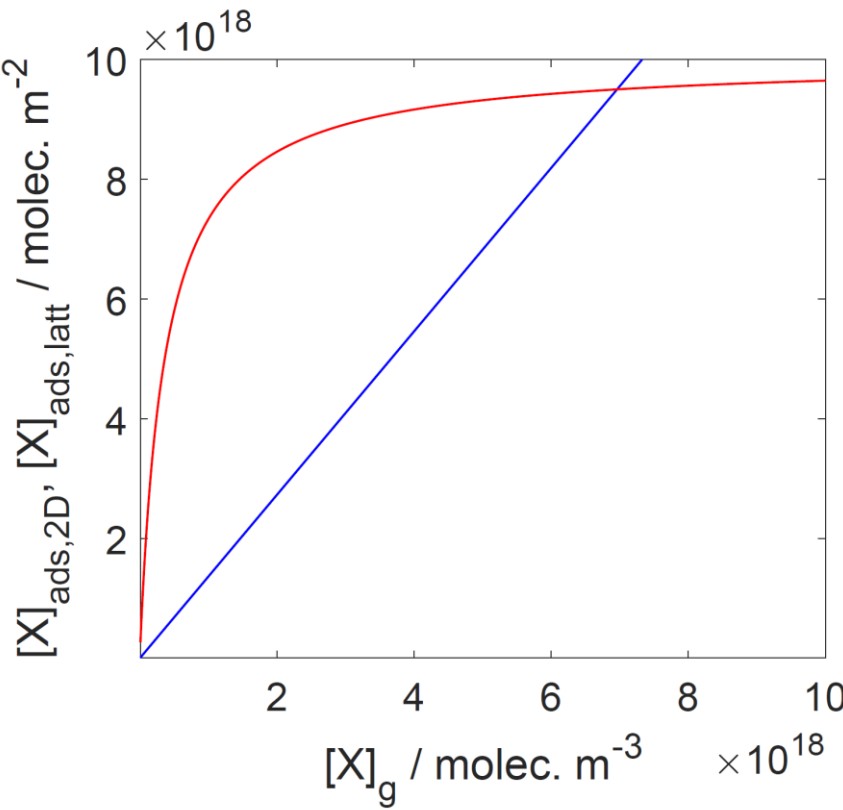

**Figure 2. Equilibrium adsorbate surface concentration as a function of gas phase concentration for the case of a 2D ideal gas (blue line) and 2D ideal lattice gas (red line). Applied $E_{des}^0$ are 63 kJ mol$^{-1}$ and 88 kJ mol$^{-1}$, respectively. We assume a desorption process without additional barrier, $E_b^0 = 0$. Thermodynamic quantities for calculation are given in Table S1.**







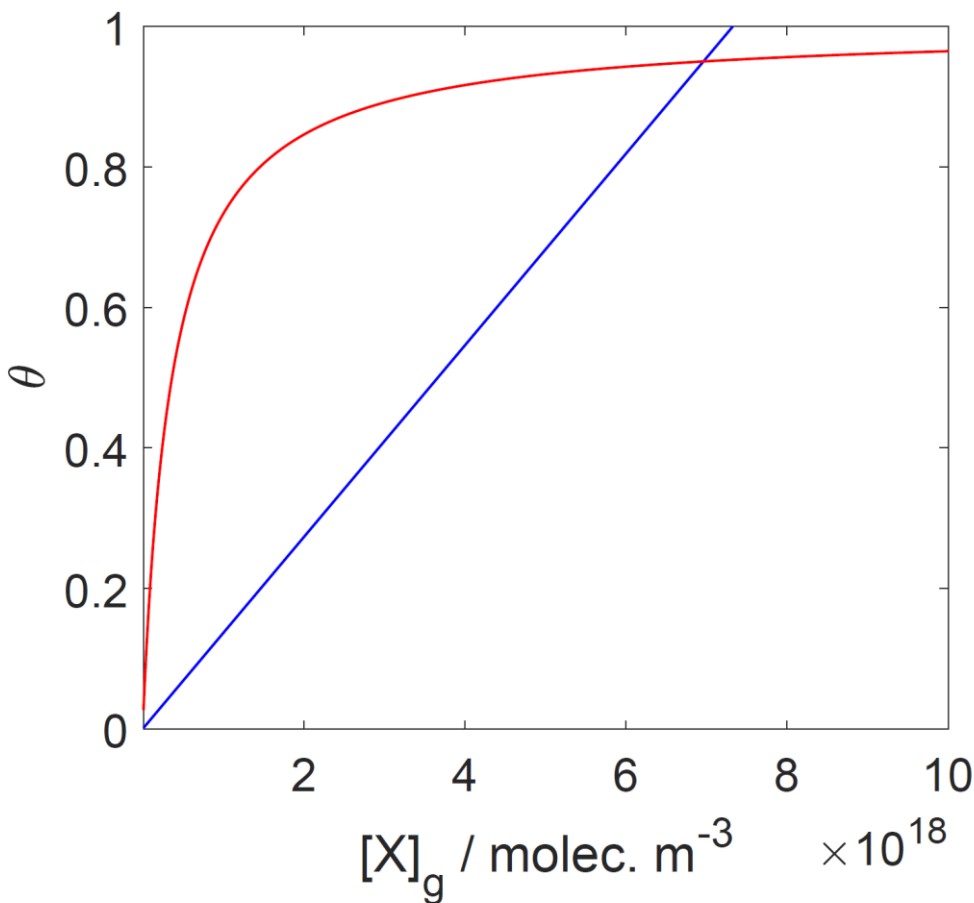

**Figure 3. Equilibrium surface coverage as a function of gas phase concentration for the case of a 2D ideal gas (blue line) and 2D ideal lattice gas (red line). The data are the same as used for derivation of Fig. 2, but surface coverages are derived by normalization with maximum number of adsorption sites. Thermodynamic quantities and standard states necessary for calculation are given in Table S1.**





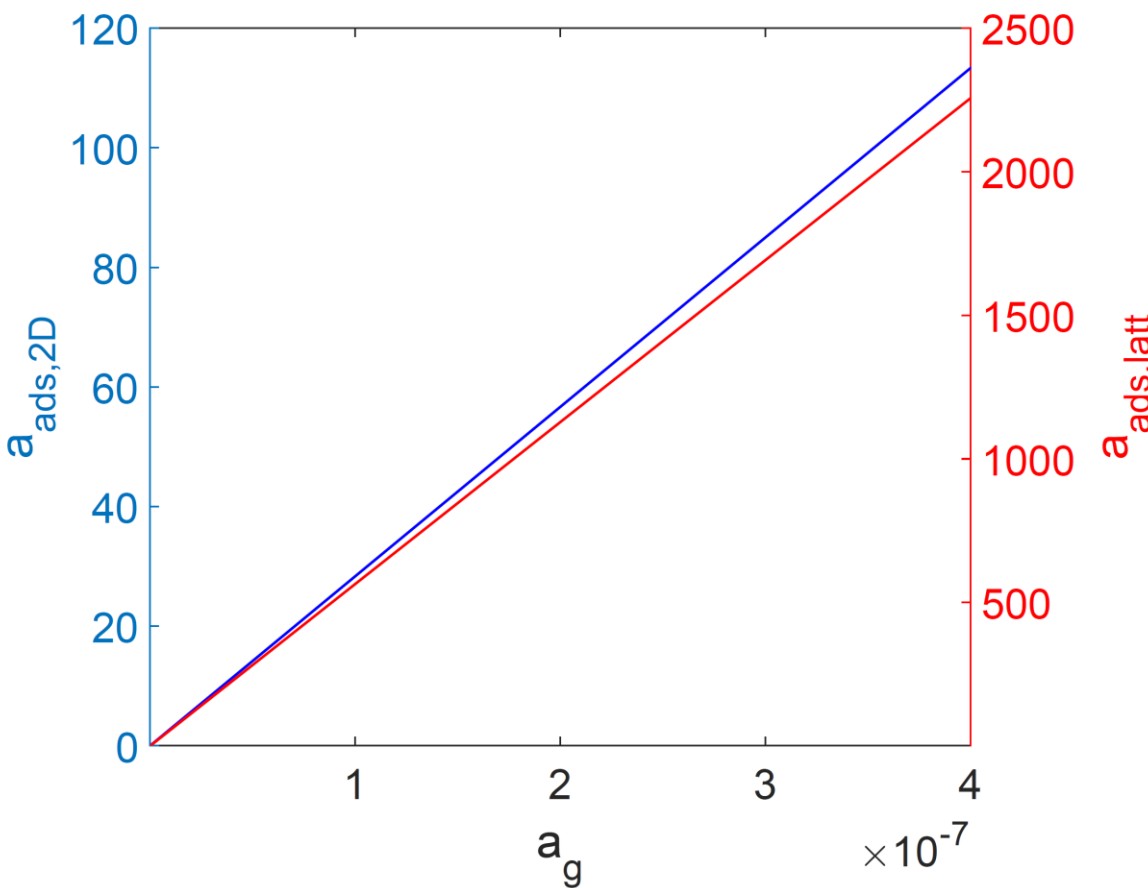

**Figure 4. Equilibrium surface activity as a function of gas phase activity for the case of a 2D ideal gas (blue line) and 2D ideal lattice gas (red line). The data are the same as used for derivation of Fig. 2. Thermodynamic quantities and standard states necessary for calculation are given in Table S1.**

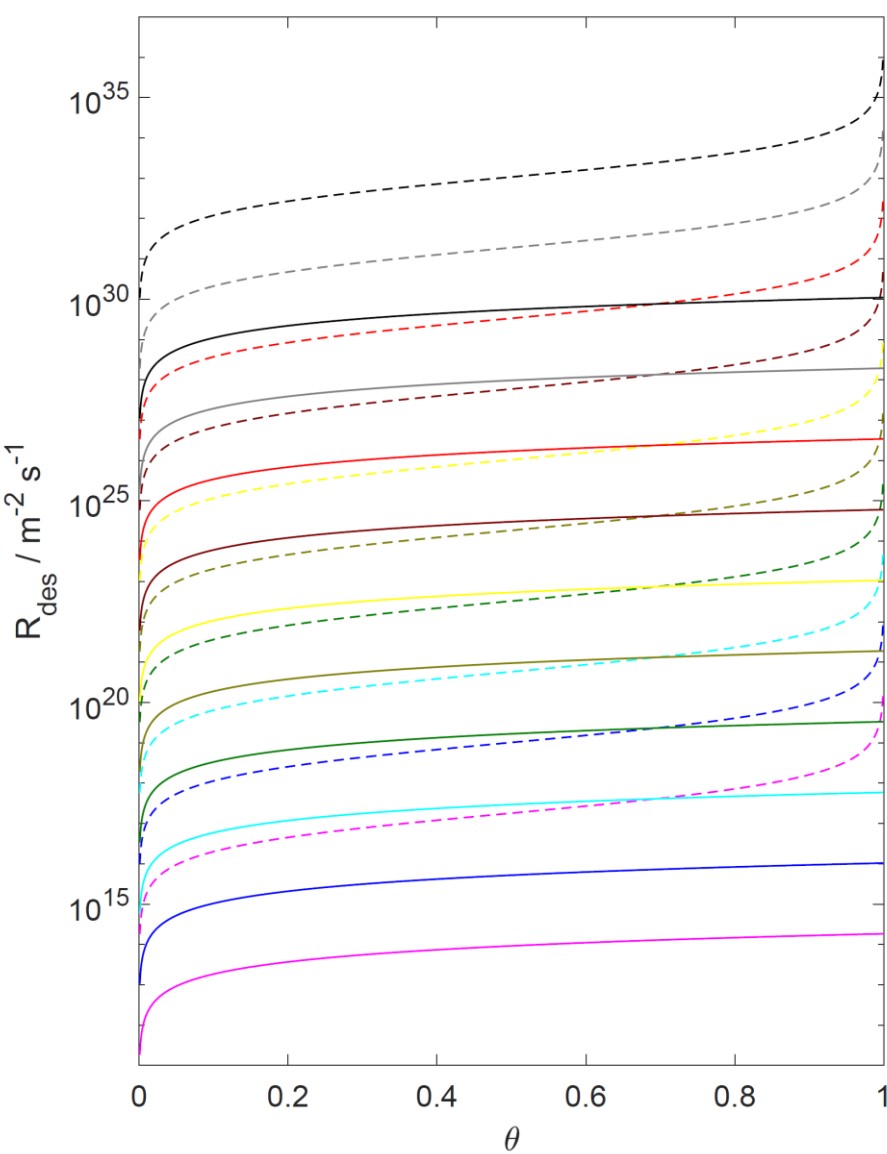

1110    **Figure 5. The change in the desorption rate for the assumption of a 2D ideal gas (solid lines) and 2D ideal lattice gas (dashed lines) are plotted as a function of adsorbate fractional surface coverage θ and variation of $E_{des}^0$ from 100 (bottom) to 10 kJ mol$^{-1}$ (top). We assume a desorption process without additional barrier, $E_b^0 = 0$. Thermodynamic quantities and standard states necessary for calculation are given in Table S1.**

1115





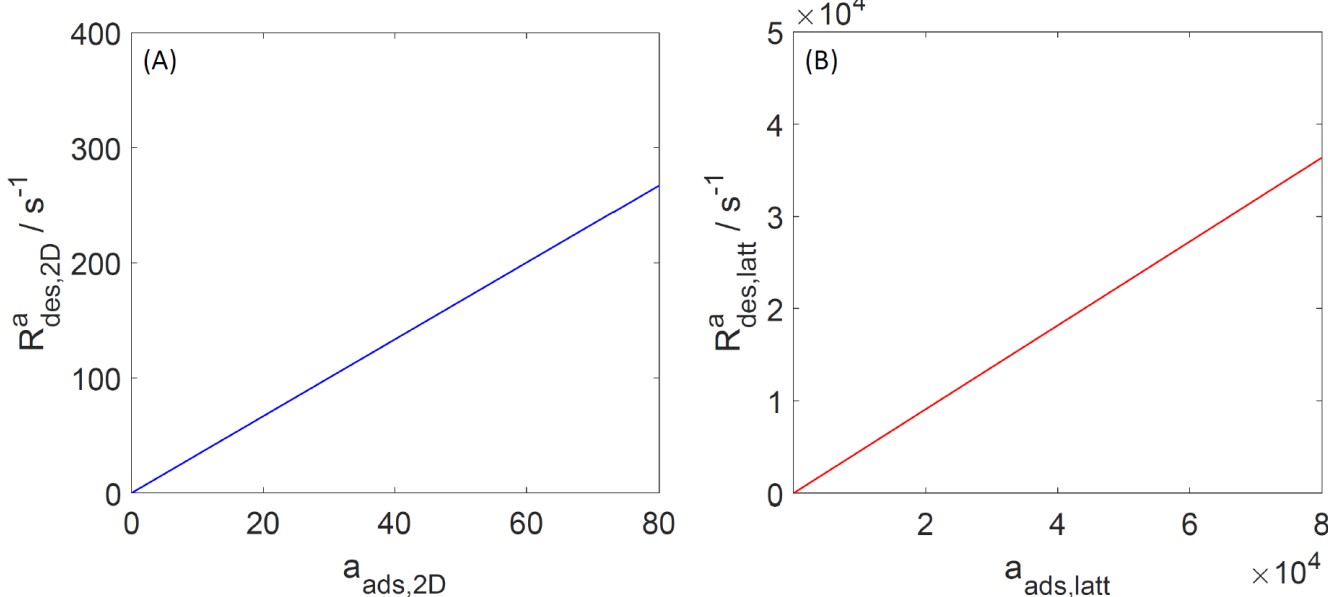

**Figure 6. The activity-based desorption rate for the case of a 2D ideal gas (blue line, A) and 2D ideal lattice gas (red line, B). Applied $E_{des}^0$ are 70 kJ mol⁻¹ and 92 kJ mol⁻¹, respectively. We assume a desorption process without additional barrier, $E_b^0 = 0$. Thermodynamic quantities and standard states necessary for calculation are given in Table S1.**

1145

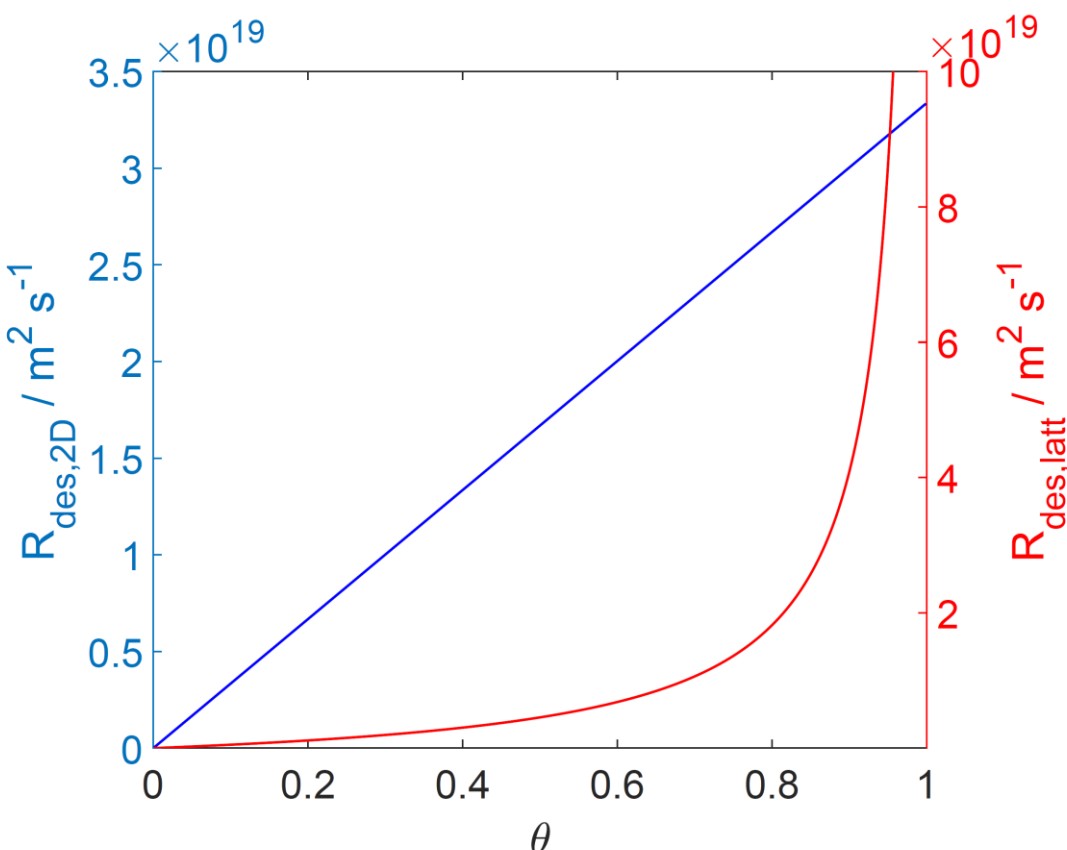

**Figure 7. The desorption rate for the case of a 2D ideal gas (blue line) and 2D ideal lattice gas (red line). Applied $E_{des}^0$ are 63 kJ mol$^{-1}$ and 88 kJ mol$^{-1}$, respectively. We assume a desorption process without additional barrier, $E_b^0 = 0$. Thermodynamic quantities and standard states necessary for calculation are given in Table S1.**





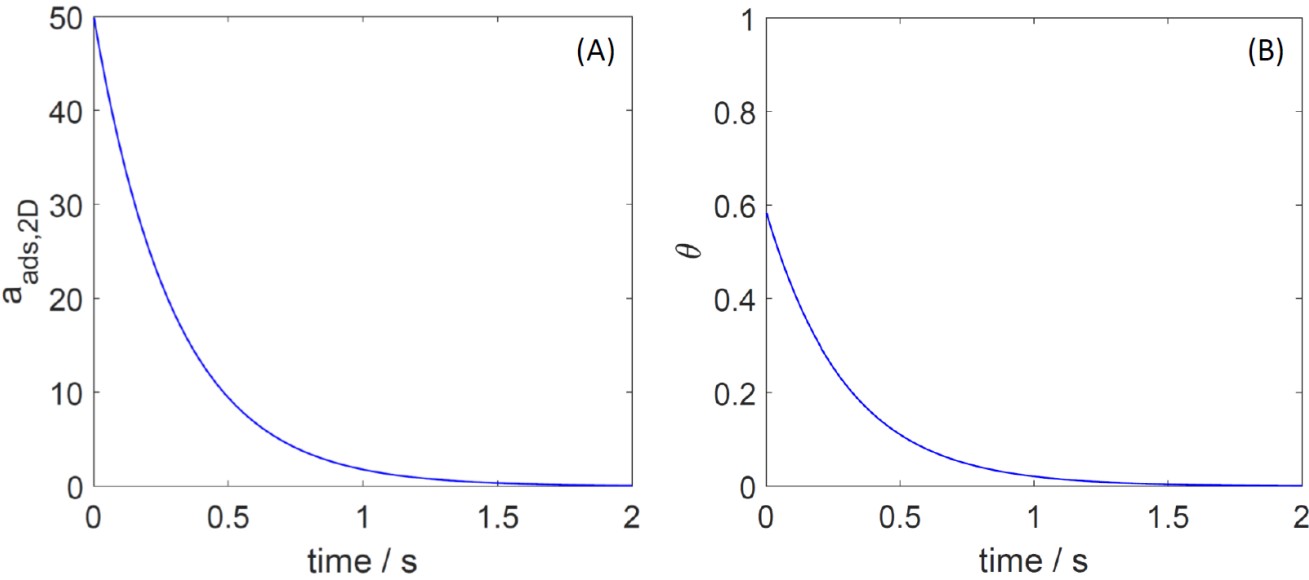

**Figure 8.** The decay of surface activity (A) and surface coverage (B) of the 2D ideal gas adsorbate as a function of time due to desorption. The applied $E^0_{des}$ is 70 kJ mol$^{-1}$ and the initial surface activity is 50. We assume a desorption process without additional barrier, $E^0_b = 0$. Thermodynamic quantities and standard states necessary for calculation are given in Table S1.





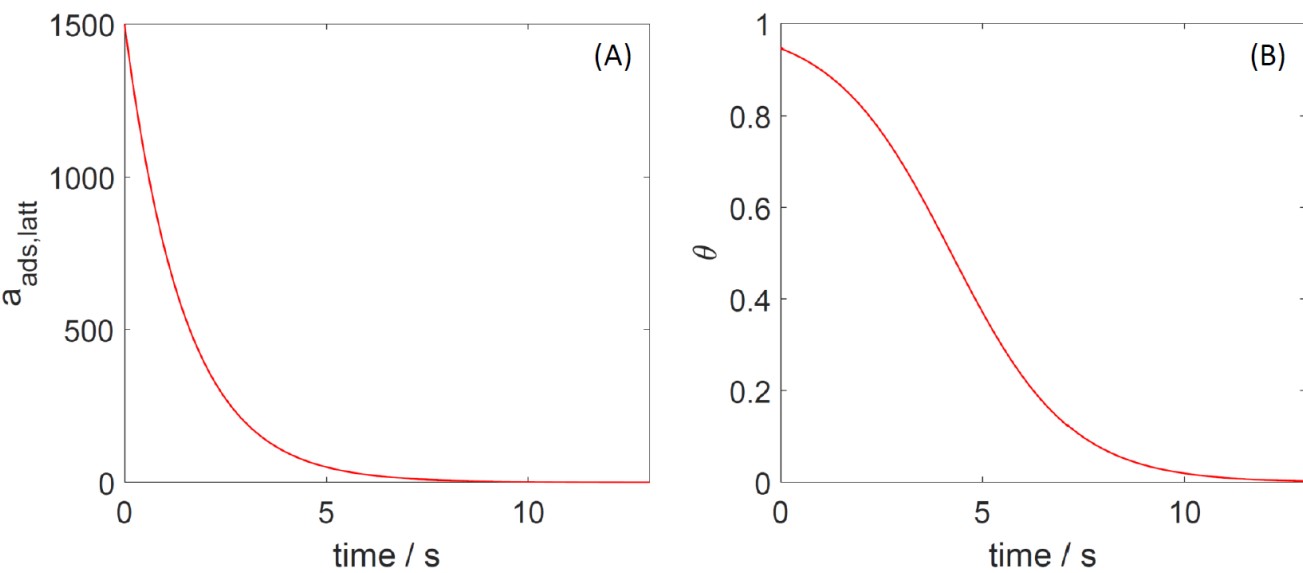


**Figure 9. The decay of surface activity (A) and surface coverage (B) of the 2D ideal lattice gas adsorbate as a function of time due to desorption. The applied $E_{des}^0$ is 91 kJ mol$^{-1}$ and the initial surface activity is 1500. We assume a desorption process without additional barrier, $E_b^0 = 0$. Thermodynamic quantities and standard states necessary for calculation are given in Table S1.**






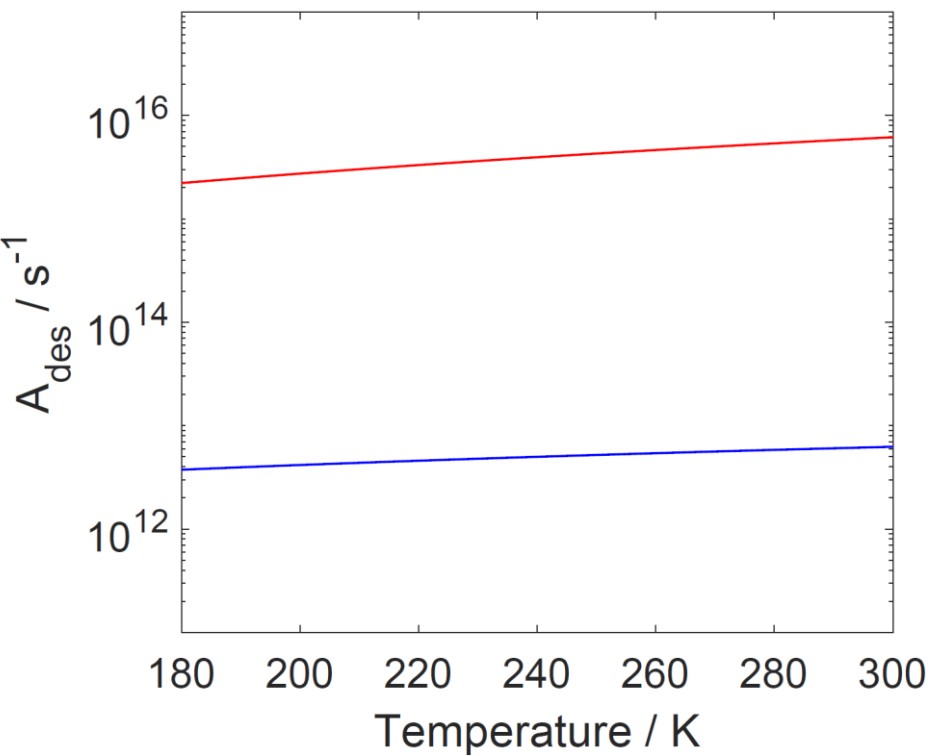

**Figure 10.** The pre-exponential factor $A_{des}$ as a function of temperature is plotted for the case of a 2D ideal gas (blue) and a 2D ideal lattice gas (red). Thermodynamic quantities and standard states necessary for calculation are given in Table S1.





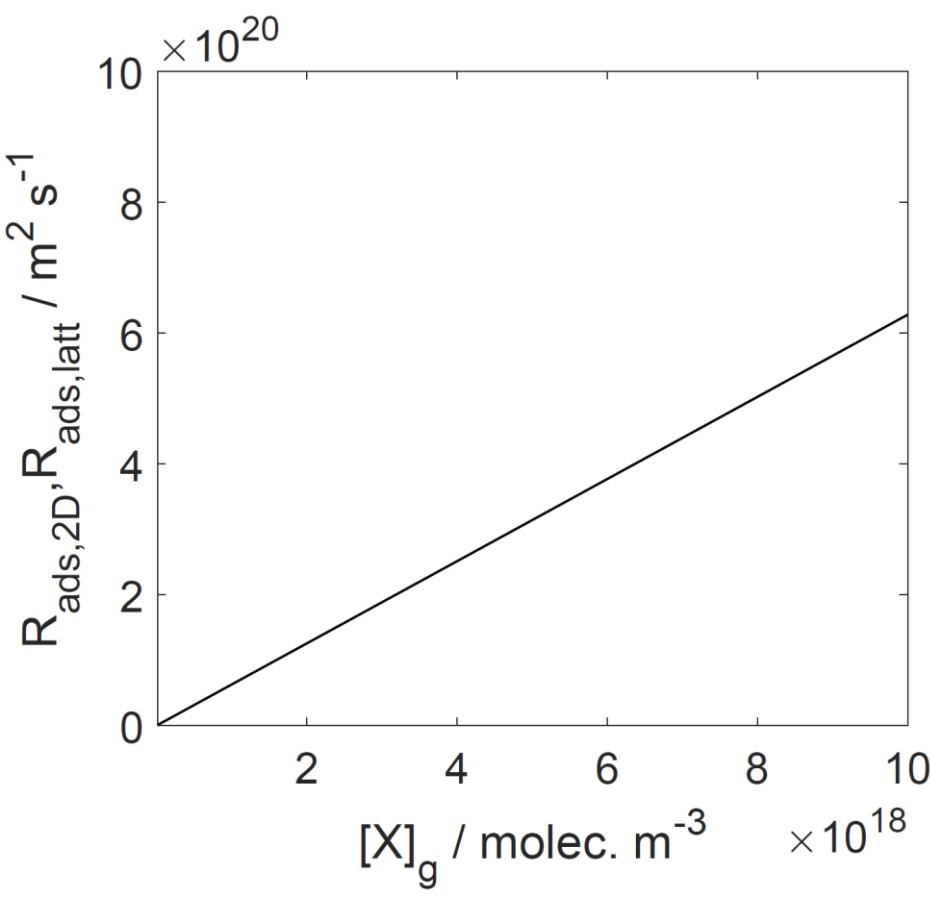

**Figure 11.** The adsorption rate for the case of a 2D ideal gas and 2D ideal lattice gas is depicted. We assume a non-activated adsorption process, $E_b^0 = 0$. Thermodynamic quantities and standard states necessary for calculation are given in Table S1.








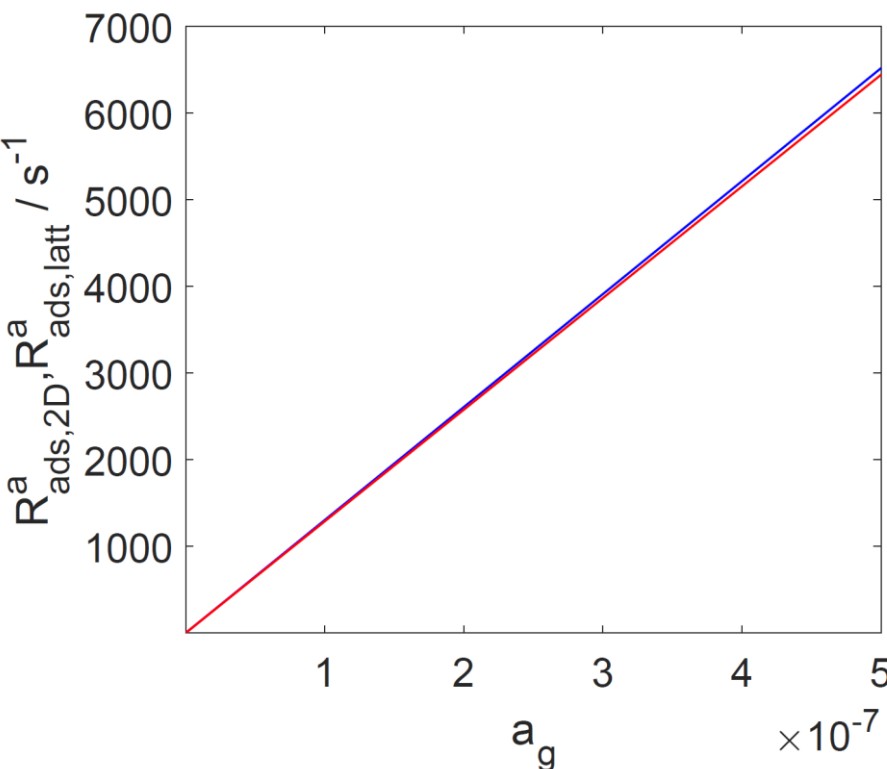

**Figure 12. The activity-based adsorption rates for the case of a 2D ideal gas (blue line) and 2D ideal lattice gas (red line) are depicted. We assume a non-activated adsorption process, $E_b^0 = 0$. Thermodynamic quantities and standard states necessary for calculation are given in Table S1.**



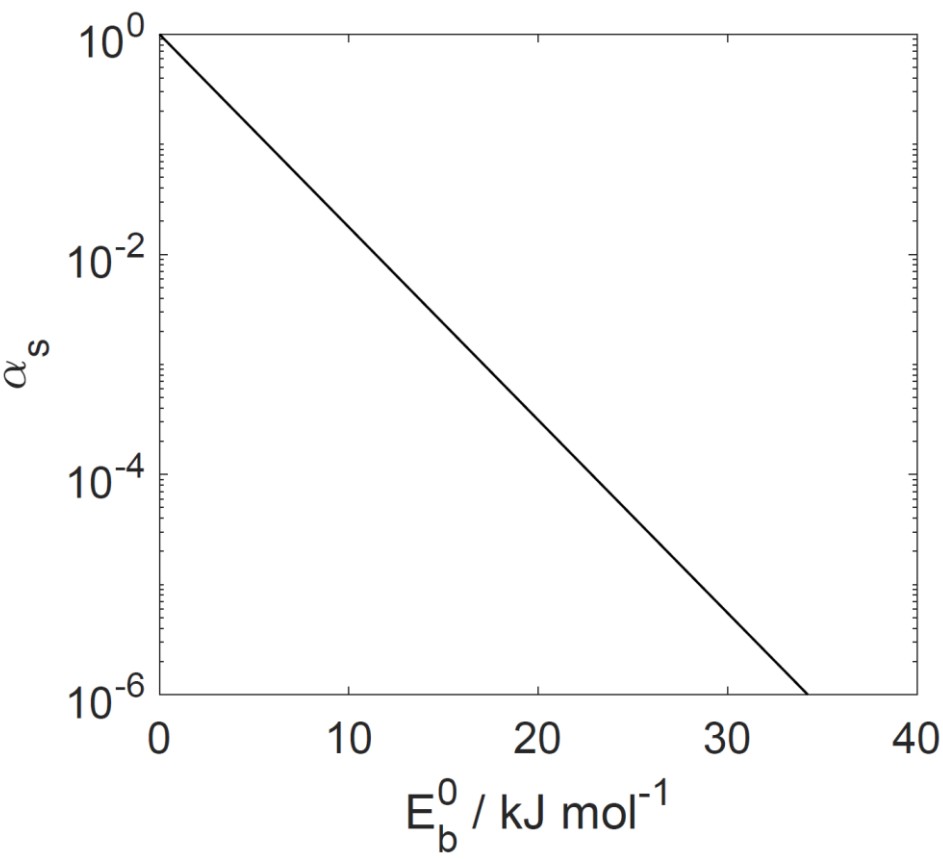

**Figure 13. The dependency of the mass accommodation coefficient, $\alpha_s$, on the adsorption activation energy, $E_b^0$. Thermodynamic**
**quantities and standard states necessary for calculation are given in Table S1.**










**Figure 14. Estimates of $\tau_d$ as a function of temperature applying results from Fig. 3 Blue and red lines represent the 2D ideal gas and 2D ideal lattice gas model, respectively. $E_{des}^0$ varies from 0 to 100 kJ mol$^{-1}$ in 5 kJ mol$^{-1}$ steps from bottom to top and is indicated by numbers on lines. We assume a desorption process without additional barrier, $E_b^0 = 0$. Thermodynamic quantities and standard states necessary for calculation are given in Table S1.**




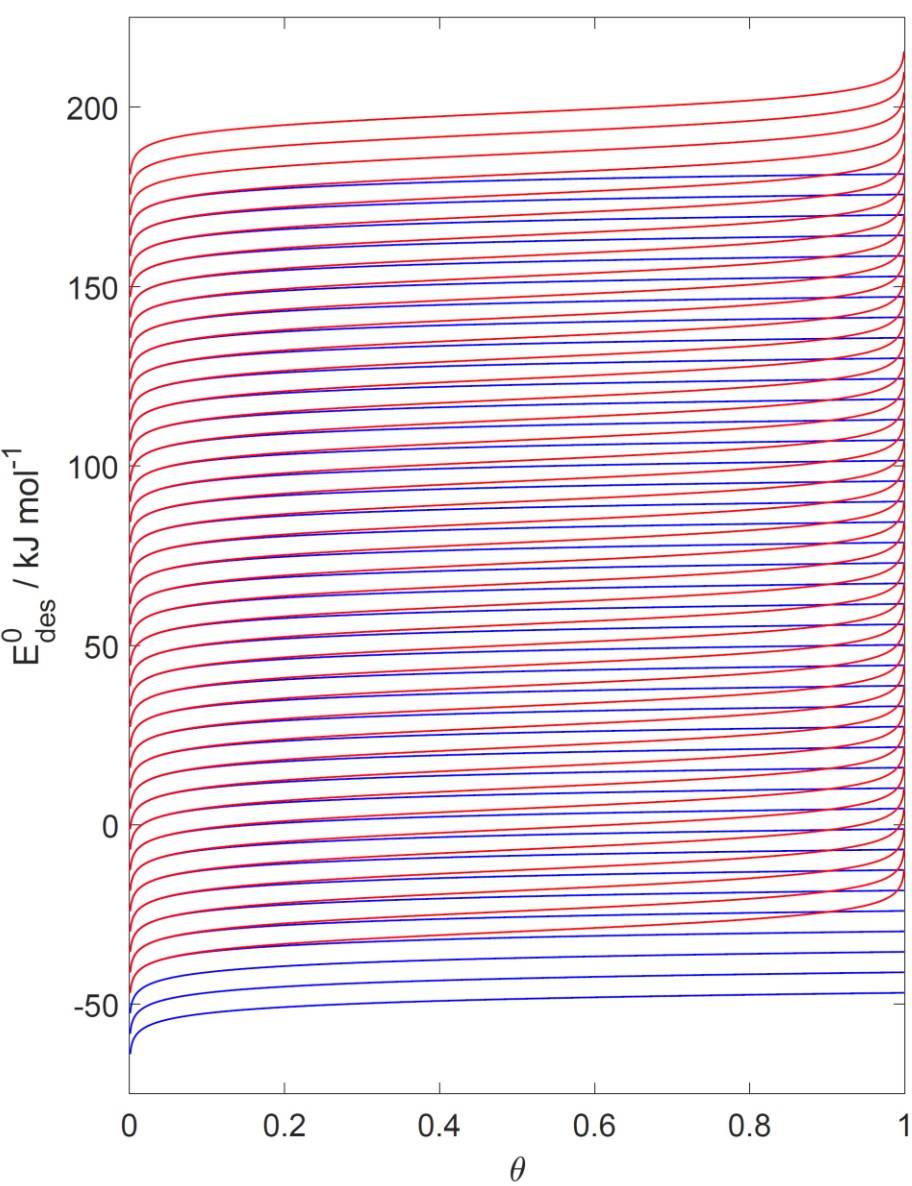

**Figure 15. Estimates of $E_{des}^0$ as a function of coverage for desorption rates from 1 m$^{-2}$ s$^{-1}$ to 10$^{40}$ m$^{-2}$ s$^{-1}$ (from top to bottom) for $T =$ 298 K. Blue and red lines indicate desorption model based on a 2D ideal gas and 2D ideal lattice gas, respectively. We assume a desorption process without additional barrier, $E_b^0 = 0$. Thermodynamic quantities and standard states necessary for calculation are given in Table S1.**