# Peer review of "Technical Note: Adsorption and desorption equilibria from statistical thermodynamics and rates from transition-state theory"

_Atmospheric Chemistry and Physics, 2021_

## Referee Comment (RC1)

Referee's Report on
Submission to ACP by Knopf & Ammann

This is an excellent paper that is certainly important enough to be published in this journal. It provides a good review of statistical thermodynamic treatments of adsorption and desorption equilibria and transition-state theory rates, using an internally self-consistent set of equations (with probably over 100 equations here) and symbology. It emphasizes the importance of using self-consistent approximations and well-defined standard state concentrations if one wants to get accurate numbers from experiments. These issues are of very high importance in atmospheric chemistry research, so it seems a timely and worthy contribution to present to this audience.

However, I suggest several moderately important revisions:

1. The title is "Classical and statistical thermodynamic treatment of adsorption and desorption kinetics and rates", but I think it would define the topics covered better if it were instead: "Statistical thermodynamic treatment of adsorption and desorption equilibria and transition-state theory rates". The authors might consider a title change.

2. Although they have 22 citations to Campbell et al., 2016 in the main text, these citations of that paper are all referring to rather specific points. The fact is, most of the overall treatment here, including the symbology, is very, very similar to or identical to that by Campbell et al., 2016. I think they also should state that broad similarity somewhere very early in this paper. Even more importantly, the derivations occupy 235 Equations in the Supplementary Information (SI), and a huge fraction of these are nearly identical to the equations from Campbell et al., 2016. Yet this SI only cites that paper by Campbell et al. one time, and that citation is made when referring to a single, very specific equation (number (56)). The SI really needs to start with a statement about the very broad similarity between these derivations and their symbology and those presented by Campbell et al..
Eq. (54) in the SI (and probably others) also needs a very specific citation to Campbell et al., since it uses an equation for the configurational entropy derived in the Appendix of Campbell et al. that was a very lengthy derivation, and original.

3. The main part that is significantly different from Campbell et al. is the inclusion here of a real barrier to adsorption in treating adsorption / desorption rates. However, the premise of that addition is not written properly, and will confuse the readers. This appears at the start of Section S2.4, where it says: "The TS for desorption is assumed to exist at some fixed distance from the surface but within a very thin layer of thickness d. The molecules do not have any interactions with the surface, but it is activated to an energy level according to the barrier height above the gas phase reference." This is physically impossible. The molecule cannot both have no interactions with the surface and at the same location be higher in potential energy than the gas

phase by the barrier height shown in Fig. 1.  A better way to imagine this, that would result in the same equations as derived here, would be to say instead: "The TS for adsorption/desorption is assumed to exist at some fixed distance from the surface but within a very thin layer of thickness d, where it experiences an increase in potential energy (relative to the gas phase at infinite separation)  to a maximum (at the barrier of energy $E^0_b$ per mole) due to its interaction with the surface (e.g., due to Pauli repulsion). We further assume for simplicity that, at this TS distance from the surface, or barrier, the potential energy does not depend on the rotational orientation of the molecule nor on the location parallel to the surface (i.e., the barrier is independent of (x,y) coordinates)."  This same problem occurs in the main text in several places, most importantly just after Eq. (7).

4. In my opinion, it will be very confusing to the readers to use of the symbol q to mean potential energy in Fig. 1 and all its associated text in the main paper and SI, because the symbol q is used extensively later for the molecular partition function.  Please change this potential energy symbol to something else like $\in$ or V instead.

5. In several places, the paper used the words "inner energy" for what most textbooks call "internal energy".  I suggest to use "internal energy" instead. I never heard of "inner energy" before.

6.  Prefactors for desorption are mentioned in several places. In at least one of these places (e.g., at the very end of Section 4.1), the authors should mention that Campbell has published a paper that shows some very important experimental trends in prefactors for desorption and their statistical thermodynamic explanation (Kinetic Prefactors of Reactions on Solid Surfaces, C. T. Campbell, L. Árnadóttir and J. R. V. Sellers, *Z. Physikalische Chemie* (in special issue celebrating Eyring / Polanyi paper introducing transition states) 227, 1435–1454 (2013)).

7.  The paper by Campbell et al., 2016 mentioned above has an important paragraph that starts with:
"It is clear from the derivation above that this $1/(1 − \theta)$ factor
in the desorption rate for an ideal 2D lattice gas applies as long
as the sticking probability remains unity. The sticking
probability, $P_s(\theta)$, in general varies with coverage, but is well known
to remain near unity for many adsorbates up to >90% of
saturation coverage because of a Kisliuk-type precursor
mechanism for adsorption (whereby the transiently adsorbed
precursor could easily visit 100 or even 1000 sites before it desorbs). …"
The authors need to read this paragraph carefully and recognize the importance of Kisliuk-type sticking probabilities somewhere in this paper by somehow summarizing the effects discussed here. The coverage dependence of the sticking probability has important consequences for the desorption rate equation, as pointed out initially by John Tully, I guess.  Kisliuk-type sticking is more often observed than straight Langmuir type sticking.

8. On page 3 it says: "… A model that can describe both extremes is, e.g., the ideal hindered translator model (Hill, 1986;Campbell et al., 2016;Sprowl et al., 2016). This new adsorbate model (Sprowl et al., 2016) is not discussed in this study. …" The authors should at least point out here or late in their paper that this paper by Sprowl et al. shows that, in reality, there is a pretty sharp transition between the two extreme models studied here (i.e., the ideal 2D gas and the ideal lattice gas models) when the activation barrier for adsorbate diffusion parallel to the surface goes from just below to just above kT. It is important for readers to recognize that it is this diffusion barrier (relative to kT) that determines which extreme model is most appropriate to use.

---

## Author Comment (AC1)

**Response to reviewer #1**

*This is an excellent paper that is certainly important enough to be published in this journal. It provides a good review of statistical thermodynamic treatments of adsorption and desorption equilibria and transition-state theory rates, using an internally self-consistent set of equations (with probably over 100 equations here) and symbology. It emphasizes the importance of using self-consistent approximations and well-defined standard state concentrations if one wants to get accurate numbers from experiments. These issues are of very high importance in atmospheric chemistry research, so it seems a timely and worthy contribution to present to this audience.*

**We thank the referee for the careful evaluation of our manuscript and for this positive evaluation.**

*However, I suggest several moderately important revisions:*

*1. The title is "Classical and statistical thermodynamic treatment of adsorption and desorption kinetics and rates", but I think it would define the topics covered better if it were instead: "Statistical thermodynamic treatment of adsorption and desorption equilibria and transition-state theory rates". The authors might consider a title change.*

**Thank you for this suggestion. This is a good point. Our rates are derived from transition-state theory. We will follow the referee's suggestion but propose a slightly varied version of the title:**

**"Technical note: Adsorption and desorption equilibria from statistical thermodynamics and rates from transition-state theory"**

**We will further made minor corrections in the abstract to emphasize that we treat adsorption and desorption in this study.**

**We will change line 10-12**

**"When describing gas uptake, gas-to-particle partitioning, and the chemical transformation of aerosol particles the desorption lifetime is a crucial parameter to assess the underlying chemical kinetics such as surface reaction and surface-to-bulk transfer."**

**to**

**"When describing gas uptake, gas-to-particle partitioning, and the chemical transformation of aerosol particles, parameters describing adsorption and desorption rates are crucial to assess the underlying chemical kinetics such as surface reaction and surface-to-bulk transfer."**

**We will change text on line 20 to:**

**"of desorption energies" to "of the variables driving adsorption and desorption"**

*2. Although they have 22 citations to Campbell et al., 2016 in the main text, these citations of that paper are all referring to rather specific points. The fact is, most of the overall treatment here, including the symbology, is very, very similar to or identical to that by Campbell et al., 2016. I think they also should state that broad similarity somewhere very early in this paper.*

**Clearly, (Campbell et al., 2016) served as a great inspiration of our work. We will follow the referee's suggestion and add on line 116:**

**"A great part of those derivations follows the treatment by Campbell et al. (2016)."**

*Even more importantly, the derivations occupy 235 Equations in the Supplementary Information (SI), and a huge fraction of these are nearly identical to the equations from Campbell et al., 2016. Yet this SI only cites that paper by Campbell et al. one time, and that citation is made when referring to a single, very specific equation (number (56)). The SI really needs to start with a statement about the very broad similarity between these derivations and their symbology and those presented by Campbell et al..*

**The referee's point is well taken, and we will add a preamble before section S1 to state the broad similarity to the derivations presented in Campbell et al. (2016). We will add on line 7 in the Supplement (at the beginning):**

**"This Supplement includes all necessary derivations of the thermodynamic equations for 3D ideal gas, 2D ideal gas, 2D ideal lattice gas, and transition state (TS). Many of these equations can be found in Campbell et al. (2016) and in classical textbooks such as by Kolasinski (2012) and Hill (1986). The Supplement consists of the following sections: (S1) Definition of desorption and adsorption equilibrium constants; (S2) Derivation of thermodynamic functions for desorption and adsorption; (S3) Standard molar enthalpies, entropies, and Gibbs free energies; (S4) Derivation of Equilibrium Constants; (S5) Standard molar Gibbs free energy change and equilibrium constant between the 3D ideal gas and the transition state for adsorption; (S6) Adsorption-Desorption Equilibrium."**

*Eq. (54) in the SI (and probably others) also needs a very specific citation to Campbell et al., since it uses an equation for the configurational entropy derived in the Appendix of Campbell et al. that was a very lengthy derivation, and original.*

**We derived Eq. (54) in a more general way following Hill (1986), i.e., starting with the partition function Q (Eq. (40) in supplement). This equation involves application of the Stirling formula (outlined in Appendix of Campbell et al. (2016)). From this we derive the thermodynamic quantities including the entropy. We substitute M and N for surface coverage Ө similar to Campbell et al. (2016).**

**In the supplement, we will add the citation to Hill (1986) at Eq. (40) and add the citation to Campbell et al. (2016) to Eqs. (54) and (135).**

**In the supplement, after Eq. (41), we will add:**

**"Campbell et al. (2016) applied this same transformation when deriving the entropy, here derived further below."**

**In main text we will add the citation to Campbell et al. (2016) to for Eqs. (53) and (55).**

*3. The main part that is significantly different from Campbell et al. is the inclusion here of a real barrier to adsorption in treating adsorption / desorption rates. However, the premise of that addition is not written properly, and will confuse the readers. This appears at the start of Section S2.4, where it says: "The TS for desorption is assumed to exist at some fixed distance from the surface but within a very thin layer of thickness d. The molecules do not have any interactions with the surface, but it is activated to an energy level according to the barrier height above the gas phase reference." This is physically impossible. The molecule cannot both have no interactions with the surface and at the same location be higher in potential energy than the gas phase by the barrier height shown in Fig. 1. A better way to imagine this, that would result in the same equations as derived here, would be to say instead: "The TS for*

*adsorption/desorption is assumed to exist at some fixed distance from the surface but within a very thin layer of thickness d, where it experiences an increase in potential energy (relative to the gas phase at infinite separation) to a maximum (at the barrier of energy E0b per mole) due to its interaction with the surface (e.g., due to Pauli repulsion). We further assume for simplicity that, at this TS distance from the surface, or barrier, the potential energy does not depend on the rotational orientation of the molecule nor on the location parallel to the surface (i.e., the barrier is independent of (x,y) coordinates)."* This same problem occurs in the main text in several places, most importantly just after Eq. (7).

**Thank you for pointing out this confusing statement. The suggested re-phrasing of the original section is excellent. We follow the reviewer's suggestion. Doing so, we realized that our definitions with regard to activation energy and energy barrier for adsorption and desorption processes is not well stated and can be ambiguous in places. We will change the sentence on line 63:**

**"Activated adsorption and desorption processes are treated by considering an additional activation barrier."**

**to**

**"Adsorption is treated as an activated process if an energy barrier exists. Desorption is always treated as an activated process; independent of whether an additional energy barrier exists."**

**In the main text we will change lines starting on 168 (following Eq. (7)):**

**"For an activated adsorption and desorption process the molecule in the TS does not have any interactions with the surface but sits on top of the energy barrier, $q_b^0$ (Fig. 1). In other words, $qb0$ must not dependent on the interaction forces that make up $E_{des}^0$."**

**to**

**"The TS for adsorption/desorption is assumed to exist at some fixed distance from the surface but within a very thin layer of thickness d, where it experiences an increase in potential energy (relative to the gas phase at infinite separation) to a maximum value expressed by the energy barrier $\epsilon_b^0$ due to its interaction with the surface (e.g., due to Pauli repulsion) as outlined in Fig. 1. We further assume for simplicity that at this TS distance from the surface, the potential energy does not depend on the rotational orientation of the molecule nor on the location parallel to the surface."**

**We will change the statement on line 225**

**"Since the TS does not interact with the surface, it is treated as a 2D ideal gas."**

**To**

**"Since the TS is assumed to exist at some fixed distance from the surface but within a very thin layer of thickness, it is treated as a 2D ideal gas, independent of the choice of model for the adsorbate."**

**We also corrected Fig. 1.**

**On line 412 we will add:**

"The TS for desorption is assumed to exist at some fixed distance from the surface but within a very thin layer of thickness $d$, where it experiences an increase in potential energy to a maximum value expressed by the energy barrier $\epsilon_b^0$ due to its interaction with the surface as outlined above."

As suggested, we will change the supplement section 2.4, lines 178 to 180, to:

"The TS for adsorption/desorption is assumed to exist at some fixed distance from the surface but within a very thin layer of thickness d, where it experiences an increase in potential energy (relative to the gas phase at infinite separation) to a maximum value expressed by the energy barrier $\epsilon_b^0$ per mole due to its interaction with the surface (e.g., due to Pauli repulsion). We further assume for simplicity that, at this TS distance from the surface, or at that energy barrier, the potential energy does not depend on the rotational orientation of the molecule nor on the location parallel to the surface (i.e., the energy barrier is independent of x,y-coordinates)."

We further corrected the following statements in the main text:

Line 146: We will change "…when describing the activated and non-activated adsorption and desorption processes."

To

"…when describing adsorption and desorption processes."

Line 178: We will change "For non-activated adsorption or desorption processes, $E_b^0 = 0$, and all equations simplify accordingly."

To

"In the absence of an energy barrier for adsorption and desorption, i.e., $E_b^0 = 0$, and all equations simplify accordingly. Note, however, and as mentioned above, in absence of a barrier, the desorption process remains an activated process with $E_{des,act}^0 = E_{des}^0$."

Line 800 and 817: We will change "activation barrier" to "energy barrier".

*4. In my opinion, it will be very confusing to the readers to use of the symbol q to mean potential energy in Fig. 1 and all its associated text in the main paper and SI, because the symbol q is used extensively later for the molecular partition function. Please change this potential energy symbol to something else like $\in$ or V instead.*

Agreed. We exchanged "q" for $\epsilon$ , as suggested.

*5. In several places, the paper used the words "inner energy" for what most textbooks call "internal energy". I suggest to use "internal energy" instead. I never heard of "inner energy" before.*

Yes, we changed "inner energy" to "internal energy" in all places.

*6. Prefactors for desorption are mentioned in several places. In at least one of these places (e.g., at the very end of Section 4.1), the authors should mention that Campbell has published a paper that shows some very important experimental trends in prefactors for desorption and their statistical thermodynamic explanation (Kinetic Prefactors of Reactions on Solid Surfaces, C. T. Campbell, L. Árnadóttir and J. R. V. Sellers, Z. Physikalische Chemie (in special issue celebrating Eyring / Polanyi paper introducing transition states) 227, 1435–1454 (2013)).*

**We will add the reference (Campbell et al., 2013) on line 79 when first mentioning the impact on the pre-exponential factor A.**

**As suggested, we will add at the end of section 4.1, line 498:**

**"Campbell (2013) showed that the observed variations in A for different adsorbates can be well described by a linear correlation between adsorbate entropies and gas-phase entropies provided the adsorbate's surface residence time is less than ∼1000 s. The underlying explanation is that the gas molecule's motions in z direction are arrested (i.e., frustrated rotational and translational modes) resulting in a steep interaction potential well in the z direction, better described by a hindered translator model."**

*7. The paper by Campbell et al., 2016 mentioned above has an important paragraph that starts with:*

*"It is clear from the derivation above that this $1/(1 − \vartheta)$ factor in the desorption rate for an ideal 2D lattice gas applies as long as the sticking probability remains unity. The sticking probability, $Ps(\vartheta)$, in general varies with coverage, but is well known to remain near unity for many adsorbates up to >90% of saturation coverage because of a Kisliuk-type precursor mechanism for adsorption (whereby the transiently adsorbed precursor could easily visit 100 or even 1000 sites before it desorbs). …"*

*The authors need to read this paragraph carefully and recognize the importance of Kisliuk-type sticking probabilities somewhere in this paper by somehow summarizing the effects discussed here. The coverage dependence of the sticking probability has important consequences for the desorption rate equation, as pointed out initially by John Tully, I guess. Kisliuk-type sticking is more often observed than straight Langmuir type sticking.*

**We are aware of the suggested interpretation of the desorption rate scaling with θ/(1-θ) using the Kisliuk-type precursor mechanism. However, our derived formalism only considers physisorption (which we will now emphasize in the introduction) and follows the definition by Kolb (2010) regarding the mass accommodation coefficient $\alpha_s$ designating a physisorptive processes. Thus, being different from a sticking coefficient, for which often inconsistent definitions are used. Also, we believe that the treatment of the Kisliuk model in terms of the statistical thermodynamics would be more complex than considered here, as it effectively encompasses a two-layer model.**

**In the presented framework, we ascribe the observed non-linearity of $\frac{R_{des}}{\mathcal{A}}$ of the 2D ideal lattice gas, being proportional to θ∕(1-θ), as a direct result of the non-linear increase of the configurational entropy.**

**The referee's point is well-taken and we will acknowledge alternative descriptions that involve more complex configurations involving already adsorbed molecules as discussed by Kisliuk and Tully.**

**We will add on line 58:**

"In the context of atmospheric sciences, adsorption is commonly described by the surface accommodation coefficient, $\alpha_s$, which is the probability that a molecule undergoing a gas kinetic collision is adsorbed at the surface (see overview and definitions by Kolb et al. (2010)). For desorption, according to…"

We will add on line 79 and refined the subsequent sentence:

"The same applies to the surface accommodation coefficient, which is not referring to the adsorbate model. Once the pre-exponential factor $A$ for desorption is expressed in terms of the free energy of activation (Campbell et al., 2016;Donaldson et al., 2012;Kolasinski, 2012;Campbell et al., 2013), the choice of adsorbate model and standard states has a significant impact on the values of the pre-exponential factor $A$ and thus $\tau_d$."

We will add on line 99:

"The presented framework only considers physisorptive processes, within the general framework of treating adsorption in atmospheric chemistry (Kolb et al., 2010;Pöschl et al., 2007)."

We will add on line 116:

"In this study, the surface accommodation coefficient follows the definition by (Kolb et al., 2010) valid for physisorptive processes and consistent with the Langmuir adsorption description but not necessarily the same as the sticking coefficient used in surface sciences or catalysis, which is often inconsistently defined, and sometimes or sometimes not lumps physisorption and chemisorption. There are alternative descriptions such as the Kisliuk-type precursor mechanism that consider more complex configurations of the adsorbate (Kisliuk, 1957;Tully, 1994;Campbell et al., 2016), not discussed in this study.

We will add on line 612 and adjust subsequent sentence:

"Adsorption is treated as a physisorptive process, but might exert a non-zero energy barrier $E_b^0$ for activated adsorption. We derive the adsorption rates of gas molecules transferring into the 2D ideal and 2D ideal lattice gas absorbates. The adsorption proceeds via the TS, which is treated as a 2D ideal gas, as in the case of desorption."

To avoid confusion we will change on line 714:

"…Hertz-Knudsen expression of the flux of molecules attempting to stick on surface atoms."

to

"…the Hertz-Knudsen expression of the flux of molecules impinging on surface atoms."

On line 727, we note once more possible alternative descriptions. We will change on lines 727:

"We can now look at the surface accommodation coefficient, $\alpha_s$, which is operationally defined as the ratio between the adsorption rate and the gas-kinetic collision rate (Kolb et al., 2010;Ammann et al., 2013;Crowley et al., 2013) in the description of the adsorption rate following "

To

"We can now look at the surface accommodation coefficient, $\alpha_s$, which is operationally defined as the ratio between the adsorption rate and the gas-kinetic collision rate (Kolb et al., 2010;Ammann et al., 2013;Crowley et al., 2013) considering only physisorptive processes, not accounting for possibly more

complex configurations involving already adsorbed molecules (Kisliuk, 1957;Tully, 1994;Campbell et al., 2016). The description of the adsorption rate follows as"

We will add on line 836:

"The observed non-linearity of $\frac{R_{des}}{\mathcal{A}}$ of the 2D ideal lattice gas, being proportional to $(\theta/(1-\theta))$, is a direct result of the non-linear increase of the configurational entropy (e.g., Eqs. (53) and (98))."

*8. On page 3 it says: "… A model that can describe both extremes is, e.g., the ideal hindered translator model (Hill, 1986;Campbell et al., 2016;Sprowl et al., 2016). This new adsorbate model (Sprowl et al., 2016) is not discussed in this study. …" The authors should at least point out here or late in their paper that this paper by Sprowl et al. shows that, in reality, there is a pretty sharp transition between the two extreme models studied here (i.e., the ideal 2D gas and the ideal lattice gas models) when the activation barrier for adsorbate diffusion parallel to the surface goes from just below to just above kT. It is important for readers to recognize that it is this diffusion barrier (relative to kT) that determines which extreme model is most appropriate to use.*

This is a good point, and we add a sentence referring to the Sprowl et al. study to explain the transition between the different adsorbate models. We will add on line 91 and clarify subsequent sentences:

"Which of the two models, the ideal 2D gas and the ideal lattice gas, are realized will depend on the activation barrier for adsorbate diffusion parallel to the surface. If this activation barrier is above $k_B T$ (Boltzmann constant times temperature), the ideal lattice gas model is the preferred model, whereas if it is below $k_B T$, diffusion of adsorbates parallel to the surface can commence and the adsorption is described by an ideal 2D gas (Sprowl et al., 2016). The hindered translator model (Sprowl et al., 2016) is not discussed in this study. It will be shown that the choice of adsorbate model and corresponding standard states will result in different equilibrium constants, pre-exponential factors, and, thus, desorption rates, but counter-intuitively, in the same adsorption rates. Ultimately, the choice of the adsorbate model…"

**References**

Ammann, M., Cox, R. A., Crowley, J. N., Jenkin, M. E., Mellouki, A., Rossi, M. J., Troe, J., and Wallington, T. J.: Evaluated kinetic and photochemical data for atmospheric chemistry: Volume VI - heterogeneous reactions with liquid substrates, Atmos. Chem. Phys., 13, 8045-8228, 10.5194/acp-13-8045-2013, 2013.

Campbell, C. T., Arnadottir, L., and Sellers, J. R. V.: Kinetic Prefactors of Reactions on Solid Surfaces, Z. Phys. Chemie-Int. J. Res. Phys. Chem. Chem. Phys., 227, 1435-1454, 10.1524/zpch.2013.0395, 2013.

Campbell, C. T., Sprowl, L. H., and Arnadottir, L.: Equilibrium Constants and Rate Constants for Adsorbates: Two-Dimensional (2D) Ideal Gas, 2D Ideal Lattice Gas, and Ideal Hindered Translator Models, J. Phys. Chem. C, 120, 10283-10297, 10.1021/acs.jpcc.6b00975, 2016.

Crowley, J. N., Ammann, M., Cox, R. A., Hynes, R. G., Jenkin, M. E., Mellouki, A., Rossi, M. J., Troc, J., and Wallington, T. J.: Evaluated kinetic and photochemical data for atmospheric chemistry: Volume V - heterogeneous reactions on solid substrates (vol 10, pg 9059, 2010), Atmos. Chem. Phys., 13, 7359-7359, 10.5194/acp-13-7359-2013, 2013.

Donaldson, D. J., Ammann, M., Bartels-Rausch, T., and Pöschl, U.: Standard States and Thermochemical Kinetics in Heterogeneous Atmospheric Chemistry, J. Phys. Chem. A, 116, 6312-6316, 10.1021/jp212015g, 2012.

Hill, T. L.: An Introduction to Statistical Thermodynamics, Dover Publications, Inc., New York, 501 pp., 1986.

Kisliuk, P.: The sticking probabilities of gases chemisorbed on the surfaces of solids, J. Phys. Chem. Solids, 3, 95-101, 10.1016/0022-3697(57)90054-9, 1957.

Kolasinski, K. W.: Surface Science: Foundations of Catalysis and Nanoscience, 3rd ed., John Wiley & Sons, Ltd., West sussex, United Kingdom, 556 pp., 2012.

Kolb, C. E., Cox, R. A., Abbatt, J. P. D., Ammann, M., Davis, E. J., Donaldson, D. J., Garrett, B. C., George, C., Griffiths, P. T., Hanson, D. R., Kulmala, M., McFiggans, G., Pöschl, U., Riipinen, I., Rossi, M. J., Rudich, Y., Wagner, P. E., Winkler, P. M., Worsnop, D. R., and O' Dowd, C. D.: An overview of current issues in the uptake of atmospheric trace gases by aerosols and clouds, Atmos. Chem. Phys., 10, 10561-10605, 2010.

Sprowl, L. H., Campbell, C. T., and Arnadottir, L.: Hindered Translator and Hindered Rotor Models for Adsorbates: Partition Functions and Entropies, J. Phys. Chem. C, 120, 9719-9731, 10.1021/acs.jpcc.5b11616, 2016.

Tully, J. C.: The dynamics of adsorption and desorption, Surf. Sci., 299, 667-677, 10.1016/0039-6028(94)90688-2, 1994.

---

## Author Comment (AC2)

**Response to reviewer #2**

*This technical note derives the desorption rates for adsorbed species using thermodynamics and statistical mechanics. The authors do these derivations with two models of the sorbed species: a 2D ideal gas and a 2D ideal lattice gas. The authors demonstrate that these models will retrieve very different values of e.g. desorption energy and timescale. This work is valuable and relevant to the study of atmospheric chemistry.*

**We thank the referee for the careful evaluation of our manuscript and for this positive evaluation.**

*The authors aim to save space and increase the readability of the manuscript by moving many of the equations to the Supplement. This was a good decision, however the reader who is reliant on the additional information found in the supplement is also the reader who needs to be guided most closely through those equations. The manuscript would benefit from additional text informing readers when equations and assumptions detailed in the Supplement have been brought into the main text. This added text should specify the range of equation numbers over which a derivation has been conducted, and not simply list the equation number of the final result.*

**We are aware that following the equations is challenging. As the referee requested, for crucial derivations in the main text we will provide more information on which supplemental equations are needed to derive the shown results.**

**This concerns references to the Supplement on lines:**

**Line 269: (see Supplement Eqs. (119-123) with Supplement Eqs. (86), (89), (91), (93), (97), (98))**

**Line 285: We change "Supplement Eq. (119ff.)" to "Supplement Eqs. (119-121)".**

**Line 299: (Supplement Eq. (121) with Supplement Eq. (89))**

**Line 330: (see Supplement Eqs. (131, 133, 134) with Supplement Eqs. (86), (89), (91), (100), (103), 105))**

**Line 478: (Supplement Eq. (148) with Supplement Eqs. (108) and (93))**

**Line 563: (Supplement Eqs. (108) and (171))**

**Line 647: (Supplement Eqs. (193-199)**

**Line 754: … given in the Supplement (Eqs. (216-220)).**

**Line 761: …is outlined in the Supplement Eqs. (221-225).**

**Line 770: … given in the Supplement Eqs. (226-230).**

**Line 780: … is given in the Supplement Eqs. (231-235).**

*237: It seems like the atmospheric chemists most likely to rely on this work are those conducting lab experiments of adsorption and desorption, where coverages may not be small. The authors should provide a quantitative estimate of the level of coverage where they believe the 2D ideal gas is no longer an appropriate model to use.*

Thank you for this comment. The text may have left the reader in confusion. In short, the level of coverage will not define the application of either model. Referee #1 requested to add a description under which conditions the 2D ideal gas model and the 2D ideal lattice gas model manifest. This information will be added on line 91:

"Which of the two models, the ideal 2D gas and the ideal lattice gas, are realized will depend on the activation barrier for adsorbate diffusion parallel to the surface. If this activation barrier is above $k_BT$ (Boltzmann constant times temperature), the ideal lattice gas model is the preferred model, whereas if it is below $k_BT$, diffusion of adsorbates parallel to the surface can commence and the adsorption is described by an ideal 2D gas (Sprowl et al., 2016)."

The interaction between the adsorbate and the adsorbent is different for the different models. Surface coverage will not define which model to use. We wanted to point out the fact that mathematically the 2D ideal gas has no limit on surface coverage since no maximum site number is implemented. Though, coverages above $10^{14}$ cm$^{-2}$ are likely not physically reasonable since those would go beyond a monolayer. As mentioned on line 235, this approach is conceptually similar to the 3D ideal gas that, by definition, does not have a limit of the applied gas species concentration (or pressure) below which the assumption of ideality is correct.

We will change the sentence on line 235:

"To remain within physically reasonable bounds, all equations in conjunction with the 2D ideal gas model relate to conditions of small surface coverages only."

To

"To remain within physically reasonable bounds, all equations in conjunction with the 2D ideal gas model relate to conditions of surface coverages below a typical monolayer coverage of about $10^{14}$ cm$^{-2}$."

*285: "Supplement Eq. (119ff.)" is likely a broken equation number link?*

We wanted to point the reader to the derivations of the free energy, enthalpy, and entropy used in Eq. 35. We change

"Supplement Eq. (119ff.)" to "Supplement Eqs. (119-121)".

*765 & 792 & 808: Can these derivations be moved to the SI?*

Regarding derivations following line 779:

We would prefer to leave those derivations in this section to keep consistency in section 6. Its structure is calculating the equilibrium first for 2D ideal gas based on concentrations followed by activities. Then (line 764), this derivation is repeated for the 2D ideal lattice gas. First on the basis of concentrations and then on the basis of activities. Leaving out the derivations on line 765 would interrupt the line of thought for the entire section.

**However, we will shorten the derivations on lines 792 and 808 by not showing all necessary intermediate steps. The full derivations have been moved to the supplement section 6 after the last derivation Eq. (235).**

*Fig 2: extend axes to zero*

**Figure 2 will be corrected as suggested.**

*Fig 5. I recommend labelling each line with its E0des value*

**Figure 5 will be corrected as suggested.**

*Fig. 15. I recommend reducing the number of lines plotted and labelling each line with its desorption rate*

**Figure 15 will be corrected as suggested.**